# A Non-Asymptotic Moreau Envelope Theory for High-Dimensional Generalized Linear Models

**Lijia Zhou**[*]
University of Chicago
zlj@uchicago.edu

**Frederic Koehler**[*]
Stanford University
fkoehler@stanford.edu

**Pragya Sur**
Harvard University
pragya@fas.harvard.edu

**Danica J. Sutherland**
University of British Columbia & Amii
dsuth@cs.ubc.ca

**Nathan Srebro**
Toyota Technological Institute at Chicago
nati@ttic.edu

Collaboration on the Theoretical Foundations of Deep Learning (deepfoundations.ai)

## Abstract

We prove a new generalization bound that shows for any class of linear predictors in Gaussian space, the Rademacher complexity of the class and the training error under any continuous loss $\ell$ can control the test error under all *Moreau envelopes* of the loss $\ell$. We use our finite-sample bound to directly recover the "optimistic rate" of Zhou et al. (2021) for linear regression with the square loss, which is known to be tight for minimal $\ell_2$-norm interpolation, but we also handle more general settings where the label is generated by a potentially misspecified multi-index model. The same argument can analyze noisy interpolation of max-margin classifiers through the squared hinge loss, and establishes consistency results in spiked-covariance settings. More generally, when the loss is only assumed to be Lipschitz, our bound effectively improves Talagrand's well-known contraction lemma by a factor of two, and we prove uniform convergence of interpolators (Koehler et al. 2021) for all smooth, non-negative losses. Finally, we show that application of our generalization bound using localized Gaussian width will generally be sharp for empirical risk minimizers, establishing a non-asymptotic Moreau envelope theory for generalization that applies outside of proportional scaling regimes, handles model misspecification, and complements existing asymptotic Moreau envelope theories for M-estimation.

## 1  Introduction

Modern machine learning models often contain more parameters than the number of training samples. Despite the capacity to overfit, training these models without explicit regularization has been empirically shown to achieve good generalization performance (Neyshabur et al. 2015; C. Zhang et al. 2017; Belkin et al. 2019). On the theoretical side, the study of minimal-norm interpolation has revealed fascinating phenomena that challenge traditional understandings of machine learning.

We now have a better understanding of how properties of the data distribution and algorithmic bias can affect generalization in high-dimensional linear regression. For example, data with a spiked covariance structure can ensure that the test error of ridge regression will be approximately constant once the regularization strength is small enough for the model to fit the signal (Zhou et al. 2021; Tsigler and Bartlett 2020), contradicting the classical U-shaped curve expected from arguments about the bias-variance tradeoff. Surprisingly, even when the signal is sparse, the risk of the minimal-$\ell_1$ norm interpolator can be shown to converge much slower to the Bayes risk than the minimal-$\ell_2$

---

[*]These authors contributed equally.

norm interpolator in the junk feature setting (Chatterji and Long 2021; Koehler et al. 2021). In contrast, the minimal-$\ell_2$ norm interpolator fails to achieve consistency in the isotropic setting, while the minimal-$\ell_1$ norm interpolator is consistent with sparsity but suffers from an exponentially slow rate in the number of parameters $d$ (G. Wang et al. 2021; Muthukumar et al. 2020). However, we can still achieve the minimax rate with minimal-$\ell_p$ norm interpolators with $p$ extremely close to 1 (Donhauser et al. 2022).

In fact, many of the intriguing phenomena from the work above may be understood using the norm of a predictor; localized notions of uniform convergence have emerged as essential tools for doing so. Compared to other techniques, uniform convergence analyses can have the benefit of requiring neither particular proportional scaling regimes nor closed-form expressions for the learned model, since only an approximate estimate of its complexity is needed. Despite uniform convergence's potential for wider applicability, though, work in this area has mostly focused on linear regression settings with strong assumptions: that the conditional expectation of the label is linear with respect to the features, and that the residual has constant variance. In contrast, classical agnostic learning guarantees established by uniform convergence usually need only much weaker assumptions on the data distribution, and apply to a broader range of losses and function classes. For example, Srebro et al. (2010) show that bounds with an "optimistic rate" hold generally for any smooth, nonnegative loss, though the hidden logarithmic factor in their result is too loose for explaining noisy interpolation; this was recently addressed by Zhou et al. (2021) in the special case of well-specified linear regression.

In this work, we take a step further towards agnostic interpolation learning and consider a high-dimensional generalized linear model (GLM) setting where the label is generated by a potentially misspecified model. We show a new generalization bound that allows us to use the Moreau envelopes of any continuous loss function as an intermediate tool. By optimizing over the smoothing parameter to balance the approximation and generalization errors, our general Moreau envelope theory yields sharp non-asymptotic generalization bounds in a wide variety of settings. Applying to linear regression with the square loss, we recover the optimistic rate of Zhou et al. (2021) and show that it can more generally be extended to handle model misspecification, such as nonlinear trends and heteroskedasticity. The generality of our result comes from the fact that taking the Moreau envelope of the square loss only scales by a constant; this property alone suffices to obtain a generalization guarantee in terms of the original square loss. The squared hinge loss enjoys the same property, and hence a completely analogous argument shows an optimistic rate in that setting. Combined with an analysis of the margin, we show a novel consistency result for max-margin classifiers.

More generally, we apply the Moreau envelope theory to obtain a generic bound for any Lipschitz loss and smooth, nonnegative loss with sharp constants. Looking specifically at the test error of an Empirical Risk Minimizer (ERM), we show our generalization bound with localized Gaussian width will be asymptotically sharp even when overfitting is not necessarily benign, yielding a version of the asymptotic Moreau envelope framework for analyzing M-estimators (El Karoui et al. 2013; Bean et al. 2013; Donoho and Montanari 2016; Thrampoulidis et al. 2018; Sur and Candès 2019) but for the problem of generalization. Numerical simulations on a variety of feature distributions and label generating processes confirm the wide applicability of our theory.

## 2   Related Work

The Moreau envelope has been useful in characterizing asymptotic properties of M-estimators in linear models (Bean et al. 2013; El Karoui et al. 2013; Donoho and Montanari 2016; El Karoui 2018; Thrampoulidis et al. 2018) and logistic regression (Sur and Candès 2019; Sur et al. 2019; Candès and Sur 2020; Salehi et al. 2019; Zhao et al. 2022). This theory focuses on estimation and inference under proportional asymptotics, rather than generalization, and does not provide any non-asymptotic results. Loureiro et al. 2021 studied proportional asymptotics in a student-teacher model similar to the one we consider, and gave finite-sample bounds on the speed of convergence to the limit. Like other works using the CGMT framework, they give formulas directly in terms of the regularization parameter used in M-estimation, and they also give results to characterize the distribution of the output of the estimator. Our work does not study the properties of the M-estimator in this level of detail. Instead, our approach is fundamentally based on establishing a nonasymptotic generalization bound. This enables our results to apply beyond proportional scaling (which is crucial to analyze benign overfitting), and yield guarantees for predictors besides the exact optima of convex M-estimation objectives. Another difference is that the class of generalized linear models we consider

is somewhat broader: we allow multi-index models for the teacher (which may also be possible with their approach) and our loss function does not need to be convex.

For linear regression, Bartlett et al. (2020) identify nearly-matching necessary and sufficient conditions for the consistency of minimal-$\ell_2$ norm interpolation; their subsequent work (Tsigler and Bartlett 2020) shows generalization bounds for overparametrized ridge regression. Following their work, Negrea et al. (2020) and Zhou et al. (2020) explore the role of uniform convergence, including showing that uniformly bounding the difference between training error and test error fails to explain interpolation learning. Zhou et al. (2020) argue, however, that uniform convergence *of interpolators* is sufficient to establish consistency in a toy example. Koehler et al. (2021) extend their result to arbitrary data covariance and norm, recovering the benign overfitting conditions of Bartlett et al. (2020) as well as proving novel consistency results for the minimal-$\ell_1$ norm interpolator. Based on this uniform convergence framework, G. Wang et al. (2021) establish tight bounds for the minimal-$\ell_1$ norm interpolator under a sparse signal with isotropic data. Earlier work (Ju et al. 2020; Chinot et al. 2020; Li and Wei 2021) also studied the minimal-$\ell_1$ norm interpolator, without showing consistency. Though the minimal-$\ell_1$ norm interpolator suffers from an exponentially slow rate, Donhauser et al. (2022) show the minimal-$\ell_p$ norm interpolator can achieve faster rates with $p$ close to 1. Zhou et al. (2021) show a risk-dependent ("localized") bound that extends the uniform convergence of interpolators guarantee to predictors with arbitrary training loss, and used it to establish generalization for regularized estimators such as Ridge and LASSO. Our Moreau envelope theory builds on the techniques developed in this line of work to apply uniform convergence in the interpolation regime.

In terms of requirements on the data distribution, Bartlett et al. (2020) and Tsigler and Bartlett (2020) only require the feature vector to be sub-Gaussian, but assume a well-specified linear model for the conditional distribution of the label. The uniform convergence-based works also assume a well-specified linear model, but the assumptions are more restrictive in the sense that the marginal distribution of the feature needs to be *exactly* Gaussian because their proof techniques rely on the Gaussian Minimax Theorem (GMT). Our Moreau envelope theory's application to linear regression significantly relaxes the assumption on the label generating process, though it is still constrained by the Gaussian data assumption. Shamir (2022) also studies model misspecification in linear regression, but allows non-Gaussian features, and shows that benign overfitting does not necessarily occur in the most general setting, even with a spiked-covariance structure (see his Example 1).

For linear classification, Muthukumar et al. (2021) analyze $\ell_2$ max-margin classifier by connecting to minimal-norm interpolation in regression. Similarly, our analysis in the classification case depends on the fact the squared hinge loss goes through the same transformation as the square loss under smoothing by Moreau envelope. Donhauser et al. (2022) prove generalization bounds for $\ell_p$ max-margin classifiers in the isotropic setting and do not consider the spiked-covariance case. Deng et al. (2021), Montanari et al. (2019), and Liang and Sur (2020) derive exact expressions for the asymptotic prediction risk of the $\ell_2$ and $\ell_p$ (with $p \in [1, 2)$) max-margin classifiers. Though their proof techniques also rely on the GMT, our approaches are drastically different. We use GMT in order to show uniform convergence for a class of predictors and establish a non-asymptotic bound, whereas their results are asymptotic and assume a proportional scaling limit. This is a key distinction, because overfitting usually cannot be benign with proportional scaling (e.g. Donhauser et al. 2022, Proposition 1). Similar lower bounds have also been shown in the context of linear regression (Muthukumar et al. 2020; G. Wang et al. 2021; Zhou et al. 2020).

Some concurrent works have obtained consistency results for max-margin classification in the spiked covariance setting. In particular, the May 2022 version of the work by Shamir (2022) also studies convergence to the minimum of the squared hinge loss, and obtains consistency under conditions similar to the benign covariance condition of Bartlett et al. (2020). During preparation of this manuscript we learned of concurrent work by Montanari et al., not yet publicly available, which also studies consistency results for classification. Comparing our Corollary 3 to Shamir (2022), their result applies to some non-Gaussian settings, but in the Gaussian setting their result is not as general as ours. (Combining Assumptions 1 and 2 of Theorem 7 there, they require the norm of the data to be bounded, whereas our Corollary 3 applies even if $o(n)$ eigenvalues of $\Sigma$ grow arbitrarily quickly with $n$.) More conceptually, our result follows from a norm-based generalization bound that applies to all predictors and outside of the "benign overfitting" conditions, generalizing the result of Koehler et al. (2021) and unlike the analysis of prior work.

## 3 Problem Formulation

**GLM setting.** Given a continuous loss function $f : \mathbb{R} \times \mathbb{R} \to \mathbb{R}$ and i.i.d. sample pairs $(x_i, y_i)$ from some data distribution $\mathcal{D}$, we can learn a linear model $(\hat{w}, \hat{b})$ by minimizing the empirical loss $\hat{L}_f$ with the goal of achieving small population loss $L_f$:

$$\hat{L}_f(w, b) = \frac{1}{n} \sum_{i=1}^{n} f(\langle w, x_i \rangle + b, y_i), \quad L_f(w, b) = \mathop{\mathbb{E}}_{(x,y) \sim \mathcal{D}} f(\langle w, x \rangle + b, y). \tag{1}$$

**Multi-index model.** We assume that the data distribution $\mathcal{D}$ over $(x, y)$ is such that

1. $x \sim \mathcal{N}(0, \Sigma)$ is a centered Gaussian with unknown covariance matrix $\Sigma$.

2. There are unknown weight vectors $w_1^*, ..., w_k^* \in \mathbb{R}^d$ such that the $\Sigma^{1/2} w_i^*$ are orthonormal, a function $g : \mathbb{R}^{k+1} \to \mathbb{R}$, and a random variable $\xi \sim \mathcal{D}_\xi$ independent of $x$ (not necessarily Gaussian) such that
$$\eta_i = \langle w_i^*, x \rangle, \quad y = g(\eta_1, ..., \eta_k, \xi). \tag{2}$$

We can assume that the distribution of $x$ is centered without loss of generality since presence of a mean term simply corresponds to changing the bias term $b$: $\langle w, x \rangle + b = \langle w, x - \mu \rangle + (b - \langle w, \mu \rangle)$. We can also assume that $\Sigma^{1/2} w_1^*, ..., \Sigma^{1/2} w_k^*$ are orthonormal without loss of generality since we have not imposed any assumption on the link function $g$. The multi-index model includes well-specified linear regression, by setting $k = 1$ and $g(\eta, \xi) = \eta + \xi$. It also allows nonlinear trends and heteroskedasticity (such as the model in Figure 1) by changing the definition of $g$. Since $g$ need not be continuous, the label $y$ can be binary, as in linear classification.

## 4 Moreau Envelope Generalization Theory

Our theory vitally depends on the Moreau envelope, defined as follows.

**Definition 1.** The *Moreau envelope* of $f : \mathbb{R} \times \mathbb{R} \to \mathbb{R}$ with parameter $\lambda \in \mathbb{R}^+$ is defined as the function $f_\lambda : \mathbb{R} \times \mathbb{R} \to \mathbb{R}$ given by

$$f_\lambda(\hat{y}, y) = \inf_u f(u, y) + \lambda(u - \hat{y})^2. \tag{3}$$

The Moreau envelope can be viewed as a smooth approximation to the original function $f$: in our parameterization, smaller $\lambda$ corresponds to more smoothing. The map that outputs the minimizer $u$, known as the *proximal operator*, plays an important role in convex analysis (Parikh and Boyd 2014; Bauschke, Combettes, et al. 2011).

Our general theory, as stated in Theorem 1 below, essentially upper bounds the generalization gap between the population Moreau envelope $L_{f_\lambda}$ and the original training loss $\hat{L}_f$ by the sum of two parts: a parametric component that can be controlled by the dimension $k$ of the "meaningful" part of $x$, and a non-parametric component that can be controlled by a dimension-free complexity measure such as the Euclidean norm of the predictor. Typically, the first term is negligible since $k$ is small, and the complexity of fitting all the noise is absorbed into the second term. More precisely, we introduce the following definitions to formalize separating out a low dimensional component:

**Definition 2.** Under the model assumptions (2), define a (possibly oblique) projection matrix $Q$ onto the space orthogonal to $w_1^*, ..., w_k^*$ and a mapping $\phi$ from $\mathbb{R}^d$ to $\mathbb{R}^{k+1}$ by

$$Q = I_d - \sum_{i=1}^{k} w_i^* (w_i^*)^T \Sigma, \quad \phi(w) = (\langle w, \Sigma w_1^* \rangle, ..., \langle w, \Sigma w_k^* \rangle, \|\Sigma^{1/2} Q w\|_2)^T. \tag{4}$$

We let $\Sigma^\perp = Q^T \Sigma Q$ denote the covariance matrix of $Q^T x$. We also define a low-dimensional *surrogate distribution* $\tilde{\mathcal{D}}$ over $\mathbb{R}^{k+1} \times \mathbb{R}$ by

$$\tilde{x} \sim \mathcal{N}(0, I_{k+1}), \quad \tilde{\xi} \sim \mathcal{D}_\xi, \quad \text{and} \quad \tilde{y} = g(\tilde{x}_1, ..., \tilde{x}_k, \tilde{\xi}). \tag{5}$$

This surrogate distribution compresses the "meaningful part" of $x$ while maintaining the test loss, as shown by our main result Theorem 1 (proved in Appendix D). Note that as a non-asymptotic statement, the functions $\epsilon_{\lambda, \delta}$ and $C_\delta$ only need hold for a specific choice of $n$ and $\mathcal{D}$.

**Theorem 1.** *Suppose $\lambda \in \mathbb{R}^+$ satisfies that for any $\delta \in (0,1)$, there exists a continuous function $\epsilon_{\lambda,\delta} : \mathbb{R}^{k+1} \to \mathbb{R}$ such that with probability at least $1 - \delta/4$ over independent draws $(\tilde{x}_i, \tilde{y}_i)$ from the surrogate distribution $\tilde{\mathcal{D}}$ defined in (5), we have uniformly over all $(\tilde{w}, \tilde{b}) \in \mathbb{R}^{k+2}$ that*

$$\frac{1}{n} \sum_{i=1}^n f_\lambda(\langle \tilde{w}, \tilde{x}_i \rangle + \tilde{b}, \tilde{y}_i) \geq \mathop{\mathbb{E}}_{(\tilde{x}, \tilde{y}) \sim \tilde{D}} [f_\lambda(\langle \tilde{w}, \tilde{x} \rangle + \tilde{b}, \tilde{y})] - \epsilon_{\lambda,\delta}(\tilde{w}, \tilde{b}). \tag{6}$$

*Further, assume that for any $\delta \in (0,1)$, there exists a continuous function $C_\delta : \mathbb{R}^d \to [0, \infty]$ such that with probability at least $1 - \delta/4$ over $x \sim \mathcal{N}(0, \Sigma)$, uniformly over all $w \in \mathbb{R}^d$,*

$$\langle Qw, x \rangle \leq C_\delta(w). \tag{7}$$

*Then it holds with probability at least $1 - \delta$ that uniformly over all $(w, b) \in \mathbb{R}^{d+1}$, we have*

$$L_{f_\lambda}(w, b) \leq \hat{L}_f(w, b) + \epsilon_{\lambda,\delta}(\phi(w), b) + \frac{\lambda C_\delta(w)^2}{n}. \tag{8}$$

*If we additionally assume that (6) holds uniformly for all $\lambda \in \mathbb{R}^+$, then (8) does as well.*

As we will see, we can generally bound the difference between $L_{f_\lambda}$ and $L_f$ when the loss is assumed to be Lipschitz. If $f$ is not Lipschitz but smooth (i.e. $\nabla f$ is Lipschitz, as for the squared loss), we can always write it as the Moreau envelope of another function $\tilde{f}$. In the special case of square loss or squared hinge loss, the Moreau envelope $f_\lambda$ is proportional to $f$, meaning that (8) becomes a generalization guarantee in terms of $L_f$. Optimizing over $\lambda$ will establish optimal bounds that recover the result of Koehler et al. (2021) and Zhou et al. (2021), and lead to other novel results.

**Remark 1.** The complexity functional $C_\delta(w)$ should be thought of as a localized, high-probability version of Rademacher complexity. This is because the Gaussian width of a convex set $\mathcal{K}$, $\mathbb{E} \sup_{w \in \mathcal{K}} \langle w, x \rangle$, is the same as the Rademacher complexity of the class of linear functions $\{x \mapsto \langle w, x \rangle : w \in \mathcal{K}\}$ (Zhou et al. 2021, Proposition 1). A somewhat similar complexity functional appears in Panchenko (2003). Also, note (6) requires only *one-sided concentration* — see Remark 3.

## 4.1 VC Theory for Low-dimensional Concentration

To apply our generalization result (Theorem 1), we should check the low-dimensional concentration assumption (6). The quantitative bounds in the low-dimensional concentration (i.e. the precise form of error term $\epsilon_{\lambda,\delta}$) will inevitably depend on the exact setting we consider (see e.g. Vapnik 1982; Koltchinskii and Mendelson 2015; Lugosi and Mendelson 2019 for discussion).

First, we recall the following result from VC theory.

**Theorem 2** (Special case of Assertion 4 of Vapnik (1982), Chapter 7.8; see also Theorem 7.6)**.** *Let $\mathcal{K} \subset \mathbb{R}^d$ and $\mathcal{B} \subset \mathbb{R}$. Suppose that a distribution $\mathcal{D}$ over $(x, y) \in \mathbb{R}^d \times \mathbb{R}$ satisfies that for some $\tau > 0$, it holds uniformly over all $(w, b) \in \mathcal{K} \times \mathcal{B}$ that*

$$\frac{\left(\mathbb{E} f(\langle w, x \rangle + b, y)^4\right])^{1/4}}{\mathbb{E} f(\langle w, x \rangle + b, y)} \leq \tau. \tag{9}$$

*Also suppose the class of functions $\{(x, y) \mapsto \mathbb{1}\{f(\langle w, x \rangle + b, y) > t\} : w \in \mathcal{K}, b \in \mathcal{B}, t \in \mathbb{R}\}$ has VC-dimension at most $h$. Then for any $n > h$, with probability at least $1 - \delta$ over the choice of $((x_1, y_1), \ldots, (x_n, y_n)) \sim \mathcal{D}^n$, it holds uniformly over all $w \in \mathcal{K}, b \in \mathcal{B}$ that*

$$\frac{1}{n} \sum_{i=1}^n f(\langle w, x_i \rangle + b, y_i) \geq \left(1 - 8\tau \sqrt{\frac{h(\log(2n/h) + 1) + \log(12/\delta)}{n}}\right) \mathbb{E} f(\langle w, x \rangle + b, y).$$

The assumption (9) is standard (indeed, this is the setting primarily focused on in Vapnik 1982) and is sometimes referred to as *hypercontractivity* or *norm equivalence* in the literature; a variant of the result holds with 4 replaced by $1 + \epsilon$. In many settings of interest, this can be directly checked using the fact that $x$ is Gaussian (for instance, see Theorem 9 and Appendix E.3). Of course, our general result can be applied without this assumption, by using low-dimensional concentration under an alternative assumption: Vapnik (1982), Panchenko (2002), Panchenko (2003), and Mendelson (2017) have further discussion and alternative results; in particular, Assertion 3 of Vapnik (1982, Chapter 7.8) gives a bound based on a fourth-moment assumption, and Panchenko (2003, Theorem 3) gives one based on a version of Rademacher complexity.

Combining Theorems 1 and 2 yields the following.

**Corollary 1.** *Under the model assumptions* (2), *suppose that $C_\delta$ satisfies condition* (7). *Also suppose that for some fixed $\lambda \geq 0$, $\mathcal{K} \subseteq \mathbb{R}^d$, and $\mathcal{B} \subseteq \mathbb{R}$, the surrogate distribution $\tilde{\mathcal{D}}$ satisfies assumption* (9) *under $f_\lambda$ uniformly over $\phi(\mathcal{K}) \times \mathcal{B}$, and that the class $\{(x, y) \mapsto \mathbb{1}\{f_\lambda(\langle \tilde{w}, \phi(x)\rangle + \tilde{b}, y) > t\} : \tilde{w} \in \phi(\mathcal{K}), \tilde{b} \in \mathcal{B}, t \in \mathbb{R}\}$ has VC-dimension at most $h$. Then with probability at least $1 - \delta$, uniformly over all $(w, b) \in \mathcal{K} \times \mathcal{B}$*

$$\left(1 - 8\tau\sqrt{\frac{h(\log(2n/h) + 1) + \log(48/\delta)}{n}}\right) L_{f_\lambda}(w, b) \leq \hat{L}_f(w, b) + \frac{\lambda C_\delta(w)^2}{n}.$$

*Furthermore, if assumption* (9) *holds uniformly for all $\{f_\lambda : \lambda \in \mathbb{R}_{\geq 0}\}$ and the class $\{(x, y) \mapsto \mathbb{1}\{f_\lambda(\langle \tilde{w}, \phi(x)\rangle + \tilde{b}, y) > t\} : (\tilde{w}, \tilde{b}) \in \phi(\mathcal{K}) \times \mathcal{B}, t \in \mathbb{R}, \lambda \in \mathbb{R}_{\geq 0}\}$ has VC-dimension at most $h$, then the same conclusion holds uniformly over $\lambda$.*

The last conclusion (uniformity over $\lambda$) follows by going through the proof of Theorem 2, since it is based on reduction to uniform control of indicators. In every situation we will consider, it is easy to check that the VC dimension $h$ in the theorem statement is $O(k)$, generally by reducing to the fact that halfspaces in $\mathbb{R}^k$ have VC dimension $k + 1$.

# 5 Applications

## 5.1 Linear Regression with Square Loss

In this section, we show how to recover optimistic rates (Zhou et al. 2021) for linear regression without assuming the model is well-specified. We will consider the square loss, $f(\hat{y}, y) = (\hat{y} - y)^2$. A key property of the square loss is that the Moreau envelope is proportional to itself:

$$f_\lambda(\hat{y}, y) = \inf_u (u - y)^2 + \lambda(u - \hat{y})^2 = \frac{\lambda}{1 + \lambda} f(\hat{y}, y). \tag{10}$$

Thus we can multiply by $(1 + \lambda)/\lambda$ in our generalization bound and solve for the optimal choice of $\lambda$.

**Corollary 2.** *Suppose $f$ is the square loss and the surrogate distribution $\tilde{\mathcal{D}}$ satisfies assumption* (9) *uniformly over $(w, b) \in \mathbb{R}^{k+1}$, then with probability at least $1 - \delta$, uniformly over all $w, b$ we have*

$$\left(1 - 8\tau\sqrt{\frac{k(\log(2n/k) + 1) + \log(48/\delta)}{n}}\right) L_f(w, b) \leq \left(\sqrt{\hat{L}_f(w, b)} + C_\delta(w)/\sqrt{n}\right)^2.$$

As mentioned earlier, assumption (9) usually holds under mild conditions on $y$. For example, $\tau$ can be chosen to be a constant when $y$ is a bounded-degree polynomial of a Gaussian due to Gaussian hypercontractivity (O'Donnell 2014, Section 11.1). Specializing Corollary 2 to interpolators ($\hat{L}_f = 0$) recovers the uniform convergence of interpolators guarantee from Koehler et al. (2021). Combined with a more general norm analysis in Section 6, we establish $\ell_2$ benign overfitting with misspecification. In the well-specified case, see Zhou et al. (2021) for detailed examples on ordinary least squares, ridge regression, and LASSO.

## 5.2 Classification with Squared Hinge Loss

In this section, we show a novel optimistic rate bound for max-margin linear classification with the squared hinge loss, $f(\hat{y}, y) = \max(0, 1 - y\hat{y})^2$. Its Moreau envelope is given by

$$f_\lambda(\hat{y}, y) = \inf_u \max(0, 1 - yu)^2 + \lambda(u - \hat{y})^2 = \begin{cases} 0 & \text{if } 1 - y\hat{y} \leq 0 \\ \frac{\lambda}{1+\lambda}(1 - y\hat{y})^2 & \text{if } 1 - y\hat{y} > 0 \end{cases} = \frac{\lambda}{1 + \lambda} f(\hat{y}, y).$$

We consider the case of a general binary response $y$ valued in $\{\pm 1\}$ satisfying the model assumptions in equation (2). In this case as well, $f$ is proportional to its Moreau envelope; thus, the same proof as for the squared loss shows that Corollary 2 continues to hold when square loss is replaced by squared hinge loss! In Appendix E.3.1, we discuss certain settings (including noisy settings) where minimizing the squared hinge loss also minimizes the zero-one loss, i.e. the misclassification rate.

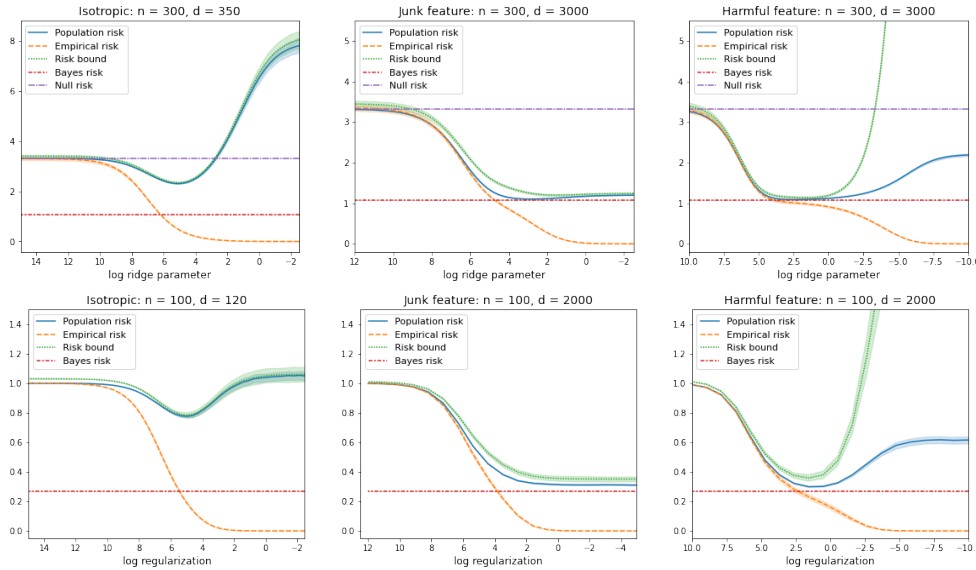

Figure 1: Top: ridge regression with model misspecification; bottom: $\ell_2$ margin classification with a logistic model. Here the features are Gaussian; see Appendix B for additional experiments on $\ell_1$ regularization and non-Gaussian features. The covariance in the first column (isotropic) is $\Sigma = I_d$, in the second column (junk features) is $\Sigma = \mathrm{diag}(1, ..., 1, 0.05^2, ..., 0.05^2)$, and in the third (harmful features) is $\Sigma = \mathrm{diag}(1, ..., 1, \frac{1}{(k+1)^2}, ..., \frac{1}{d^2})$. For regression, the number of leading eigenvalues is $k = 3$, and the label is generated according to $y = 1.5\,x_1 + |x_1|\cos x_2 + x_3 \cdot \mathcal{N}(0, 0.5)$. For classification, the number of leading eigenvalues is $k = 1$ and $\Pr(y = 1 \mid x) = \mathrm{sigmoid}(5x_1 + 3)$. The risk bound is calculated using the expression $\left(\sqrt{\hat{L}} + \sqrt{\|w\|_2^2 \mathrm{Tr}(\Sigma^\perp)/n}\right)^2$, which corresponds to the choice of $C(w)$ from Lemma 1, and is expected to be tight in the junk features setting. In the isotropic case, we use an easy improvement of the bound where $w$ is projected to the orthogonal complement of the Bayes predictor $w^*$ (Zhou et al. 2021). In the other cases, we use covariance splitting without projection of $w$. Each point on the curve is the average from repeated experiments, and shaded areas correspond to 95% bootstrap confidence intervals.

## 5.3 Further applications

In this section, we discuss further interesting examples where our general theory applies. As before, we can obtain a generalization bound by appealing to the general Corollary 1: we omit stating the formal corollary for each case, and simply show the Moreau envelope and its consequences.

$L_1$ **loss (LAD) and Hinge.** For the $L_1$ loss $f(\hat{y}, y) = |\hat{y} - y|$ its Moreau envelope is given by

$$f_\lambda(\hat{y}, y) = \inf_u |u - y| + \lambda(u - \hat{y})^2 = \begin{cases} \lambda(\hat{y} - y)^2 & \text{if } |\hat{y} - y| \le \frac{1}{2\lambda} \\ |\hat{y} - y| - \frac{1}{4\lambda} & \text{if } |\hat{y} - y| > \frac{1}{2\lambda} \end{cases},$$

which is $2\lambda$ times the Huber loss with parameter $\delta = \frac{1}{2\lambda}$. Therefore, the population Huber loss is controlled by the empirical $L_1$ loss. Clearly, interpolators have zero training error under both $L_1$ and $L_2$ training losses. We already see that Corollary 2 implies $(1 - o(1))\,\mathbb{E}(\langle w, x\rangle + b - y)^2 \le C_\delta(w)^2/n$. Now, considering Corollary 1 with $f$ the $L_1$ loss and using the above Moreau envelope calculation, we can check that when $\lambda \to 0$ we reproduce the exact same bound, since the Huber loss becomes the squared loss in the limit. Further insight into this phenomenon appears later in Theorem 4: the Huber loss naturally shows up when computing the training error of the LAD estimator. An entirely analogous situation occurs when $f$ is the hinge loss $f(\hat{y}, y) := \max(0, 1 - \hat{y}y)$: its Moreau envelope $f_\lambda$ will be a rescaling of the Huber hinge loss (c.f. T. Zhang 2004a).

**Lipschitz loss: improved contraction.** If $f$ is $M$-Lipschitz in $\hat{y}$, Proposition 3.4 of Planiden and X. Wang (2019) gives that $0 \le f - f_\lambda \le \frac{M^2}{4\lambda}$. Thus, assuming $k/n = o(1)$, Corollary 1 implies

$(1 - o(1))(L_f(w) - \frac{M^2}{4\lambda}) \leq \hat{L}_f(w) + \frac{\lambda C_\delta(w)^2}{n}$; optimizing over $\lambda$ yields

$$(1 - o(1))L_f(w) \leq \hat{L}_f(w) + M\sqrt{\frac{C_\delta(w)^2}{n}}. \tag{11}$$

For $w \in \mathcal{K}$, a high-probability upper bound on the Rademacher complexity of the function class $x \mapsto \langle w, x \rangle$ upper bounds the second term (Remark 1). In comparison, the standard symmetrization and contraction argument (Bartlett and Mendelson 2002) loses a factor of two. Note that if $f$ is the $L_1$ loss, this applies with $M = 1$, and it can further be shown that the constant factor cannot be improved further (see Appendix E.4), but this bound is also not as tight as the version with Huber test loss.

**Uniform Convergence of Interpolators for Smooth Losses.** Suppose that $f$ is an $H$-smooth (in the sense that $|\frac{\partial^2 f}{\partial \hat{y}^2}| \leq H$) and convex function of $\hat{y}$. In addition, assume that the minimum of $f(\hat{y}, y)$ is zero for any fixed $y$. Since $f$ is $H$-smooth, there exists a function $\tilde{f}$ such that $f = \tilde{f}_{H/2}$ (Bauschke, Combettes, et al. 2011, Corollary 18.18). If $w$ satisfies $\hat{L}_f(w) = 0$, then $\hat{L}_{\tilde{f}}(w) = 0$ as well.[1] Finally, by Corollary 2, if $k/n = o(1)$ we have uniformly over all $w$ such that $\hat{L}_f(w) = 0$ that

$$(1 - o(1))L_f(w) \leq \frac{H}{2} \cdot \frac{C_\delta(w)^2}{n}, \tag{12}$$

which generalizes the main result of Koehler et al. (2021) to arbitrary smooth losses.

# 6   $\ell_2$ Benign Overfitting

We can obtain conditions for consistency, like those of Bartlett et al. (2020), by simply combining a norm-based uniform generalization bound with an upper bound on the norm of interpolators. Koehler et al. (2021) used the same strategy, but our more powerful and general tools give better results:

1. For squared loss regression, we show that the assumption that the ground truth is generated by a linear model (i.e. well-specified), made in previous work, is not required. The same result holds under the much more general model[2] assumption of Section 3.

2. We show an analogous result in the *classification* setting, replacing the squared loss with the squared hinge loss. In fact, the argument is exactly the same in the two cases: all we need to use is that $f_\lambda = \frac{\lambda}{1+\lambda} f$ and that $f$ is the square of a Lipschitz function.

First, the following lemma (essentially just the standard Rademacher complexity bound for the $\ell_2$ ball) combines with Corollary 2 and its squared hinge loss version to give generalization bounds. These bounds are demonstrated in Figure 1 and Appendix B.

**Lemma 1.** *In the setting of Theorem 1, letting $\Sigma^\perp = Q^T \Sigma Q$, the following $C_\delta(w)$ will satisfy (7):*
$C_\delta(w) = \|w\|_2 \left[ \sqrt{\text{Tr}(\Sigma^\perp)} + 2\sqrt{\|\Sigma^\perp\|_{op} \log(8/\delta)} \right].$

Next, we provide a sufficient condition for a zero-training error predictor $w$ to exist in an $\ell_2$ ball. In the case of classification, this allows us to lower-bound the margin of the max-margin halfspace.

**Definition 3** (Bartlett et al. 2020). *The effective ranks of a covariance matrix $\Sigma$ are*

$$r(\Sigma) = \frac{\text{Tr}(\Sigma)}{\|\Sigma\|_{op}} \quad \text{and} \quad R(\Sigma) = \frac{\text{Tr}(\Sigma)^2}{\text{Tr}(\Sigma^2)}.$$

---

[1]Since $f \leq \tilde{f}$ is nonnegative, $\tilde{f}$ is nonnegative as well. When $f(\hat{y}, y) = 0$, there must be $u_\varepsilon$ such that $\tilde{f}(u_\varepsilon, y) + \lambda(u_\varepsilon - \hat{y})^2 < \varepsilon$; this is only possible if we have $u_\varepsilon \to \hat{y}$, implying since $f$ is smooth that $\tilde{f}(\hat{y}, y) = 0$. If $\hat{L}_f(w) = 0$, we must have for every $i$ that $f(\hat{y}_i, y_i) = 0$; thus $\tilde{f}(\hat{y}_i, y_i) = 0$ and $L_{\tilde{f}}(w) = 0$.

[2]Theorem 3 of concurrent work by Shamir (2022) also proves benign overfitting for some misspecified models, but requires the very strong assumption $n^2 \|\Sigma^\perp\|_{op} \to 0$ that fails to hold in several examples from Bartlett et al. (2020).

**Lemma 2.** *Suppose that $f(\hat{y}, y)$ is either squared loss or squared hinge loss. Let $(w^\sharp, b^\sharp) \in \mathbb{R}^{d+1}$ be an arbitrary vector satisfying $Qw^\sharp = 0$ and with probability at least $1 - \delta/4$,*

$$\hat{L}_f(w^\sharp, b^\sharp) \le L_f(w^\sharp, b^\sharp) + \rho_1(w^\sharp, b^\sharp) \tag{13}$$

*for some $\rho_1(w^\sharp, b^\sharp) > 0$. Then for any $\rho_2 \in (0, 1)$, provided $\Sigma^\perp = Q^T \Sigma Q$ satisfies*

$$R(\Sigma^\perp) = \Omega\left(\frac{n \log^2(4/\delta)}{\rho_2}\right), \tag{14}$$

*we have that with probability at least $1 - \delta$ that $\min_{\|w\| \le B} L_f(w, b^\sharp) = 0$ for $B > 0$ defined by $B^2 = \|w^\sharp\|_2^2 + (1 + \rho_2)\frac{n}{\operatorname{Tr}(\Sigma^\perp)}(L_f(w^\sharp, b^\sharp) + \rho_1)$.*

We note that for any vector $w^\sharp$, we have $L_f((I - Q)w^\sharp, b^\sharp) < L_f(w^\sharp, b^\sharp)$ by Jensen's inequality over $Q^T x$, so the assumption $Qw^\sharp = 0$ in the lemma is always satisfied for the minimizer of $L_f(w, b)$. Combining the norm bound Lemma 2 and the generalization bound Lemma 1 yields the following.

**Theorem 3.** *Let $(\hat{w}, \hat{b}) = \arg\min_{w \in \mathbb{R}^d, b \in \mathbb{R} : \hat{L}_f(w,b)=0} \|\hat{w}\|_2$ be the minimum-$\ell_2$ norm predictor with zero training error. In the setting of Lemma 2, we have*

$$L_f(\hat{w}, \hat{b}) - \epsilon_\delta(\phi(\hat{w}), \hat{b}) \le (1 + \rho_3) \inf_{w^\sharp \in \mathbb{R}^d, b^\sharp \in \mathcal{B}} \left(L_f(w^\sharp, b^\sharp) + \rho_1(w^\sharp, b^\sharp) + \frac{\|w^\sharp\|_2^2 \operatorname{Tr}(\Sigma^\perp)}{n}\right),$$

*where $\rho_3 > 0$ is defined by $1 + \rho_3 = (1 + \rho_2)\left[1 + 2\sqrt{\frac{\log(2/\delta)}{r(\Sigma^\perp)}}\right]^2$ and we recall $\rho_1(w^\sharp, b^\sharp)$ from (13).*

We now show that this formally implies convergence to the optimal test loss (i.e. consistency) under the $\ell_2$ benign overfitting conditions (15) from Bartlett et al. (2020) and Tsigler and Bartlett (2020):

**Corollary 3.** *Suppose that $\mathcal{D}_n$ is a sequence of data distributions following our model assumptions (2), with $k_n$ such that $y = g(\eta_1, \ldots, \eta_{k_n}, \xi)$, and projection operator $Q_n$ defined as in (4). Suppose $f$ is either the squared loss or the squared hinge loss, and define $(w_n^\sharp, b_n^\sharp) = \arg\min_{w,b} L_{f,n}(w, b)$ where $L_{f,n}(w, b)$ is the population loss over distribution $\mathcal{D}_n$ with loss $f$. Suppose that the hypercontractivity assumption (9) holds with some fixed $\tau > 0$ for all $\mathcal{D}_n$. Define $\Sigma_n := \mathbb{E}_{\mathcal{D}_n}[xx^T]$ and $\Sigma_n^\perp = Q_n^T \Sigma_n Q_n$. Suppose that as $n \to \infty$, we have*

$$\frac{n}{R(\Sigma_n^\perp)} \to 0, \quad \frac{\|w_n^\sharp\|_2^2 \operatorname{Tr}(\Sigma_n^\perp)}{n} \to 0, \quad \frac{k_n}{n} \to 0. \tag{15}$$

*Then we have the following convergence in probability, as $n \to \infty$:*

$$\frac{L_{f,n}(\hat{w}_n, \hat{b}_n)}{L_{f,n}(w_n^\sharp, b_n^\sharp)} \to 1, \tag{16}$$

*where $(\hat{w}_n, \hat{b}_n) = \arg\min_{w \in \mathbb{R}^d, b \in \mathbb{R} : \hat{L}_f(w,b)=0} \|w\|_2$ is the minimum-norm interpolator, and $\hat{L}_{f,n}$ is the training error based on $n$ i.i.d. samples from the distribution $\mathcal{D}_n$.*

Note when applying Corollary 3, we have the flexibility to increase $k_n$ and shrink $\Sigma_n^\perp$ by choosing additional weights $w_i^*$ and letting the link function $g$ ignore the extra components.

**Remark 2** (Flatness of the test loss along regularization path)**.** Our method can easily show a slightly stronger statement: let $(\hat{w}_n, \hat{b}_n) \in \arg\min_{\|w\| \le B_n, b \in \mathbb{R}} \hat{L}_{f,n}(w_n^\sharp, b_n^\sharp)$ such that, if there are multiple minima, we pick the one with smallest $\|w\|$. As long as $B_n \ge \|w_n^\sharp\|$, we still have (16), and this is established uniformly over all sequences $B_n$ satisfying the constraint. Therefore, under the benign overfitting conditions we get consistency as long as we do not over-regularize the predictor. See Figure 1 for an experimental demonstration of the flatness.

## 7 Training Error and Local Gaussian Width

Theorem 1 shows how to upper-bound the test error of a predictor (under the Moreau envelope loss) by its training error and an upper bound on the class complexity. The following theorem is the dual result, which upper-bounds the training error of the constrained ERM (Empirical Risk Minimizer) by the Moreau envelope and a complexity term. In particular, this general result is used to derive the norm bound for interpolators in Lemma 2 above.

**Theorem 4.** *Let $\mathcal{K}, \mathcal{B}$ be bounded convex sets, and let $f(\hat{y}, y)$ be convex in $\hat{y}$. Suppose that $\tau$ is such that with probability at least $1 - \delta$, for $(\tilde{x}, \tilde{y})_{i=1}^n$ sampled i.i.d. from $\tilde{\mathcal{D}}$ we have*

$$\min_{\tilde{w} \in \phi(\mathcal{K}), b_0 \in \mathcal{B}} \max_{\lambda \geq 0} \left[ \frac{1}{n} \sum_{i=1}^n f_\lambda(\langle \tilde{w}, \tilde{x} \rangle + b_0, y_i) - \frac{\lambda}{n} \max_{w_0 \in \phi^{-1}(\tilde{w}) \cap \mathcal{K}} \langle x, Q w_0 \rangle^2 \right] \leq \tau. \quad (17)$$

*Then with probability at least $1 - 2\delta$, $\min_{w \in \mathcal{K}, b \in \mathcal{B}} \hat{L}_f(w, b) \leq \tau$.*

Note that the assumption (17) implicitly suggests a low-dimensional concentration assumption: we expect $\frac{1}{n} \sum_{i=1}^n f_\lambda(\langle \tilde{w}, \tilde{x} \rangle + b_0, y_i)$ to be approximately the test loss of $(\tilde{w}, b_0)$ under the surrogate distribution $\tilde{\mathcal{D}}$. As we discuss more in Appendix F, combining this training error bound with the correct choice of $C_\delta(w)$ in Theorem 1, which is essentially $C_\delta(w) = \mathbb{E} \max_{w_0 \in \phi^{-1}(\tilde{w}) \cap \mathcal{K}} \langle x, Q w_0 \rangle^2$), yields a matching lower bound to (8) on the Moreau envelope test loss and so our generalization bound is asymptotically sharp. This establishes a non-asymptotic analogue of the existing asymptotic Moreau envelope theory (see Section 2), and recovers the special case of well-specified linear models (Zhou et al. 2021).

# 8 Discussion

In this work, we significantly extend the localized uniform convergence technique developed in the study of noisy interpolation to any loss function and label generating process under mild conditions. Though the application of Moreau envelope to study GLMs is not new in the statistical literature, our general theory establishes novel non-asymptotic generalization bounds in a wide variety of overparameterized settings. We believe the generality of our framework may allow further applications in other areas of statistics, such as robust statistics and high-dimensional inference.

As mentioned in Section 2, the applicability of our theory is still considerably limited by the Gaussian data assumption, required by our use of the Gaussian minimax theorem. It does appear experimentally that it may hold much more broadly (Appendix B); proving that this is the case could allow us to study kernel methods and bring us closer to a theoretical understanding of deep neural networks. Some work has been done in related settings to extend Gaussian-based results to broader distributions via universality arguments (e.g. Hu and Lu 2022; Liang and Sur 2020; Montanari and Saeed 2022), but it is not yet clear how to apply those techniques to our general framework. The GMT formulation also does not allow for multi-class classification or two-layer networks, because of their vector-valued outputs. Overcoming these two challenges seems to be crucial avenues for future work.

## Acknowledgments and Disclosure of Funding

F.K. was supported in part by NSF award CCF-1704417, NSF award IIS-1908774, and N. Anari's Sloan Research Fellowship. P.S. was supported in part by NSF award DMS-2113426. D.J.S. was supported in part by the Canada CIFAR AI Chairs program. Part of this work was initiated when F.K., P.S., and N.S. were visiting the Simons Institute for the Theory of Computing for their program on Computational Complexity of Statistical Inference. This work was done as part of the Collaboration on the Theoretical Foundations of Deep Learning (`deepfoundations.ai`).

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
