## A  Organization of the Appendices

In this appendix, we provide additional simulation results and complete proofs of all the results in the main text. In Appendix B, we provide additional simulation results. In Appendix C, we introduce standard notation and tools which we use throughout the remainder of the appendices. In Appendix D, we give a proof of our main result Theorem 1. In Appendix E, we apply VC theory to handle low-dimensional concentration and prove the generalization guarantees for linear regression and classification. In Appendix F, we prove Theorem 4. In Appendix G, we establish a norm bound for interpolators and apply our generalization bound of Section 5 to show consistency.

## B  Additional Numerical Simulations

This section presents additional numerical simulations on synthetic data to confirm our theory and test it beyond the case of Gaussian covariates. All code is available from https://github.com/zhoulijia/moreau-envelope.[3]

### B.1  Linear Regression

We fit linear models to minimize the square loss with $\ell_1$ and $\ell_2$ penalty. For simplicity, we ignore the intercept term in this section, but we will consider models with intercept in the context of linear classification. We can obtain many data distributions by combining the different options below:

**Feature Distribution.** The marginal distribution of $x$ is always given by $x = \Sigma^{1/2} z$, where $z$ is a random vector with i.i.d. coordinates that have mean 0 and variance 1.

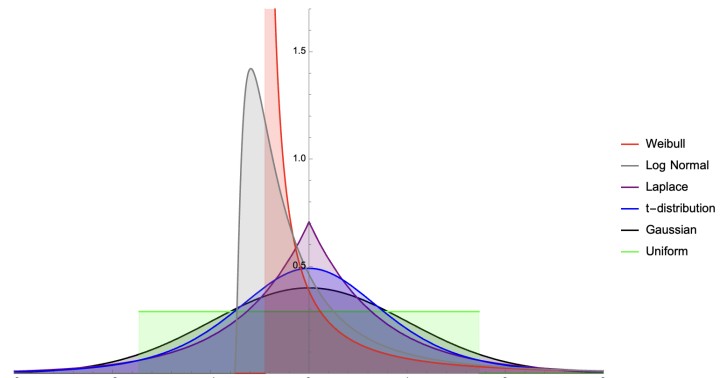

Figure 2: Probability density plot for the continuous distributions of $z$ that we consider.

The coordinate distributions of $z$ that we consider in the simulations include

- Gaussian
  - the standard Gaussian distribution has density $p(z) = \frac{1}{\sqrt{2\pi}} e^{-\frac{1}{2}z^2}$
- Uniform
  - the uniform distribution between 0 and 1 has mean 0 and variance $\frac{1}{12}$. After normalization, it becomes the uniform distribution between $-\sqrt{3}$ and $\sqrt{3}$. It's symmetric, bounded from above and below, and therefore sub-Gaussian
- Laplace
  - Laplace distribution with scale parameter $b$ has density $p(z) = \frac{1}{2b} e^{-\frac{|z|}{b}}$ and variance $2b^2$, so we should choose $b = \frac{1}{\sqrt{2}}$

---

[3]The ridge path is computed using SVD implemented by `np.linalg.svd`. The LASSO path is computed using coordinate descent implemented by `sklearn.linear_model.lasso_path`, and $\ell_1$ and $\ell_2$ margin classifiers are fitted using `sklearn.svm.LinearSVC` with the default squared hinge loss option.

– it is symmetric, unbounded, and has fatter tails compared to Gaussian (sub-exponential)

We also consider discrete distributions

- Rademacher
  - the discrete distribution with equal chance of being $-1$ or $1$. It is easy to see that it has mean $0$ and unit variance.
- Poisson
  - Poisson distribution with rate parameter $1$ is supported on the non-negative integers (skewed and bounded from below) and has density $\Pr(\tilde{z} = k) = \frac{e^{-1}}{k!}$. Its mean and variance are both equal to $1$, and so we take $z = \tilde{z} - 1$ to normalize

and heavy-tailed distributions

- Student's t-distribution
  - t-distribution with 5 degrees of freedom has density $p(\tilde{z}) = \frac{8}{3\sqrt{5}\pi\left(1 + \frac{\tilde{z}^2}{5}\right)^3}$
  - It has variance $\frac{5}{3}$ and so we let $z = \sqrt{\frac{3}{5}}\tilde{z}$. It is symmetric, unbounded and has finite fourth moment. However, moments of order 5 or higher do not exist.
- Weibull
  - Weibull distribution with scale parameter $\lambda = 1$ and shape parameter $k = 0.5$ has density $p(\tilde{z}) = \frac{e^{-\sqrt{\tilde{z}}}}{2\sqrt{\tilde{z}}}\mathbb{1}_{\{\tilde{z} \geq 0\}}$. It has mean $2$ and variance $20$ and so we take $\tilde{z} = \frac{z-2}{\sqrt{20}}$
- Log-Normal
  - the distribution of $e^Z$, where $Z$ follows the standard Gaussian distribution. It has mean $\sqrt{e}$ and variance $e(e-1)$, and so we can choose $z = \frac{e^Z - \sqrt{e}}{\sqrt{e(e-1)}}$

**Covariance Matrix and Scaling.** For simplicity, we choose $\Sigma$ to be diagonal and consider

- Isotropic features $\Sigma = I_d$ in the proportional scaling ($n = 300, d = 350$)
- Junk features in the over-parameterized scaling ($n = 300, d = 3000$)

$$\Sigma_{kk} = \begin{cases} 1 & \text{if } k = 1, 2, 3 \\ 0.05^2 & \text{otherwise} \end{cases}$$

- Non-benign features in the over-parameterized scaling ($n = 300, d = 3000$)

$$\Sigma_{kk} = \begin{cases} 1 & \text{if } k = 1, 2, 3 \\ \frac{1}{k^2} & \text{otherwise} \end{cases}$$

The junk features setting is known to satisfy the benign overfitting conditions (Zhou et al. 2020; Bartlett et al. 2020), by which the minimal $\ell_2$-norm interpolator is consistent. In contrast, Bartlett et al. (2020) also shows that overfitting is not benign in the second case, but the theory from Zhou et al. (2021) shows that the optimally-tuned ridge regression can be consistent.

**Conditional Distribution of $y$.** Let

$$w^* = (1.5, 0, ..., 0)$$
$$\xi \sim \mathcal{N}(0, 0.5)$$

and consider

- a well-specified linear model:

$$y = \langle w^*, x \rangle + \xi$$

- a mis-specified model:

$$y = \underbrace{\langle w^*, x \rangle}_{\text{linear signal}} + \underbrace{|x_1| \cdot \cos x_2}_{\text{non-linear term}} + \underbrace{x_3 \cdot \xi}_{\text{heteroscedasticity}}$$

The second model does not satisfy the classical assumptions for linear regression because the Bayes predictor

$$\mathbb{E}[y|x] = \langle w^*, x \rangle + |x_1| \cdot \cos x_2$$

is non-linear and the variance of the residual also depends on $x_4$. Even though statistical inference can be challenging for models like this, we can hope to learn a model that competes with the optimal linear predictor (which is not necessarily the same as $w^*$) in terms of prediction error.

### B.1.1 Speculative Risk Bounds for Non-Gaussian Features

Though our theory is restricted to Gaussian features, we conjecture that it can be extended to a more general class of distributions using Rademacher complexity and we use numerical simulations to confirm our conjecture.

**Ridge Regression**

1. Isotropic features: similar to Lemma 10 in Zhou et al. (2021), we can choose $C_\delta$ in corollary 2 by the simple Cauchy-Schwarz bound

$$\langle Qw, x \rangle \leq \|Qw\|_2 \cdot \|x\|_2 \approx \sqrt{d}\|Qw\|_2$$

resulting in the following bound

$$L_f(w) \leq (1 + o(1)) \left( \sqrt{\hat{L}_f(w)} + \sqrt{\frac{d}{n}} \cdot \|Qw\|_2 \right)^2 \tag{18}$$

2. Junk and non-benign features: choosing $C_\delta$ in corollary 2 according to Lemma 1 yields

$$L_f(w) \leq (1 + o(1)) \left( \sqrt{\hat{L}_f(w)} + \|w\|_2 \sqrt{\frac{\text{Tr}(\Sigma^\perp)}{n}} \right)^2 \tag{19}$$

In all of the experiments, we use a constant close to 1 to replace the $1 + o(1)$ factor in our generalization bounds. Note that (19) can be interpreted in terms of Rademacher complexity:

$$\mathop{\mathbb{E}}_{\substack{x_1,\ldots,x_n \sim \mathcal{D} \\ s \sim \text{Unif}(\{\pm 1\}^n)}} \left[ \sup_{\|w\|_2 \leq B} \left| \frac{1}{n} \sum_{i=1}^n s_i \langle w, Q^T x_i \rangle \right| \right] = \frac{B}{n} \cdot \mathop{\mathbb{E}}_{\substack{x_1,\ldots,x_n \sim \mathcal{D} \\ s \sim \text{Unif}(\{\pm 1\}^n)}} \left[ \left\| \sum_{i=1}^n s_i Q^T x_i \right\|_2 \right]$$

$$\leq B \cdot \sqrt{\frac{\text{Tr}(\Sigma^\perp)}{n}}$$

The last inequality holds generally for any distribution with $\mathbb{E}_{x \sim \mathcal{D}}[xx^T] = \Sigma$ by Cauchy-Schwarz inequality. In our examples, $x = \Sigma^{1/2}z$ and $z$ is scaled to satisfy $\mathbb{E}[zz^T] = I_d$. Therefore, we will use equation (18) and (19) even for non-Gaussian data.

Equations (18) and (19) are qualitatively similar with subtle technical differences. Compared with equation (19), the bound (18) uses the smaller norm $\|Qw\|_2$ and figure 2 of Koehler et al. (2021) demonstrates that this projection is crucial for tight bounds in the isotropic setting. On the other hand, equation (19) incorporates the covariance splitting technique (Bartlett et al. 2020) because large eigenvalues of $\Sigma$ can be killed off in $\Sigma^\perp$ by projection $Q$ while $\text{Tr}(I_d) = d$ in the isotropic case. It is shown in our corollary 3 that this bound without the projection is already tight enough to establish the consistency of minimal-$\ell_2$ norm interpolator in the junk feature setting. Hence, we expect (19) to be tight throughout the ridge path. In contrast, the theory in Zhou et al. (2021) predicts that (19) is tight for the non-benign setting only up to the point where the ridge estimate has norm as large as the optimal linear predictor. We believe using the local Gaussian width theory introduced in Section 7 (i.e. an optimal choice of $C_\delta(w)$) can get tight bound throughout the ridge path in this setting, but we do not have experiments in this appendix to confirm it.

In the theoretical analysis of Zhou et al. (2021), they further write $\|Qw\|_2$ as a function of $\|w\|_2$, $\|w^*\|_2$ and the excess risk $\|w - w^*\|_\Sigma^2$ in the isotropic case, then solve the equation in terms of $\|w - w^*\|_\Sigma^2$ to get a norm-based generalization bound as a function of $\|w\|_2$ when $\hat{L}_f(w) = 0$ (see their theorem 6). Since the solution for general non-zero $\hat{L}_f(w)$ can have a quite tedious expression, for the purpose of numerically checking the applicability and tightness of this approach, we will use simpler equation (18) in the experiments.

**LASSO Regression** Similar to the section above, we use the analogy to Rademacher complexity to extend our theory to the $\ell_1$ case. Since we can no longer bound the $\ell_\infty$ norm of a sum using the Cauchy-Schwarz inequality, it is easier to directly work with the empirical Rademacher complexity (which also should be similar to the expected Rademacher complexity in the settings that we consider)

$$\frac{\|w\|_1}{n} \cdot \mathop{\mathbb{E}}_{s \sim \text{Unif}(\{\pm 1\}^n)} \left[ \left\| \sum_{i=1}^n s_i Q^T x_i \right\|_\infty \right]$$

and we can estimate the expected norm by

$$\frac{1}{B} \sum_{k=1}^B \left\| \sum_{i=1}^n s_{k,i} Q^T x_i \right\|_\infty$$

for a large value of $B$ and $s_1, ..., s_B$ sampled independently from $\text{Unif}(\{\pm 1\}^n)$. In our implementation, $s_1, ..., s_B$ are fresh samples each time the risk bound is computed. To summarize, we use the following expression for the calculation of risk bound:

1. Isotropic features:

$$\left( \sqrt{\hat{L}_f(w)} + \|Qw\|_1 \cdot \frac{1}{nB} \sum_{k=1}^B \left\| \sum_{i=1}^n s_{k,i} x_i \right\|_\infty \right)^2 \tag{20}$$

2. Junk and non-benign features:

$$\left( \sqrt{\hat{L}_f(w)} + \|w\|_1 \cdot \frac{1}{nB} \sum_{k=1}^B \left\| \sum_{i=1}^n s_{k,i} Q^T x_i \right\|_\infty \right)^2 \tag{21}$$

which are analogous to (18) and (19).

We note that it is important to use the Rademacher complexity to extend to non-Gaussian features in the $\ell_1$ case, rather than a bound similar to $\frac{\|w\|_1 \mathbb{E} \|x\|_\infty}{\sqrt{n}}$. Empirically, the latter is too small to provide a valid upper bound on the test loss. This is because $\|x\|_\infty$ is deterministic for distributions like the Rademacher distribution, while the random signs in the definition of Rademacher complexity allows a tail behavior more similar to Gaussian and so we can regain a log factor in the norm component.

### B.1.2 Experimental Results

For both ridge and LASSO regression, risk curves measured in the square loss are shown in three figures corresponding to the different data covariances. Within each figure, there are 16 subplots corresponding to the different combinations of one of the eight feature distributions and label generating process (well-specified vs mis-specified) as defined at the beginning of the section. Therefore, there are 96 subplots in total. Discussion of the experimental outcome can be found in the caption of each figure.

Similar to the situation in the rest of the experiments, the training error is close to 0 with sufficiently small regularization, and the confidence bands are wider with heavy-tailed distributions. Also, the null risk and the Bayes risk are different across different feature distributions when there is model misspecification (see the calculation in the next subsection for more details).

**Ridge Regression.** The plots for isotropic, junk and non-benign features in the ridge regression setting can be found in figures 3, 4 and 5, respectively. Generally speaking, the experiments confirm the tightness and wide applicability of our generalization guarantees. The specific feature distribution and model misspecification do not seem to affect the shape of test error curve.

**LASSO Regression.** The plots for isotropic, junk, and non-benign features in the LASSO regression setting can be found in Figures 6 to 8. The risk bounds in the $\ell_1$ case are not as tight as in the $\ell_2$ case because they are only expected to be tight in certain parts of the entire regularization path. As mentioned earlier, we can get sharp bounds for the entire path using local Gaussian width, but it requires a more fine-grained analysis than (20) and (21). Similar results and experiments were obtained by G. Wang et al. (2021) and Donhauser et al. (2022).

### B.1.3 Note on Computing the Optimal Linear Predictor and Population Risk

Since we are considering quite high-dimensional settings and we need many repeated experiments for different regularization strengths, we generally want to avoid drawing a large test set to estimate the prediction error when it is possible. In the case of square loss, we can always write the population loss (using the Mahalanobis norm notation (22)) as

$$L_f(w) = L_f(\tilde{w}) + \|w - \tilde{w}\|_\Sigma^2$$

where $\tilde{w}$ is the optimal linear predictor satisfying the first order condition:

$$\mathbb{E}[x(x^T\tilde{w} - y)] = 0.$$

**Linear Model.** In the well-specified case, by the independence between $x$ and $\xi$, the above becomes

$$\Sigma\tilde{w} = \Sigma w^* \implies \tilde{w} = w^*.$$

Therefore, we have $L_f(\tilde{w}) = \mathbb{E}[(y - \langle w^*, x \rangle)^2] = \sigma^2$.

**Mis-specified Model.** To determine the optimal linear predictor in this case, we want to set

$$\begin{aligned}
\Sigma\tilde{w} &= \mathbb{E}[xy] \\
&= \mathbb{E}[x(\langle w^*, x \rangle + |x_1| \cdot \cos x_2)] \\
&= \Sigma w^* + \mathbb{E}[x_1 \cdot |x_1|]\,\mathbb{E}[\cos x_2]e_1 + \mathbb{E}[|x_1|]\,\mathbb{E}[x_2 \cos x_2]e_2
\end{aligned}$$

and so

$$\tilde{w} = w^* + \mathbb{E}[x_1 \cdot |x_1|]\,\mathbb{E}[\cos x_2]\Sigma^{-1}e_1 + \mathbb{E}[|x_1|]\,\mathbb{E}[x_2 \cos x_2]\Sigma^{-1}e_2.$$

At the same time, it is routine to check that the optimal error is given by

$$L_f(\tilde{w}) = \mathbb{E}[y^2] - \langle \mathbb{E}[xy], \Sigma^{-1}\,\mathbb{E}[xy] \rangle.$$

It remains to compute the null risk

$$\begin{aligned}
\mathbb{E}[y^2] &= \mathbb{E}[(\langle w^*, x \rangle + |x_1| \cdot \cos x_2 + x_3\xi)^2] \\
&= \mathbb{E}[(\langle w^*, x \rangle + |x_1| \cdot \cos x_2)^2] + \Sigma_{33}\sigma^2 \\
&= \langle w^*, \Sigma w^* \rangle + \mathbb{E}[x_1^2]\,\mathbb{E}[\cos^2 x_2] + 2\,\mathbb{E}[\langle w^*, x \rangle(|x_1| \cdot \cos x_2)] + \Sigma_{33}\sigma^2 \\
&= \langle w^*, \Sigma w^* \rangle + \mathbb{E}[x_1^2]\,\mathbb{E}[\cos^2 x_2] + 2\left(\mathbb{E}[x_1 \cdot |x_1|]\,\mathbb{E}[\cos x_2]w_1^* + \mathbb{E}[|x_1|]\,\mathbb{E}[x_2 \cos x_2]w_2^*\right) + \Sigma_{33}\sigma^2
\end{aligned}$$

and

$$\begin{aligned}
\langle \mathbb{E}[xy], \Sigma^{-1}\,\mathbb{E}[xy] \rangle &= \langle \Sigma w^* + \mathbb{E}[|x_1|\cos(x_2)x], w^* + \Sigma^{-1}\,\mathbb{E}[|x_1|\cos(x_2)x] \rangle \\
&= \langle \Sigma w^*, w^* \rangle + 2\langle w^*, \mathbb{E}[|x_1|\cos(x_2)x] \rangle + \langle \mathbb{E}[|x_1|\cos(x_2)x], \Sigma^{-1}\,\mathbb{E}[|x_1|\cos(x_2)x] \rangle.
\end{aligned}$$

Therefore, we have

$$L_f(\tilde{w}) = \mathbb{E}[x_1^2]\,\mathbb{E}[\cos^2 x_2] + \Sigma_{33}\sigma^2 - \mathbb{E}[x_1 \cdot |x_1|]^2\,\mathbb{E}[\cos x_2]^2\Sigma_{11}^{-1} - \mathbb{E}[|x_1|]^2\,\mathbb{E}[x_2 \cos(x_2)]^2\Sigma_{22}^{-1}$$

It remains to compute quantities like $\mathbb{E}[|x|], \mathbb{E}[x \cdot |x|], \mathbb{E}[\cos x], \mathbb{E}[x \cos x]$ for each of the eight feature distributions. Since they are one dimensional quantities, we can afford to draw a very large number of samples to estimate them.

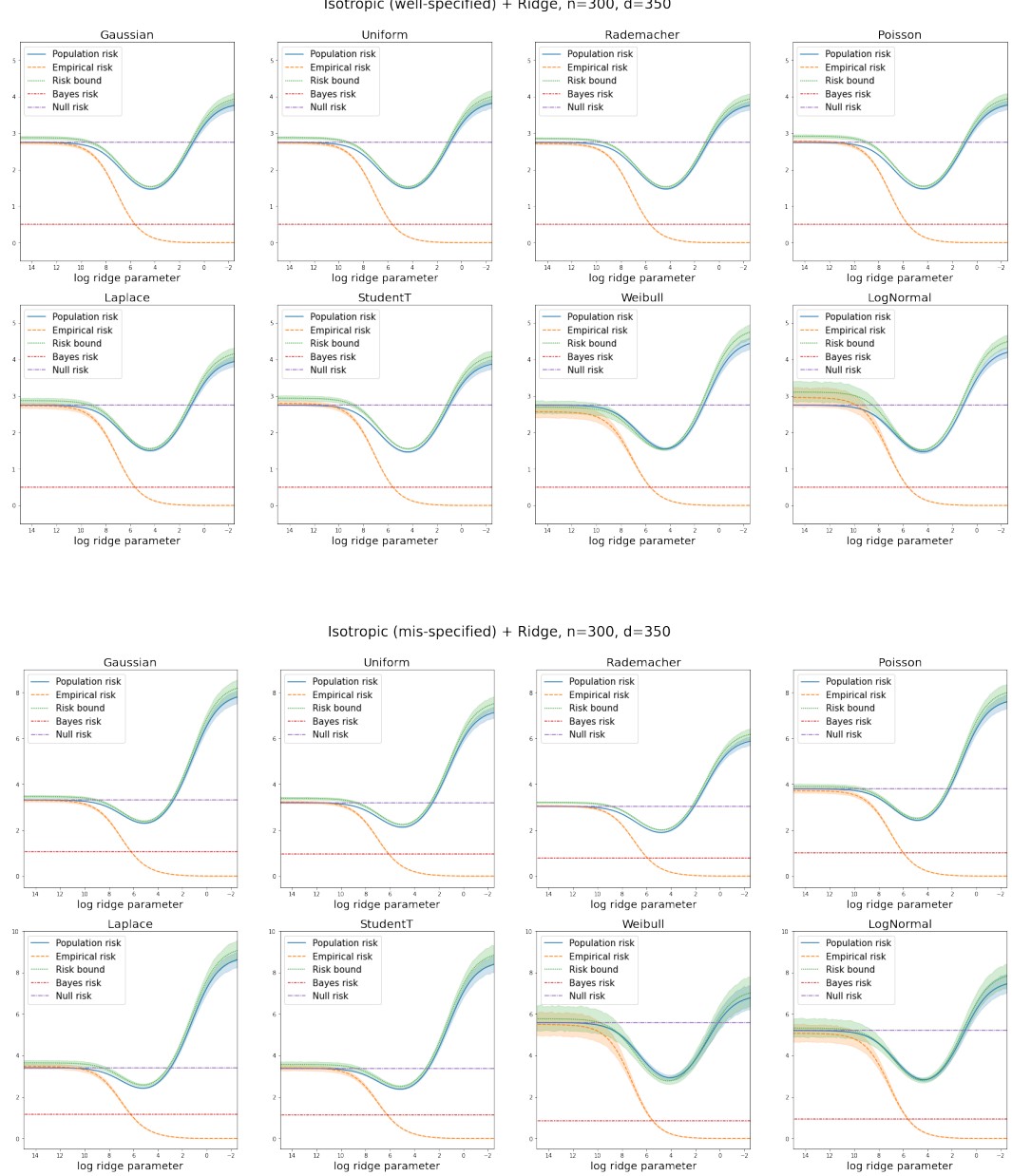

Figure 3: Ridge regression with isotropic data ($n = 300, d = 350$). As proved by theorem 7 in Zhou et al. (2021), the risk bound (18) follows the test error curve closely. This is true even in the non-Gaussian and mis-specified settings. Note that we do not have benign-overfitting because we are in the proportional scaling regime with $d$ close to $n$, and the population risk of the minimal-$\ell_2$ norm interpolator is even worse than the null-risk (more significantly so with misspecification). The optimally-tuned ridge regression has risk better than the null risk, but it is still far from the Bayes risk because the consistency result of optimally-tuned ridge regression in Zhou et al. (2021) assumes $\text{Tr}(\Sigma)/n \to 0$.

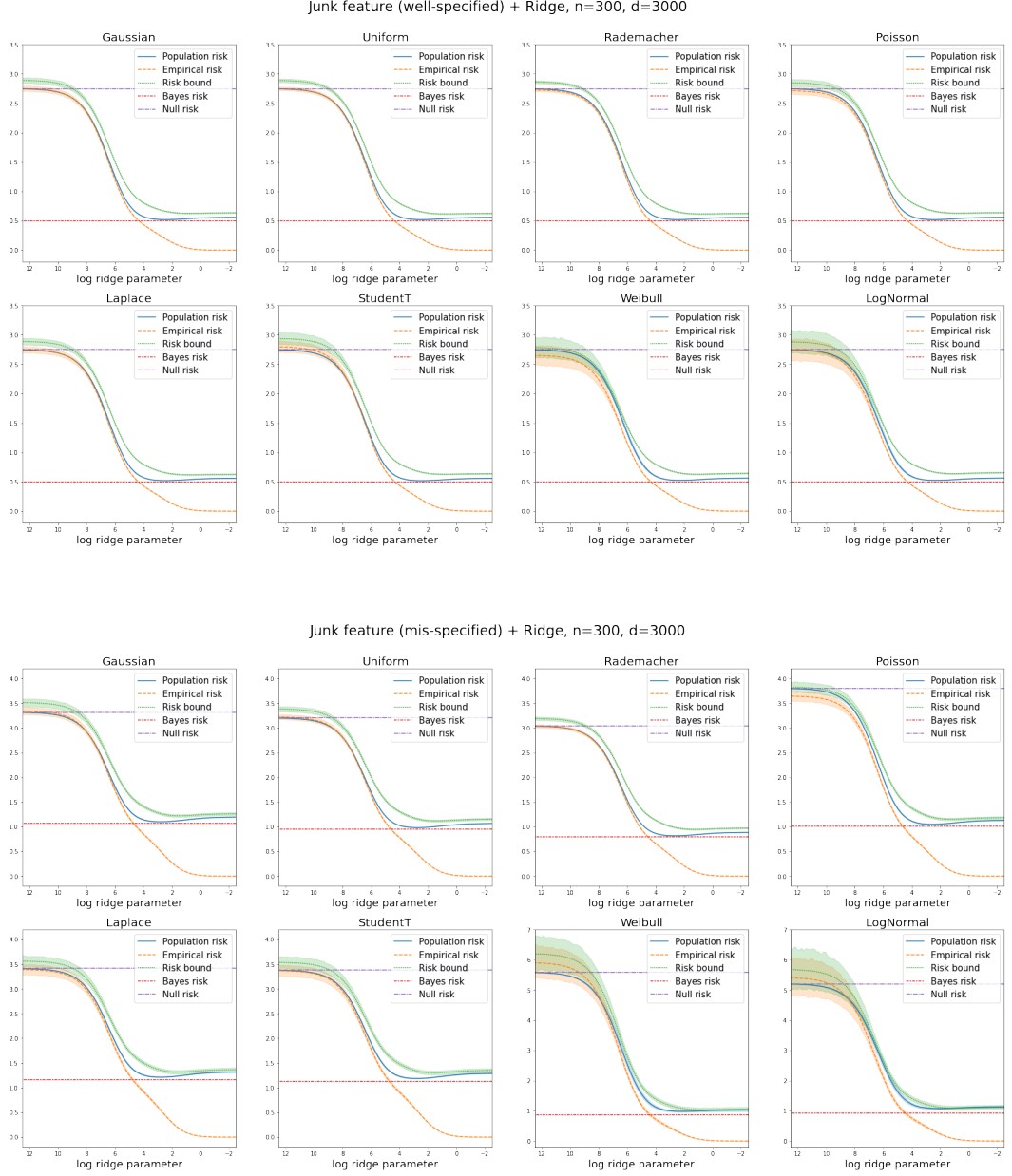

Figure 4: Ridge regression with junk features ($n = 300, d = 3000$). In the junk features setting, as predicted in section 6, the test error curve is essentially flat once the regularization is small enough to fit the signal, and we get nearly optimal population risk as long as we do not over-regularize the predictor. The test error curve can be expected to be more flat with increasing $d$. This phenomenon is also consistent across different feature distributions and label generating processes. Our bound (19) closely tracks the performance of ridge regression along the entire regularization path.

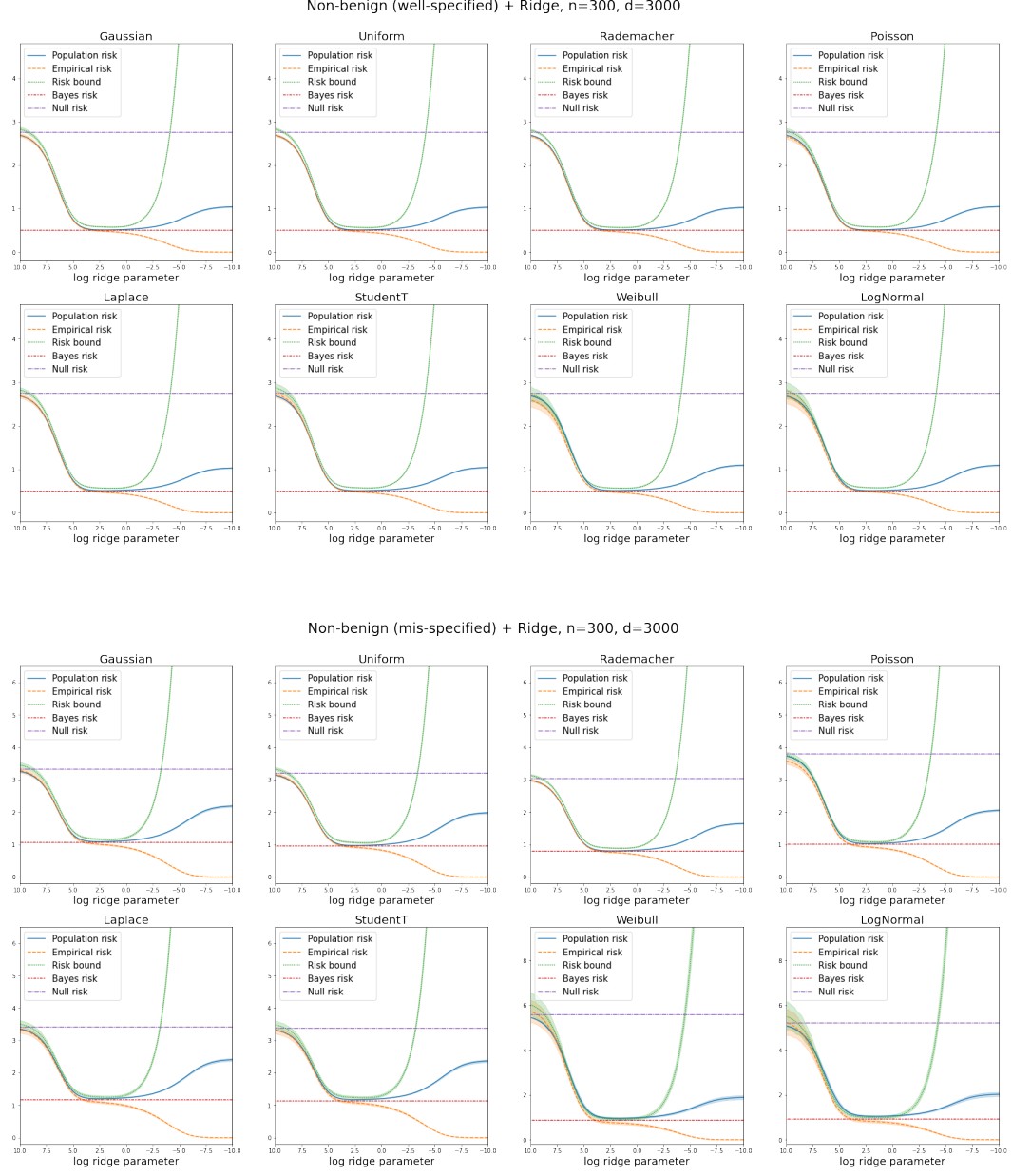

Figure 5: Ridge regression with non-benign features ($n = 300, d = 3000$). In the non-benign features setting, as proved by corollary 3 in Zhou et al. (2021), the optimally-tuned ridge regression achieves nearly optimal prediction risk. Our risk bound is tight up to the point up to the point where the test error starts to increase. As expected, the minimal norm interpolator fails to achieve consistency even though we are in the overparameterized regime. Note that bound (19) is dramatically more pessimistic in the under-regularized part of the ridge path. Once again, the data distribution and model misspecification has no effect on the shape of the test error curve and risk bound.

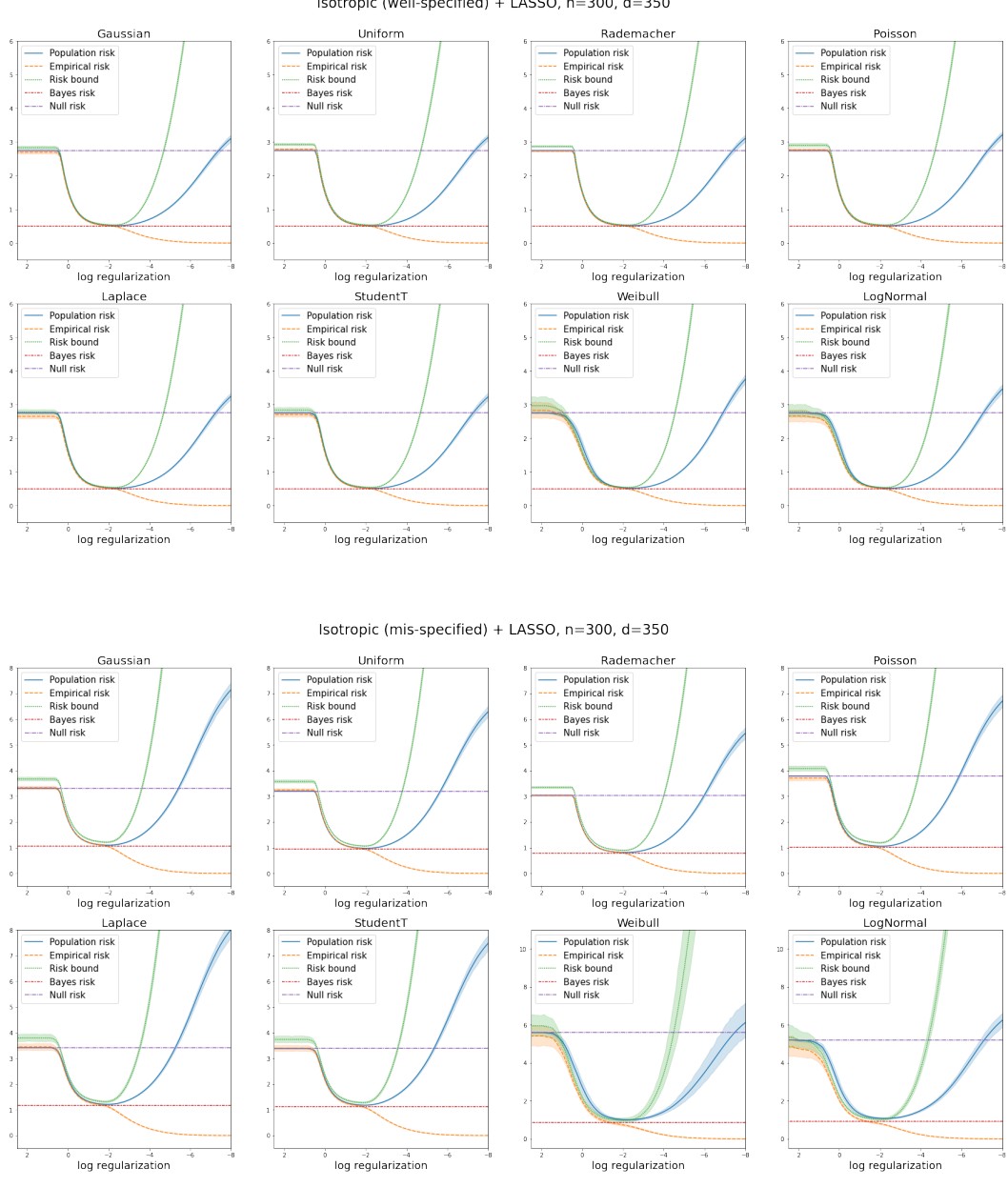

Figure 6: LASSO regression with isotropic data ($n = 300, d = 350$). Contrary to the inconsistency of optimally-tuned ridge regression in this setting, the regularized LASSO estimator can achieve nearly optimal population risk thanks to sparsity. The risk bound (20) appears to be valid and sufficient for the consistency of optimal LASSO in the distributions that we consider, though it is not very tight for interpolation. Recall that the minimal-$\ell_1$ norm interpolator suffers from an exponentially slow convergence rate when $d = n^\alpha$ (G. Wang et al. 2021) and observe that the population risk of the minimal-$\ell_1$ norm interpolator is again worse than the null-risk.

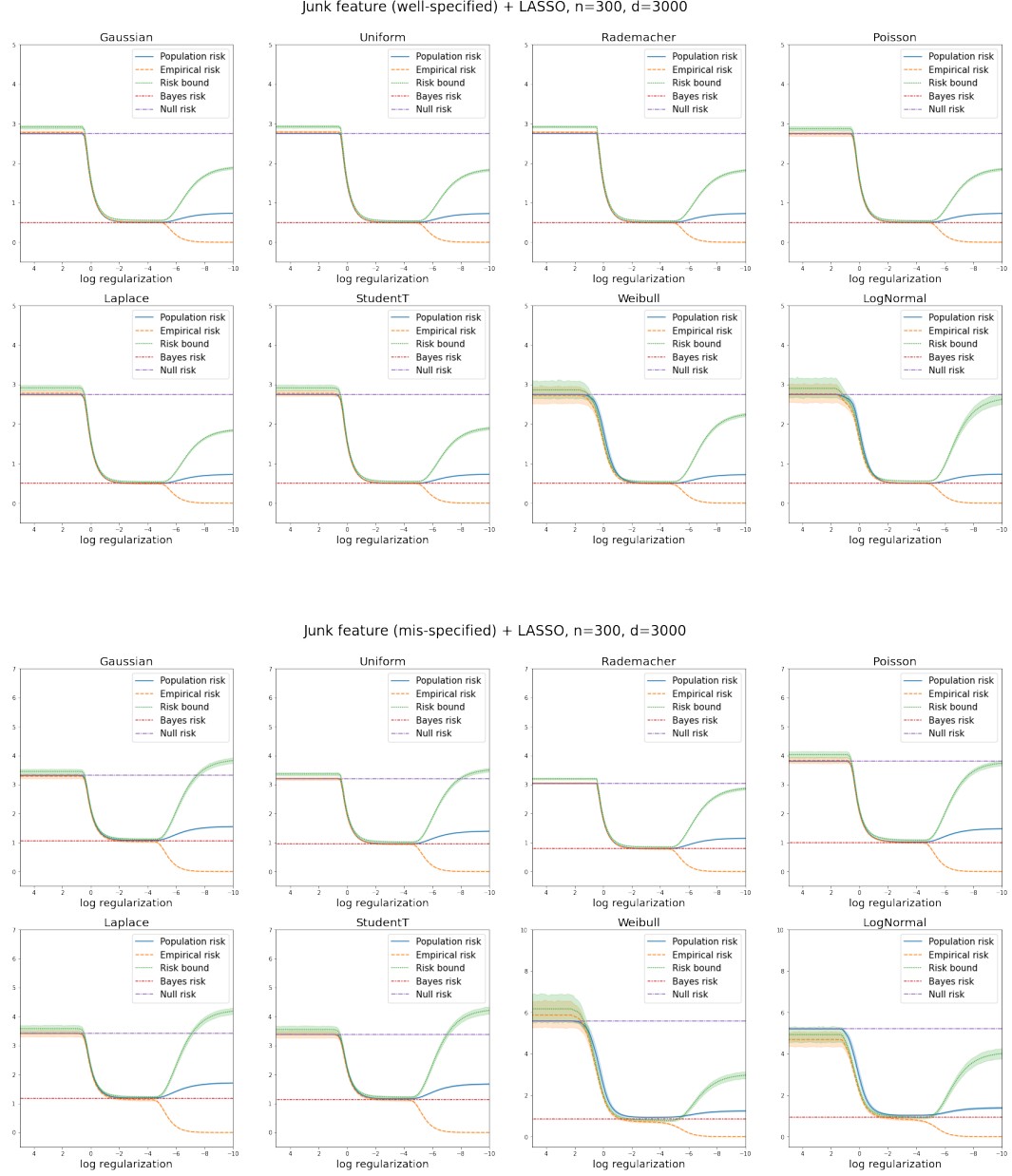

Figure 7: LASSO regression with junk features ($n = 300, d = 3000$). Similar to the isotropic setting, the regularized LASSO can achieve nearly optimal prediction risk and the risk bound (21) is sufficient to explain this phenomenon. Once again, the data distribution and model misspecification appear to have no effect on the shape of the test error curve. It is theoretically possible to use a nearly identical risk bound to show the consistency of minimal-$\ell_1$ norm interpolator when $n$ is large and $d$ is super-exponential in $n$ (Koehler et al. 2021), but as we can see, $n = 300$ and $d = 3000$ is not quite large enough yet. On the other hand, overfitting is more benign than what (21) predicts, suggesting a better analysis may yield a weaker condition required for consistency.

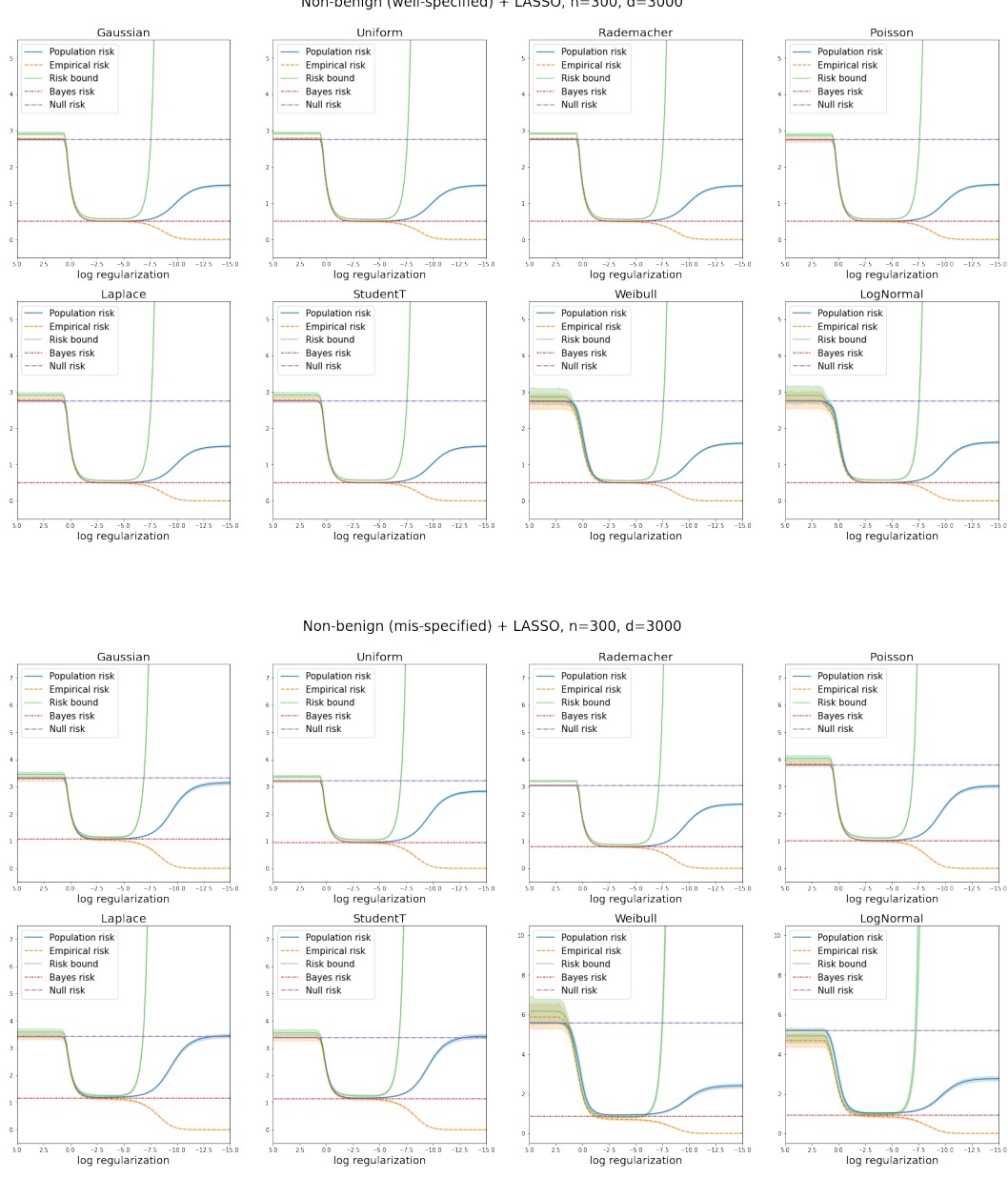

Figure 8: LASSO regression with non-benign features ($n = 300, d = 3000$). Though the population risk and the associated risk bound of regularized LASSO can be quite close to the Bayes risk, overfitting with minimal-$\ell_1$ norm interpolator does not appear to be benign (and there is no existing theoretical result suggesting that consistency is possible with a larger $n$ or $d$). In particular, its $\ell_1$ norm increases much more quickly than the junk-features case. Though the (21) is not tight throughout the entire regularization path, it is still a valid upper bound on the test error across different feature distributions and label generating processes.

## B.2 Linear Classification

Similarly, we fit linear models to minimize the squared hinge loss with $\ell_2$ and $\ell_1$ penalty. We can consider the same feature distributions and data covariance structure as in the preceding section. For faster computation (because margin classifiers can be slower to compute than regressors), we take $k = 1$, and $n = 100, d = 120$ in the proportional scaling and $n = 100, d = 2000$ in the overparameterized scaling. The label $y$ is generated by the following model:

$$\eta = \langle w^*, x \rangle + b^*, \quad \Pr(y = 1 \mid x) = 1 - \Pr(y = -1 \mid x) = g(\eta)$$

where $g : \mathbb{R} \to [0, 1]$ is the logistic link function. Since we use the squared hinge loss for learning (which is not the negative log-likelihood function), the linear model that we learn is not necessarily well-calibrated and so this can also be considered as a mis-specified setting. Therefore, we will only consider one label generating process in the classification context. Finally, by our Moreau envelope theory, we can use completely the same risk bounds from (18) to (21) for $\ell_2$ and $\ell_1$ margin classifiers.

### B.2.1 Experimental Results

The plots for $\ell_2$ and $\ell_1$ margin classifiers can be found in Figures 9 and 10. Each figure contain three subplots, and each subplot corresponds to one of the data covariance and contains the risk curves measured in squared hinge loss for the eight feature distributions.

$\ell_2$-**Margin Classifiers.** As in the regression case, overfitting is not benign when the features are isotropic and the population risk of $\ell_2$ max-margin classifier can be worse than the null risk. The risk bounds tightly control the test errors across different feature distributions. The difference between risk bound and the actual test error is larger when the feature distribution is heavy-tailed, but the confidence interval is also wider due to the relatively small sample size.

In the junk feature setting, the under-regularized part of the regularization path is essentially flat for all feature distributions. Overall, the experimental result is very similar to Figure 4, as predicted by our theory in section 6. The non-benign case is also similar to Figure 5 except that the U-shape curve is quite narrower near the optimal amount of regularization.

$\ell_1$-**Margin Classifiers.** In each of the subplots, the risk bound is tight only up to a certain point before the $\ell_1$ norm starts to increase quite a lot, leading to loose bound near interpolation. However, the risk bound is tight enough to establish consistency of optimally-tuned predictor in the junk and non-benign features setting. Again, the population risk of $\ell_1$ max-margin classifier can be worse than the null risk even in the junk features setting. Observe that different distributions do not seem to change the shape of generalization curve, and there is an interesting multiple descent phenomenon in the non-benign feature case, which has already been discovered in previous literature (Li and Wei 2021; Liang et al. 2020; Chen et al. 2021).

### B.2.2 Note on Computing the Population Risk with Gaussian Features

When the feature distribution is Gaussian, we can estimate

$$L_f(w, b) = \mathbb{E}\left[\max(0, 1 - y(\langle w, x \rangle + b))^2\right]$$

without drawing a new high-dimensional dataset from $\mathcal{D}$. First, we can write $x = \Sigma^{1/2} z$. Note that conditioning on $\eta$ is the same as conditioning on $\langle w^*, x \rangle = \langle \Sigma^{1/2} w^*, z \rangle \sim \mathcal{N}(0, \|w^*\|_\Sigma^2)$ and the conditional distribution of $z$ is

$$\frac{\eta - b^*}{\|w^*\|_\Sigma^2} \Sigma^{1/2} w^* + Pz$$

where $P = I - \frac{(\Sigma^{1/2} w^*)(\Sigma^{1/2} w^*)^T}{\|w^*\|_\Sigma^2}$ and so the conditional distribution of $\langle w, x \rangle + b$ is

$$\left\langle w, \Sigma^{1/2}\left(\frac{\eta - b^*}{\|w^*\|_\Sigma^2} \Sigma^{1/2} w^* + Pz\right)\right\rangle + b$$

$$= b + \frac{\langle w, \Sigma w^* \rangle}{\|w^*\|_\Sigma^2}(\eta - b^*) + \langle P\Sigma^{1/2} w, z \rangle \sim \mathcal{N}\left(\mu(\eta), \sigma^2\right)$$

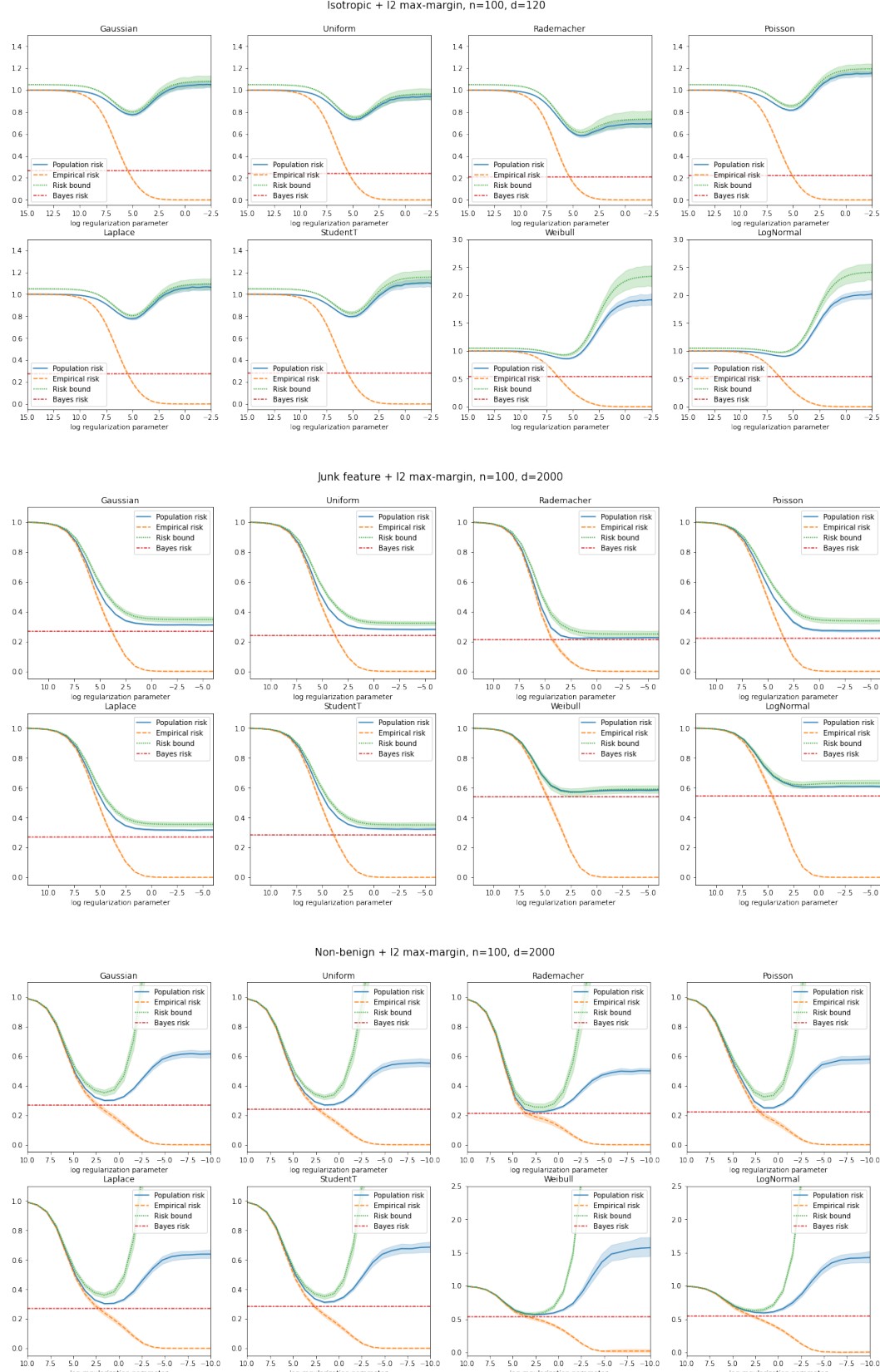

Figure 9: $\ell_2$ margin classification: isotropic, junk and non-benign features.

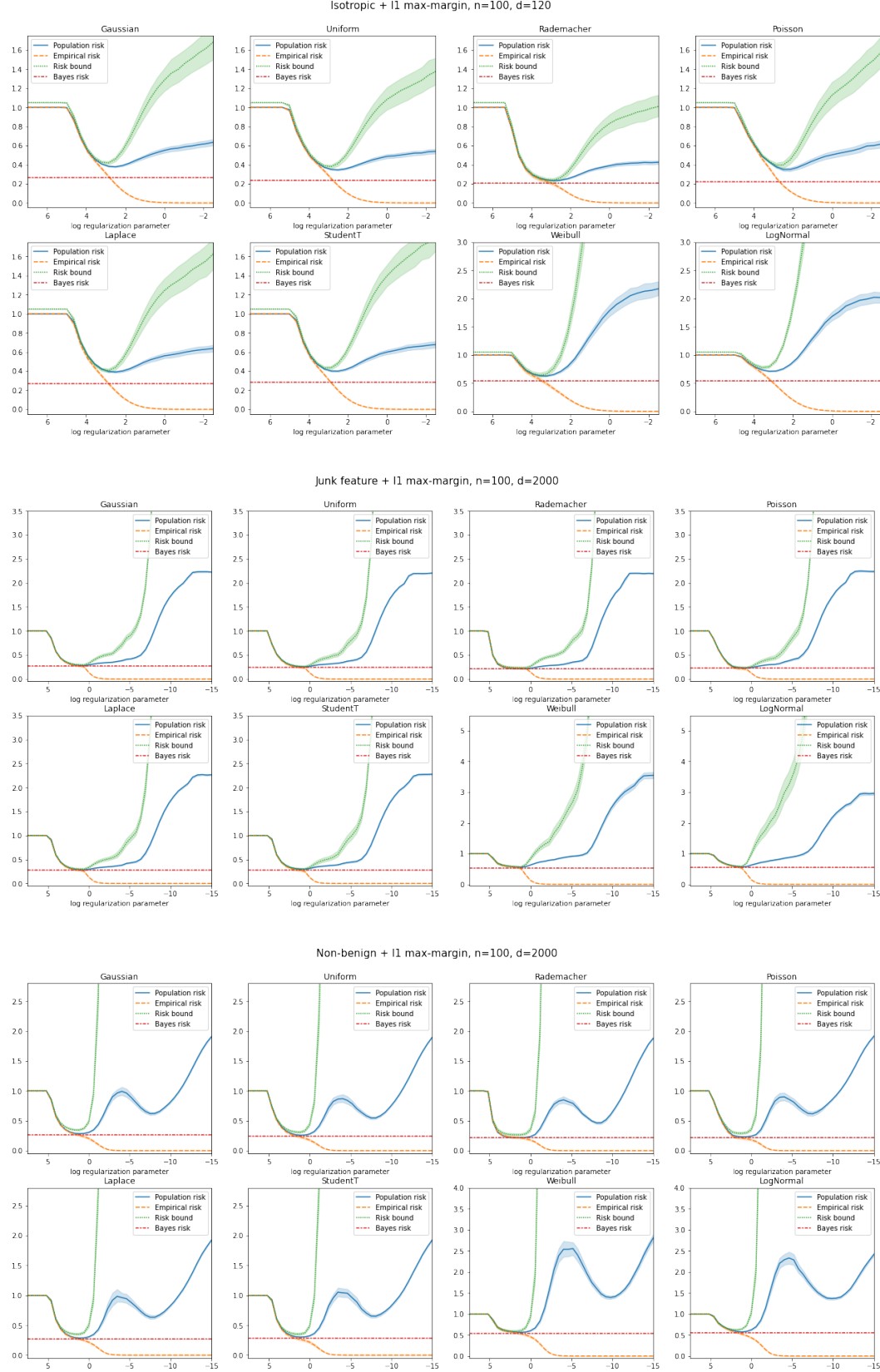

Figure 10: $\ell_1$ margin classification: isotropic, junk and non-benign features.

where $\mu(\eta) = b + \frac{\langle w, \Sigma w^* \rangle}{\|w^*\|_\Sigma^2}(\eta - b^*)$ and

$$\sigma^2 = w^T(\Sigma^{1/2} P \Sigma^{1/2})w = w^T \Sigma w - \frac{\langle w, \Sigma w^* \rangle^2}{\|w^*\|_\Sigma^2}.$$

Since $x$ is independent of $y$ conditioned on $\eta$, we have that

$$L(w, b) = \mathbb{E}\left[\mathbb{E}\left[\max(0, 1 - y(\langle w, x \rangle + b))^2 \mid \eta\right]\right]$$
$$= \mathbb{E}\left[g(\eta) \cdot \max(0, 1 - \mu(\eta) - \sigma z)^2 + (1 - g(\eta)) \cdot \max(0, 1 + \mu(\eta) + \sigma z)^2\right]$$

We can then estimate the population error by drawing samples from a two-dimensional distribution.

### B.2.3 Note on Computing the Optimal Linear Predictor

The linear predictor that minimizes the population squared hinge loss generally does not have a simple closed-form expression, but we can run SGD on the population objective in order to find the optimal linear predictor $\tilde{w}, \tilde{b}$. For simplicity, we choose

$$w^* = (5, 0, ..., 0) \quad \text{and} \quad b^* = 3.$$

In this case, we can simplify the optimization problem to an one-dimensional problem by observing that $\tilde{w}_i = 0$ for $i \neq 1$. Indeed, we can check the first order condition holds

$$\frac{\partial}{\partial w_i} L_f(\tilde{w}, \tilde{b}) = -2 \mathbb{E}\left[y \max(0, 1 - y(\langle \tilde{w}, x \rangle + \tilde{b}))x_i\right]$$
$$= -2 \mathbb{E}\left[y \max(0, 1 - y(\tilde{w}_1 x_1 + \tilde{b}))\right] \mathbb{E}[x_i] = 0$$

because $y$ is independent of $x_i$ with $i \neq 1$. Therefore, we can just generate $\{x_{i,1}, y_i\}$ from $\mathcal{D}$ and perform one-pass SGD (e.g. theorem 6.1 of Bubeck 2015) to find $\tilde{w}_1, \tilde{b}$. In the experiments, we find choosing the initial step size to be 0.1 works well.

## C  Preliminaries

**General Notation.**     Following the tradition in statistics, we denote $X = (x_1, ..., x_n)^T \in \mathbb{R}^{n \times d}$ as the design matrix. In the proof section, we slightly abuse the notation of $\eta_i$ to mean $Xw_i^*$ and $\xi$ to mean the $n$-dimensional random vector whose $i$-th component satisfies $y_i = g(\eta_{1,i}, ..., \eta_{k,i}, \xi_i)$. Note that we can write $X = Z\Sigma^{1/2}$ where $Z$ is a random matrix with i.i.d. standard normal entries.

We use the standard notation

$$\|x\|_\Sigma := \sqrt{\langle x, \Sigma x \rangle} \tag{22}$$

to denote the *Mahalonobis norm* with respect to positive semidefinite matrix $\Sigma$.

**Additional Covariance Split Notation.**     Because we will need to refer to the two parts of $\phi(w)$ often, in the remainder of the appendix we introduce the further notation $w^\perp = Qw$, $w^\| = (I - Q)w$ for the $\Sigma$-projection of $w$ onto the span of $w_1^*, \ldots, w_k^*$, and

$$r(w) := \|\Sigma^{1/2} Qw\| = \|Qw\|_\Sigma$$

for the Mahalanobis norm in the orthogonal space. We also will use the notation $X^\| = XQ$ and $X^\perp = X(I - Q)$ for the corresponding projections of the design matrix $X$, which are independent of each other.

**Concentration of Lipschitz functions.**     Recall that a function $f : \mathbb{R}^n \to \mathbb{R}$ is $L$-Lipschitz with respect to the norm $\|\cdot\|$ if it holds for all $x, y \in \mathbb{R}^n$ that $|f(x) - f(y)| \leq L\|x - y\|$. We use the concentration of Lipschitz functions of a Gaussian.

**Theorem 5** (van Handel 2014, Theorem 3.25). *If $f$ is $L$-Lipschitz with respect to the Euclidean norm and $Z \sim \mathcal{N}(0, I_n)$, then*

$$\Pr(|f(Z) - \mathbb{E}f(Z)| \geq t) \leq 2e^{-t^2/2L^2}. \tag{23}$$

The following straightforward concentration result is Lemma 2 of Koehler et al. (2021).

**Lemma 3.** *Suppose that $Z \sim \mathcal{N}(0, I_n)$. Then*

$$\Pr\left(\left|\|Z\|_2 - \sqrt{n}\right| \geq t\right) \leq 4e^{-t^2/4}. \tag{24}$$

We will use the following to help relate our problem to the surrogate distribution in our proof of Theorem 1.

**Lemma 4.** *Fix any integer $k < d$ and any $k$ vectors $w_1^*, ..., w_k^*$ in $\mathbb{R}^d$ such that $\Sigma^{1/2}w_1^*, ..., \Sigma^{1/2}w_k^*$ are orthonormal. Denoting*

$$P = I_d - \sum_{i=1}^{k}(\Sigma^{1/2}w_i^*)(\Sigma^{1/2}w_i^*)^T, \tag{25}$$

*the distribution of $X$ conditional on $Xw_1^* = \eta_1, ..., Xw_k^* = \eta_k$ is the same as that of*

$$\sum_{i=1}^{k}\eta_i(\Sigma w_i^*)^T + ZP\Sigma^{1/2}. \tag{26}$$

*Proof.* We can write $X = Z\Sigma^{1/2}$. The key observation is that $ZP, Z\Sigma^{1/2}w_1^*, ..., Z\Sigma^{1/2}w_k^*$ are independent. To see why this is the case, we can vectorize each term:

$$\begin{pmatrix} \text{vec}(ZP) \\ \text{vec}(Z\Sigma^{1/2}w_1^*) \\ ... \\ \text{vec}(Z\Sigma^{1/2}w_k^*) \end{pmatrix} = \begin{pmatrix} P \otimes I_n \\ (\Sigma^{1/2}w_1^*)^T \otimes I_n \\ ... \\ (\Sigma^{1/2}w_k^*)^T \otimes I_n \end{pmatrix} \text{vec}(Z)$$

From the above representation, we see that the joint distribution is multivariate Gaussian and the covariance matrix is

$$\begin{pmatrix} P \otimes I_n \\ (\Sigma^{1/2}w_1^*)^T \otimes I_n \\ ... \\ (\Sigma^{1/2}w_k^*)^T \otimes I_n \end{pmatrix} \begin{pmatrix} P \otimes I_n \\ (\Sigma^{1/2}w_1^*)^T \otimes I_n \\ ... \\ (\Sigma^{1/2}w_k^*)^T \otimes I_n \end{pmatrix}^T = \text{diag}\left(P \otimes I_n, I_n, ..., I_n\right)$$

Therefore, the distribution of $ZP$ remains unchanged after conditioning on $Z\Sigma^{1/2}w_1^*, ..., Z\Sigma^{1/2}w_k^*$, and we can write

$$Z = Z\left(\sum_{i=1}^{k}(\Sigma^{1/2}w_i^*)(\Sigma^{1/2}w_i^*)^T\right) + ZP$$

$$= \sum_{i=1}^{k}\eta_i(\Sigma^{1/2}w_i^*)^T + ZP.$$

The proof is concluded by the fact that $X = Z\Sigma^{1/2}$. $\square$

A key ingredient of our technique is the Gaussian minimax theorem.

**Theorem 6** ((Convex) Gaussian Minmax Theorem; Thrampoulidis et al. 2015; Gordon 1985)**.** *Let $Z : n \times d$ be a matrix with i.i.d. $N(0, 1)$ entries and suppose $G \sim \mathcal{N}(0, I_n)$ and $H \sim \mathcal{N}(0, I_d)$ are independent of $Z$ and each other. Let $S_w, S_u$ be compact sets and $\psi : S_w \times S_u \to \mathbb{R}$ be an arbitrary continuous function. Define the* Primary Optimization (PO) *problem*

$$\Phi(Z) := \min_{w \in S_w} \max_{u \in S_u} \langle u, Zw \rangle + \psi(w, u) \tag{27}$$

*and the* Auxiliary Optimization (AO) *problem*

$$\phi(G, H) := \min_{w \in S_w} \max_{u \in S_u} \|w\|_2 \langle G, u \rangle + \|u\|_2 \langle H, w \rangle + \psi(w, u). \tag{28}$$

*Under these assumptions, $\Pr(\Phi(Z) < c) \leq 2\Pr(\phi(G, H) \leq c)$ for any $c \in \mathbb{R}$.*

*Furthermore, if we suppose that $S_w, S_u$ are convex sets and $\psi(w, u)$ is convex in $w$ and concave in $u$, then $\Pr(\Phi(Z) > c) \leq 2\Pr(\phi(G, H) \geq c)$.*

# D    Proof of Theorem 1

First, let's try to formulate the generalization problem as a PO:

**Lemma 5.** *For any deterministic function $F : \mathbb{R}^d \times \mathbb{R} \to \mathbb{R}^+$, define the primary optimization (PO) problem conditioned on $\eta_1, ..., \eta_k, \xi$ as*

$$\Phi := \sup_{\substack{(w,b)\in\mathbb{R}^{d+1} \\ u\in\mathbb{R}^n}} \inf_{\lambda\in\mathbb{R}^n} \langle \lambda, Z(P\Sigma^{1/2}w)\rangle + \psi(w, b, u, \lambda \,|\, \eta_1, ..., \eta_k, \xi) \tag{29}$$

*where $P$ is defined by (25) in Lemma 4 and*

$$\psi(w, b, u, \lambda \,|\, \eta_1, ..., \eta_k, \xi) = F(w, b) + \langle \lambda, \sum_{i=1}^{k} \eta_i \langle w, \Sigma w_i^* \rangle - u \rangle \tag{30}$$
$$- \frac{1}{n}\sum_{i=1}^{n} f(u_i + b, g(\eta_{1,i}, ..., \eta_{k,i}, \xi_i)).$$

*Then it holds that for any $t \in \mathbb{R}$*

$$\Pr\left( \sup_{(w,b)\in\mathbb{R}^{d+1}} F(w,b) - \hat{L}_f(w,b) > t \,\Big|\, \eta_1, ..., \eta_k, \xi \right) = \Pr\left( \Phi > t \right) \tag{31}$$

*and the probability over $\Phi$ is taken only over the randomness of $Z$.*

*Proof.* By introducing a variable $u = Xw$, we have

$$\sup_{(w,b)\in\mathbb{R}^{d+1}} F(w,b) - \hat{L}_f(w,b)$$

$$= \sup_{(w,b)\in\mathbb{R}^{d+1}} F(w,b) - \frac{1}{n}\sum_{i=1}^{n} f(\langle w, x_i \rangle + b, y_i)$$

$$= \sup_{\substack{(w,b)\in\mathbb{R}^{d+1},u\in\mathbb{R}^n \\ u=Xw}} F(w,b) - \frac{1}{n}\sum_{i=1}^{n} f(u_i + b, g(\eta_{1,i}, ..., \eta_{k,i}, \xi_i))$$

$$= \sup_{(w,b)\in\mathbb{R}^{d+1},u\in\mathbb{R}^n} \inf_{\lambda\in\mathbb{R}^n} \langle \lambda, Xw - u \rangle + F(w,b) - \frac{1}{n}\sum_{i=1}^{n} f(u_i + b, g(\eta_{1,i}, ..., \eta_{k,i}, \xi_i))$$

and so by independence of $\xi$ and $X$ and Lemma 4, it holds that for any $t \in \mathbb{R}$

$$\Pr\left( \sup_{(w,b)\in\mathbb{R}^{d+1}} F(w,b) - \hat{L}_f(w,b) > t \,\Big|\, \eta_1, ..., \eta_k, \xi \right)$$

$$= \Pr\left( \sup_{\substack{(w,b)\in\mathbb{R}^{d+1} \\ u\in\mathbb{R}^n}} \inf_{\lambda\in\mathbb{R}^n} \langle \lambda, \left( \sum_{i=1}^{k} \eta_i(\Sigma w_i^*)^T + ZP\Sigma^{1/2} \right) w - u \rangle + F(w,b) - \frac{1}{n}\sum_{i=1}^{n} f(u_i + b, g(\eta_{1,i}, ..., \eta_{k,i}, \xi_i)) > t \right)$$

$$= \Pr\left( \sup_{\substack{(w,b)\in\mathbb{R}^{d+1} \\ u\in\mathbb{R}^n}} \inf_{\lambda\in\mathbb{R}^n} \langle \lambda, ZP\Sigma^{1/2}w \rangle + \psi(w, b, u, \lambda \,|\, \eta_1, ..., \eta_k, \xi) > t \right)$$

$$= \Pr\left( \Phi > t \right).$$

Note that this probability is a random variable measurable with respect to the random vectors $\eta_1, ..., \eta_k$ and $\xi$. □

Next, let's use a truncation argument similar to the one in Koehler et al. (2021) and then apply GMT. Proving the following two lemmas is an exercise in real analysis, which we include for completeness.

**Lemma 6.** *Let $f : \mathbb{R}^d \to \mathbb{R}$ be an arbitrary function and $\mathcal{S}_r^d = \{x \in \mathbb{R}^d : \|x\|_2 \leq r\}$, then for any set $\mathcal{K}$, it holds that*

$$\lim_{r \to \infty} \sup_{w \in \mathcal{K} \cap \mathcal{S}_r^d} f(w) = \sup_{w \in \mathcal{K}} f(w). \tag{32}$$

*If $f$ is a random function, then for any $t \in \mathbb{R}$*

$$\Pr\left(\sup_{w \in \mathcal{K}} f(w) > t\right) = \lim_{r \to \infty} \Pr\left(\sup_{w \in \mathcal{K} \cap \mathcal{S}_r^d} f(w) > t\right). \tag{33}$$

*Proof.* We consider two cases:

1. Suppose that $\sup_{w \in \mathcal{K}} f(w) = \infty$. Then for any $M > 0$, there exists $x_M \in \mathcal{K}$ such that $f(x_M) > M$. Hence for any $r > \|x_M\|_2$, it holds that

$$\sup_{w \in \mathcal{K} \cap \mathcal{S}_r^d} f(w) > M \implies \liminf_{r \to \infty} \sup_{w \in \mathcal{K} \cap \mathcal{S}_r^d} f(w) \geq M$$

   As the choice of $M$ is arbitrary, we have $\lim_{r \to \infty} \sup_{w \in \mathcal{K} \cap \mathcal{S}_r^d} f(w) = \infty$ as desired.

2. Suppose that $\sup_{w \in \mathcal{K}} f(w) = M < \infty$. Then for any $\epsilon > 0$, there exists $x_\epsilon \in \mathcal{K}$ such that $f(x_\epsilon) > M - \epsilon$. Hence for any $r > \|x_\epsilon\|_2$, it holds that

$$\sup_{w \in \mathcal{K} \cap \mathcal{S}_r^d} f(w) > M - \epsilon \implies \liminf_{r \to \infty} \sup_{w \in \mathcal{K} \cap \mathcal{S}_r^d} f(w) \geq M - \epsilon$$

   As the choice of $\epsilon$ is arbitrary, we have $\liminf_{r \to \infty} \sup_{w \in \mathcal{K} \cap \mathcal{S}_r^d} f(w) \geq M$. On the other hand, it must be the case (by definition of supremum) that

$$\sup_{w \in \mathcal{K} \cap \mathcal{S}_r^d} f(w) \leq M \implies \limsup_{r \to \infty} \sup_{w \in \mathcal{K} \cap \mathcal{S}_r^d} f(w) \leq M$$

   Consequently, the limit of $\sup_{w \in \mathcal{K} \cap \mathcal{S}_r^d} f(w)$ exists and equals $M$.

Finally, by the fact that the supremum is increasing in $r$ and the continuity of probability measure, we have

$$\Pr\left(\sup_{w \in \mathcal{K}} f(w) > t\right) = \Pr\left(\lim_{r \to \infty} \sup_{w \in \mathcal{K} \cap \mathcal{S}_r^d} f(w) > t\right)$$

$$= \Pr\left(\bigcup_{r \in \mathbb{N}} \bigcap_{R \geq r} \sup_{w \in \mathcal{K} \cap \mathcal{S}_R^d} f(w) > t\right)$$

$$= \lim_{r \to \infty} \Pr\left(\bigcap_{R \geq r} \sup_{w \in \mathcal{K} \cap \mathcal{S}_R^d} f(w) > t\right)$$

$$= \lim_{r \to \infty} \Pr\left(\sup_{w \in \mathcal{K} \cap \mathcal{S}_r^d} f(w) > t\right). \qquad \square$$

**Lemma 7.** *Let $\mathcal{K}$ be a compact set and $f, g$ be continuous real-valued functions on $\mathbb{R}^d$. Then it holds that*

$$\lim_{r \to \infty} \sup_{w \in \mathcal{K}} \inf_{0 \leq \lambda \leq r} \lambda f(w) + g(w) = \sup_{w \in \mathcal{K}: f(w) \geq 0} g(w). \tag{34}$$

*If $f$ and $g$ are random functions, then for any $t \in \mathbb{R}$*

$$\Pr\left(\sup_{w \in \mathcal{K}: f(w) \geq 0} g(w) \geq t\right) = \lim_{r \to \infty} \Pr\left(\sup_{w \in \mathcal{K}} \inf_{0 \leq \lambda \leq r} \lambda f(w) + g(w) \geq t\right). \tag{35}$$

*Proof.* We consider two cases:

1. The limiting problem is infeasible: $\forall w \in \mathcal{K}, f(w) < 0$. Then by compactness and the continuity of $f$, there exists $\mu < 0$ such that for all $w \in \mathcal{K}$

$$f(w) < \mu \implies \sup_{w \in \mathcal{K}} \inf_{0 \leq \lambda \leq r} \lambda f(w) + g(w) \leq r\mu + \sup_{w \in \mathcal{K}} g(w).$$

   By compactness and the continuity of $g$ again, we have $\sup_{w \in \mathcal{K}} g(w) < \infty$ and so

$$\lim_{r \to \infty} \sup_{w \in \mathcal{K}} \inf_{0 \leq \lambda \leq r} \lambda f(w) + g(w) = -\infty$$

   as desired.

2. The limiting problem is feasible: $\exists w_0 \in \mathcal{K}, f(w_0) \geq 0$. In this case, let

$$w_r = \arg\max_{w \in \mathcal{K}} \inf_{0 \leq \lambda \leq r} \lambda f(w) + g(w)$$
$$= \arg\max_{w \in \mathcal{K}} r \cdot f(w) \mathbb{1}_{\{f(w) \leq 0\}} + g(w)$$

   be an arbitrary maximizer for each $r$. Note that a maximizer necessarily exists in $\mathcal{K}$ by compactness of $\mathcal{K}$ and the continuity of $f$ and $g$. By compactness of $\mathcal{K}$ again, the sequence $\{w_r\}$ at positive integer values of $r$ has a subsequential limit: $\exists r_n \to \infty$ and $w_\infty \in \mathcal{K}$ such that $w_{r_n} \to w_\infty$.

   For the sake of contradiction, assume that $f(w_\infty) < 0$, then by continuity, there exists $\mu < 0$ such that for all sufficiently large $n$

$$f(w_{r_n}) < \mu \implies \sup_{w \in \mathcal{K}} \inf_{0 \leq \lambda \leq r_n} \lambda f(w) + g(w) = r_n \cdot f(w_{r_n}) + g(w_{r_n}) \leq r_n \mu + \sup_{w \in \mathcal{K}} g(w)$$

   which is unbounded from below as $n \to \infty$. On the other hand, we have

$$\sup_{w \in \mathcal{K}} \inf_{0 \leq \lambda \leq r_n} \lambda f(w) + g(w) \geq g(w_0)$$

   and so we have reached a contradiction; thus $f(w_\infty) \geq 0$. Observe that

$$\sup_{w \in \mathcal{K}} \inf_{0 \leq \lambda \leq r_n} \lambda f(w) + g(w) = r_n \cdot f(w_{r_n}) \mathbb{1}_{\{f(w_{r_n}) \leq 0\}} + g(w_{r_n}) \leq g(w_{r_n})$$

   and so by continuity of $g$

$$\limsup_{n \to \infty} \sup_{w \in \mathcal{K}} \inf_{0 \leq \lambda \leq r_n} \lambda f(w) + g(w) \leq g(w_\infty) \leq \sup_{w \in \mathcal{K}: f(w) \geq 0} g(w).$$

   The $\liminf$ direction follows immediately from the definition, and so the limit exists and equals $\sup_{w \in \mathcal{K}: f(w) \geq 0} g(w)$. We can conclude that

$$\lim_{r \to \infty} \sup_{w \in \mathcal{K}} \inf_{0 \leq \lambda \leq r} \lambda f(w) + g(w) = \sup_{w \in \mathcal{K}: f(w) \geq 0} g(w)$$

   because it is a monotonic sequence.

Finally, by the fact that the supremum is decreasing in $r$ and the continuity of probability measure, we have

$$\Pr\left(\sup_{w \in \mathcal{K}: f(w) \geq 0} g(w) \geq t\right) = \Pr\left(\lim_{r \to \infty} \sup_{w \in \mathcal{K}} \inf_{0 \leq \lambda \leq r} \lambda f(w) + g(w) \geq t\right)$$
$$= \Pr\left(\cap_r \sup_{w \in \mathcal{K}} \inf_{0 \leq \lambda \leq r} \lambda f(w) + g(w) \geq t\right)$$
$$= \lim_{r \to \infty} \Pr\left(\sup_{w \in \mathcal{K}} \inf_{0 \leq \lambda \leq r} \lambda f(w) + g(w) \geq t\right). \qquad \square$$

We are now ready to apply the GMT:

**Lemma 8.** *Let $F$ be a continuous function. Consider the auxiliary problem*

$$\Psi := \sup_{\substack{(w,b) \in \mathbb{R}^{d+1}, u \in \mathbb{R}^n \\ \langle H, P\Sigma^{1/2}w \rangle \geq \left\| G\|P\Sigma^{1/2}w\|_2 + \sum_{i=1}^{k} \langle w, \Sigma w_i^* \rangle \eta_i - u \right\|_2}} F(w,b) - \frac{1}{n} \sum_{i=1}^{n} f(u_i + b, g(\eta_{1,i}, ..., \eta_{k,i}, \xi_i)).$$

*It holds that for any $t \in \mathbb{R}$ and $\Phi$ defined as in Lemma 5 that*

$$\Pr(\Phi > t) \leq 2 \Pr(\Psi \geq t). \tag{36}$$

*Proof.* Define the truncated problems

$$\Phi_r := \sup_{(w,b,u) \in \mathcal{S}_r^{d+n+1}} \inf_{\lambda \in \mathbb{R}^n} \langle \lambda, Z(P\Sigma^{1/2}w) \rangle + \psi(w, b, u, \lambda \,|\, \eta_1, ..., \eta_k, \xi) \tag{37}$$

and

$$\Phi_{r,s} := \sup_{(w,b,u) \in \mathcal{S}_r^{d+n+1}} \inf_{\|\lambda\|_2 \leq s} \langle \lambda, Z(P\Sigma^{1/2}w) \rangle + \psi(w, b, u, \lambda \,|\, \eta_1, ..., \eta_k, \xi). \tag{38}$$

By definition, we have $\Phi_r \leq \Phi_{r,s}$ and so

$$\Pr(\Phi_r > t) \leq \Pr(\Phi_{r,s} > t).$$

The corresponding auxiliary problems are

$$\Psi_{r,s} := \sup_{(w,b,u) \in \mathcal{S}_r^{d+n+1}} \inf_{\|\lambda\|_2 \leq s} \|\lambda\|_2 \langle H, P\Sigma^{1/2}w \rangle + \|P\Sigma^{1/2}w\|_2 \langle G, \lambda \rangle + \psi(w, b, u, \lambda \,|\, \eta_1, ..., \eta_k, \xi)$$

$$= \sup_{(w,b,u) \in \mathcal{S}_r^{d+n+1}} \inf_{\|\lambda\|_2 \leq s} \|\lambda\|_2 \langle H, P\Sigma^{1/2}w \rangle + \langle G\|P\Sigma^{1/2}w\|_2 + \sum_{i=1}^{k} \eta_i \langle w, \Sigma w_i^* \rangle - u, \lambda \rangle$$

$$+ F(w,b) - \frac{1}{n} \sum_{i=1}^{n} f(u_i + b, g(\eta_{1,i}, ..., \eta_{k,i}, \xi_i))$$

$$= \sup_{(w,b,u) \in \mathcal{S}_r^{d+n+1}} \inf_{0 \leq \lambda \leq s} \lambda \left( \langle H, P\Sigma^{1/2}w \rangle - \left\| G\|P\Sigma^{1/2}w\|_2 + \sum_{i=1}^{k} \eta_i \langle w, \Sigma w_i^* \rangle - u \right\|_2 \right)$$

$$+ F(w,b) - \frac{1}{n} \sum_{i=1}^{n} f(u_i + b, g(\eta_{1,i}, ..., \eta_{k,i}, \xi_i))$$

and

$$\Psi_r := \sup_{\substack{(w,b,u) \in \mathcal{S}_r^{d+n+1} \\ \langle H, P\Sigma^{1/2}w \rangle \geq \left\| G\|P\Sigma^{1/2}w\|_2 + \sum_{i=1}^{k} \langle w, \Sigma w_i^* \rangle \eta_i - u \right\|_2}} F(w,b) - \frac{1}{n} \sum_{i=1}^{n} f(u_i + b, g(\eta_{1,i}, ..., \eta_{k,i}, \xi_i)).$$

By definition, it holds that $\Psi_r \leq \Psi$ and so

$$\Pr(\Psi_r \geq t) \leq \Pr(\Psi \geq t).$$

Thus

$$\begin{aligned}
\Pr(\Phi > t) &= \lim_{r \to \infty} \Pr(\Phi_r > t) \\
&\leq \lim_{r \to \infty} \lim_{s \to \infty} \Pr(\Phi_{r,s} > t) &&\text{by Lemma 6} \\
&\leq 2 \lim_{r \to \infty} \lim_{s \to \infty} \Pr(\Psi_{r,s} \geq t) &&\text{by Theorem 6} \\
&= 2 \lim_{r \to \infty} \Pr(\Psi_r \geq t) &&\text{by Lemma 7} \\
&\leq 2 \Pr(\Psi \geq t). &&\square
\end{aligned}$$

**Lemma 9.** *Let $\Psi$ be as in Lemma 8. Under the assumptions (6) and (7) in Theorem 1, it holds with probability at least $1 - \delta/2$ that*

$$\Psi \leq \sup_{(w,b) \in \mathbb{R}^{d+1}} F(w,b) - L_{f_\lambda}(w,b) + \epsilon_{\lambda,\delta}(\phi(w), b) + \frac{\lambda C_\delta(w)^2}{n}$$

*and if assumption* (6) *holds uniformly over all* $\lambda \in \mathbb{R}^+$*, then*

$$\Psi \leq \sup_{(w,b)\in\mathbb{R}^{d+1}} F(w,b) - \sup_{\lambda\in\mathbb{R}^+} \left[ L_{f_\lambda}(w,b) - \epsilon_{\lambda,\delta}(\phi(w),b) - \frac{\lambda C_\delta(w)^2}{n} \right]$$

*where the randomness is taken over* $H, G, \eta_1, ..., \eta_k$ *and* $\xi$.

*Proof.* First, let's simplify the auxiliary problem. Changing variables to subtract $G_i \left\| P\Sigma^{1/2}w \right\|_2 + \sum_{l=1}^k \langle w, \Sigma w_l^* \rangle \eta_{l,i}$ from each of the former $u_i$, we have that

$$\Psi = \sup_{\substack{(w,b,u)\in\mathbb{R}^{d+n+1} \\ \|u\|_2 \leq \langle H, P\Sigma^{1/2}w \rangle}} F(w,b) - \frac{1}{n}\sum_{i=1}^n f\left( G_i\|P\Sigma^{1/2}w\|_2 + \sum_{l=1}^k \langle w, \Sigma w_l^* \rangle \eta_{l,i} + b + u_i, g(\eta_{1,i}, ..., \eta_{k,i}, \xi_i) \right)$$

$$= \sup_{(w,b)\in\mathbb{R}^{d+1}} F(w,b) - \inf_{\substack{u\in\mathbb{R}^n \text{ s.t.} \\ \|u\|_2 \leq \langle \Sigma^{1/2}PH, w \rangle}} \frac{1}{n}\sum_{i=1}^n f\left( G_i\|P\Sigma^{1/2}w\|_2 + \sum_{l=1}^k \langle w, \Sigma w_l^* \rangle \eta_{l,i} + b + u_i, g(\eta_{1,i}, ..., \eta_{k,i}, \xi_i) \right)$$

We can analyze the second term. If $\langle \Sigma^{1/2}PH, w \rangle < 0$ then the constraint on $u$ is not satisfiable and so the infimum is $\infty$. Otherwise, by duality

$$\inf_{\substack{u\in\mathbb{R}^n \text{ s.t.} \\ \|u\|_2 \leq \langle \Sigma^{1/2}PH, w \rangle}} \sum_{i=1}^n f\left( G_i\|P\Sigma^{1/2}w\|_2 + \sum_{l=1}^k \langle w, \Sigma w_l^* \rangle \eta_{l,i} + b + u_i, g(\eta_{1,i}, ..., \eta_{k,i}, \xi_i) \right)$$

$$= \inf_{u\in\mathbb{R}^n} \sup_{\lambda\geq 0} \lambda(\|u\|_2^2 - \langle \Sigma^{1/2}PH, w \rangle^2) + \sum_{i=1}^n f\left( G_i\|P\Sigma^{1/2}w\|_2 + \sum_{l=1}^k \langle w, \Sigma w_l^* \rangle \eta_{l,i} + b + u_i, g(\eta_{1,i}, ..., \eta_{k,i}, \xi_i) \right)$$

$$= \sup_{\lambda\geq 0} -\lambda\langle \Sigma^{1/2}PH, w \rangle^2 + \inf_{u\in\mathbb{R}^n} \sum_{i=1}^n f\left( G_i\|P\Sigma^{1/2}w\|_2 + \sum_{l=1}^k \langle w, \Sigma w_l^* \rangle \eta_{l,i} + b + u_i, g(\eta_{1,i}, ..., \eta_{k,i}, \xi_i) \right) + \lambda u_i^2$$

$$= \sup_{\lambda\geq 0} -\lambda\langle \Sigma^{1/2}PH, w \rangle^2 + \sum_{i=1}^n \inf_{u\in\mathbb{R}} f\left( G_i\|P\Sigma^{1/2}w\|_2 + \sum_{l=1}^k \langle w, \Sigma w_l^* \rangle \eta_{l,i} + b + u, g(\eta_{1,i}, ..., \eta_{k,i}, \xi_i) \right) + \lambda u^2$$

$$= \sup_{\lambda\geq 0} \sum_{i=1}^n f_\lambda\left( G_i\|P\Sigma^{1/2}w\|_2 + \sum_{l=1}^k \langle w, \Sigma w_l^* \rangle \eta_{l,i} + b, g(\eta_{1,i}, ..., \eta_{k,i}, \xi_i) \right) - \lambda\langle \Sigma^{1/2}PH, w \rangle^2,$$

recalling Definition 1. For simplicity of notation, write

$$\tilde{x}_i = (\eta_{1,i}, ..., \eta_{k,i}, G_i) \sim \mathcal{N}(0, I_{k+1});$$

then the joint distribution of $(\tilde{x}_i, y_i)$ is exactly the same as the surrogate distribution $\tilde{D}$ given by (5). Moreover, we can check that

$$P\Sigma^{1/2}w = \left( I_d - \sum_{i=1}^k (\Sigma^{1/2}w_i^*)(\Sigma^{1/2}w_i^*)^T \right) \Sigma^{1/2}w$$

$$= \Sigma^{1/2}\left( I_d - \sum_{i=1}^k w_i^*(w_i^*)^T\Sigma \right) w$$

$$= \Sigma^{1/2}Qw$$

and

$$\Sigma^{1/2}PH = \Sigma^{1/2}\left( I_d - \sum_{i=1}^k (\Sigma^{1/2}w_i^*)(\Sigma^{1/2}w_i^*)^T \right) H$$

$$= \left( I_d - \sum_{i=1}^k (\Sigma w_i^*)(w_i^*)^T \right) \Sigma^{1/2}H = Q^T\Sigma^{1/2}H$$

where $Q$ is given by equation (4). Then using the definition of $\phi$ from (4), we can write

$$G_i\|P\Sigma^{1/2}w\|_2 + \sum_{l=1}^{k}\langle w, \Sigma w_l^*\rangle\eta_{l,i} = \langle\phi(w), \tilde{x}_i\rangle,$$

giving that

$$\frac{1}{n}\sum_{i=1}^{n} f_\lambda\left(G_i\|P\Sigma^{1/2}w\|_2 + \sum_{l=1}^{k}\langle w, \Sigma w_l^*\rangle\eta_{l,i} + b, g(\eta_{1,i}, \ldots, \eta_{k,i}, \xi_i)\right) = \frac{1}{n}\sum_{i=1}^{n} f_\lambda(\langle\phi(w), \tilde{x}_i\rangle + b, y_i).$$

By our assumption (6) and the observation in Lemma 4 that the joint distribution of $(\langle\phi(w), \tilde{x}\rangle, y)$ is the same as that of $(\langle w, x\rangle, y)$, we have

$$\frac{1}{n}\sum_{i=1}^{n} f_\lambda(\langle\phi(w), \tilde{x}_i\rangle + b, y_i) \geq \mathop{\mathbb{E}}_{(\tilde{x},\tilde{y})\sim\tilde{D}}[f_\lambda(\langle\phi(w), \tilde{x}\rangle + b, \tilde{y})] - \epsilon_{\lambda,\delta}(\phi(w), b)$$

$$= L_{f_\lambda}(w, b) - \epsilon_{\lambda,\delta}(\phi(w), b)$$

with probability at least $1 - \delta/4$.

In addition, noting that $\Sigma^{1/2}H \sim \mathcal{N}(0, \Sigma)$, our assumption (7) implies that with probability at least $1 - \delta/4$,

$$\langle\Sigma^{1/2}PH, w\rangle = \langle Q^T x, w\rangle = \langle Qw, x\rangle \leq C_\delta(w).$$

The proof concludes by a union bound and plugging the above estimates into the expression for $\Psi$. $\qquad\square$

Finally, we can prove our main theorem, restated here for convenience:

**Theorem 1.** *Suppose $\lambda \in \mathbb{R}^+$ satisfies that for any $\delta \in (0, 1)$, there exists a continuous function $\epsilon_{\lambda,\delta} : \mathbb{R}^{k+1} \to \mathbb{R}$ such that with probability at least $1 - \delta/4$ over independent draws $(\tilde{x}_i, \tilde{y}_i)$ from the surrogate distribution $\tilde{D}$ defined in (5), we have uniformly over all $(\tilde{w}, \tilde{b}) \in \mathbb{R}^{k+2}$ that*

$$\frac{1}{n}\sum_{i=1}^{n} f_\lambda(\langle\tilde{w}, \tilde{x}_i\rangle + \tilde{b}, \tilde{y}_i) \geq \mathop{\mathbb{E}}_{(\tilde{x},\tilde{y})\sim\tilde{D}}[f_\lambda(\langle\tilde{w}, \tilde{x}\rangle + \tilde{b}, \tilde{y})] - \epsilon_{\lambda,\delta}(\tilde{w}, \tilde{b}). \tag{6}$$

*Further, assume that for any $\delta \in (0, 1)$, there exists a continuous function $C_\delta : \mathbb{R}^d \to [0, \infty]$ such that with probability at least $1 - \delta/4$ over $x \sim \mathcal{N}(0, \Sigma)$, uniformly over all $w \in \mathbb{R}^d$,*

$$\langle Qw, x\rangle \leq C_\delta(w). \tag{7}$$

*Then it holds with probability at least $1 - \delta$ that uniformly over all $(w, b) \in \mathbb{R}^{d+1}$, we have*

$$L_{f_\lambda}(w, b) \leq \hat{L}_f(w, b) + \epsilon_{\lambda,\delta}(\phi(w), b) + \frac{\lambda C_\delta(w)^2}{n}. \tag{8}$$

*If we additionally assume that (6) holds uniformly for all $\lambda \in \mathbb{R}^+$, then (8) does as well.*

*Proof.* By Lemma 5 and Lemma 8, we have

$$\Pr\left(\sup_{(w,b)\in\mathbb{R}^{d+1}} F(w, b) - \hat{L}_f(w, b) > t \,\Big|\, \eta_1, ..., \eta_k, \xi\right) \leq 2\Pr(\Psi \geq t).$$

By the tower law and choosing

$$F(w, b) = L_{f_\lambda}(w, b) - \epsilon_{\lambda,\delta}(\phi(w), b) - \frac{\lambda C_\delta(w)^2}{n}$$

in Lemma 9, we get that

$$\Pr\left(\sup_{(w,b)\in\mathbb{R}^{d+1}} L_{f_\lambda}(w, b) - \epsilon_{\lambda,\delta}(\phi(w), b) - \frac{\lambda C_\delta(w)^2}{n} - \hat{L}_f(w, b) > 0\right) \leq \delta.$$

as desired. If assumption (6) holds uniformly over $\lambda \in \mathbb{R}^+$, then we can choose

$$F(w, b) = \sup_{\lambda\in\mathbb{R}^+} L_{f_\lambda}(w, b) - \epsilon_{\lambda,\delta}(\phi(w), b) - \frac{\lambda C_\delta(w)^2}{n}.$$

It is straightforward to check that $F$ is continuous and the same proof goes through. $\qquad\square$

**Remark 3.** Since the dimension of $\tilde{x}$ is small, we can typically expect (6) to hold for reasonable settings with a sufficiently large sample size. Note that this is our only assumption on $f, g$ and $\xi$, and this is required to avoid pathological learning problems. A useful aspect of the assumption (6) is that it only requires *one-sided concentration* of the training loss. As emphasized by many works in statistical learning theory (e.g. Lecué and Mendelson 2013; Mendelson 2014; Koltchinskii and Mendelson 2015; Mendelson 2017), lower bounds on the training loss are both more convenient to establish and hold in more generic settings than upper bounds do. In this paper, we will largely apply results from VC theory to handle the low-dimensional problem; the results we appeal to are indeed one-sided and can handle relatively heavy-tailed noise (Vapnik 1982).

# E    Proof for VC theory and Section 5

## E.1    Low-Dimensional Concentration

Recall the following definition of VC-dimension from Shalev-Shwartz and Ben-David (2014).

**Definition 4.** Let $\mathcal{H}$ be a class of functions from $\mathcal{X}$ to $\{0,1\}$ and let $C = \{c_1, ..., c_m\} \subset \mathcal{X}$. The restriction of $\mathcal{H}$ to $C$ is

$$\mathcal{H}_C = \{(h(c_1), ..., h(c_m)) : h \in \mathcal{H}\}.$$

A hypothesis class $\mathcal{H}$ *shatters* a finite set $C \subset \mathcal{X}$ if $|\mathcal{H}_C| = 2^{|C|}$. The VC-dimension of $\mathcal{H}$ is the maximal size of a set that can be shattered by $\mathcal{H}$. If $\mathcal{H}$ can shatter sets of arbitrary large size, we say $\mathcal{H}$ has infinite VC-dimension.

Also, we have the following well-known result for the class of nonhomogenous halfspaces in $\mathbb{R}^d$ (Theorem 9.3 of Shalev-Shwartz and Ben-David 2014):

**Theorem 7.** *The class $\{x \mapsto sign(\langle w, x \rangle + b) : w \in \mathbb{R}^d, b \in \mathbb{R}\}$ has VC-dimension $d + 1$.*

We will make use of the following result from Vapnik (1982):

**Theorem 2** (Special case of Assertion 4 of Vapnik (1982), Chapter 7.8; see also Theorem 7.6)**.** *Let $\mathcal{K} \subset \mathbb{R}^d$ and $\mathcal{B} \subset \mathbb{R}$. Suppose that a distribution $\mathcal{D}$ over $(x, y) \in \mathbb{R}^d \times \mathbb{R}$ satisfies that for some $\tau > 0$, it holds uniformly over all $(w, b) \in \mathcal{K} \times \mathcal{B}$ that*

$$\frac{\left(\mathbb{E}\, f(\langle w, x \rangle + b, y)^4]\right)^{1/4}}{\mathbb{E}\, f(\langle w, x \rangle + b, y)} \leq \tau. \tag{9}$$

*Also suppose the class of functions $\{(x, y) \mapsto \mathbb{1}\{f(\langle w, x \rangle + b, y) > t\} : w \in \mathcal{K}, b \in \mathcal{B}, t \in \mathbb{R}\}$ has VC-dimension at most $h$. Then for any $n > h$, with probability at least $1 - \delta$ over the choice of $((x_1, y_1), \ldots, (x_n, y_n)) \sim \mathcal{D}^n$, it holds uniformly over all $w \in \mathcal{K}, b \in \mathcal{B}$ that*

$$\frac{1}{n} \sum_{i=1}^n f(\langle w, x_i \rangle + b, y_i) \geq \left(1 - 8\tau \sqrt{\frac{h(\log(2n/h) + 1) + \log(12/\delta)}{n}}\right) \mathbb{E}\, f(\langle w, x \rangle + b, y).$$

Combining with theorem 1, we obtain the following corollary.

**Corollary 1.** *Under the model assumptions (2), suppose that $C_\delta$ satisfies condition (7). Also suppose that for some fixed $\lambda \geq 0$, $\mathcal{K} \subseteq \mathbb{R}^d$, and $\mathcal{B} \subseteq \mathbb{R}$, the surrogate distribution $\tilde{\mathcal{D}}$ satisfies assumption (9) under $f_\lambda$ uniformly over $\phi(\mathcal{K}) \times \mathcal{B}$, and that the class $\{(x, y) \mapsto \mathbb{1}\{f_\lambda(\langle \tilde{w}, \phi(x) \rangle + \tilde{b}, y) > t\} : \tilde{w} \in \phi(\mathcal{K}), \tilde{b} \in \mathcal{B}, t \in \mathbb{R}\}$ has VC-dimension at most $h$. Then with probability at least $1 - \delta$, uniformly over all $(w, b) \in \mathcal{K} \times \mathcal{B}$*

$$\left(1 - 8\tau \sqrt{\frac{h(\log(2n/h) + 1) + \log(48/\delta)}{n}}\right) L_{f_\lambda}(w, b) \leq \hat{L}_f(w, b) + \frac{\lambda C_\delta(w)^2}{n}.$$

*Furthermore, if assumption (9) holds uniformly for all $\{f_\lambda : \lambda \in \mathbb{R}_{\geq 0}\}$ and the class $\{(x, y) \mapsto \mathbb{1}\{f_\lambda(\langle \tilde{w}, \phi(x) \rangle + \tilde{b}, y) > t\} : (\tilde{w}, \tilde{b}) \in \phi(\mathcal{K}) \times \mathcal{B}, t \in \mathbb{R}, \lambda \in \mathbb{R}_{\geq 0}\}$ has VC-dimension at most $h$, then the same conclusion holds uniformly over $\lambda$.*

*Proof.* By theorem 2, we can take

$$\epsilon_{\lambda,\delta}(\tilde{w}, \tilde{b}) = \begin{cases} 8\tau\sqrt{\frac{h(\log(2n/h)+1)+\log(48/\delta)}{n}} \; \mathbb{E}_{(\tilde{x},\tilde{y})\sim\tilde{\mathcal{D}}}[f_\lambda(\langle\tilde{w},\tilde{x}\rangle + \tilde{b}, \tilde{y})] & \text{if } (\tilde{w}, \tilde{b}) \in \phi(\mathcal{K}) \times \mathcal{B} \\ \infty & \text{otherwise} \end{cases}$$

and the desired conclusion follows by the observation that

$$\mathbb{E}_{(\tilde{x},\tilde{y})\sim\tilde{\mathcal{D}}}[f_\lambda(\langle\phi(w),\tilde{x}\rangle + b, \tilde{y})] = L_{f_\lambda}(w, b).$$

The last conclusion (uniformity over $\lambda$) follows by going through the proof of Theorem 2, since it is based on reduction to uniform control of indicators. □

### E.2 Linear Regression

First, we provide a VC-dimension bound for the square loss class.

**Lemma 10.** *Suppose $f$ is the square loss, then the VC-dimension of the class*

$$\{(x, y) \mapsto \mathbb{1}\{f_\lambda(\langle\tilde{w}, \phi(x)\rangle + \tilde{b}, y) > t\} : (\tilde{w}, \tilde{b}) \in \mathbb{R}^{k+2}, t \in \mathbb{R}, \lambda \in \mathbb{R}_{\geq 0}\}$$

*is $O(k)$.*

*Proof.* Since the square loss is non-negative, we only need to consider $t \geq 0$. Recall that $f_\lambda = \frac{\lambda}{1+\lambda}f$ for the square loss and so

$$f_\lambda(\langle\tilde{w}, \phi(x)\rangle + \tilde{b}, y) > t \iff (\langle\tilde{w}, \phi(x)\rangle + \tilde{b} - y)^2 > \frac{(1+\lambda)t}{\lambda}$$

which happens if

$$\left\langle \begin{pmatrix} \tilde{w} \\ -1 \end{pmatrix}, \begin{pmatrix} \phi(x) \\ y \end{pmatrix} \right\rangle + \left(\tilde{b} - \sqrt{\frac{(1+\lambda)t}{\lambda}}\right) > 0 \quad \text{or} \quad \left\langle \begin{pmatrix} -\tilde{w} \\ 1 \end{pmatrix}, \begin{pmatrix} \phi(x) \\ y \end{pmatrix} \right\rangle - \left(\tilde{b} + \sqrt{\frac{(1+\lambda)t}{\lambda}}\right) > 0.$$

In particular, if this concept class can shatter $m$ points, so can the class of the union of two non-homogenous halfspaces in $\mathbb{R}^{k+2}$. The desired conclusion follows by the well-known fact that the VC-dimension of the union of two halfspaces is $O(k)$. For example, by combining Theorem 7 with Lemma 3.23 of Blumer et al. (1989), the VC-dimension cannot be larger than $4\log 6 \cdot (k + 3)$. □

Specializing our generalization theory to the square loss, we have:

**Corollary 2.** *Suppose $f$ is the square loss and the surrogate distribution $\tilde{\mathcal{D}}$ satisfies assumption* (9) *uniformly over $(w, b) \in \mathbb{R}^{k+1}$, then with probability at least $1 - \delta$, uniformly over all $w, b$ we have*

$$\left(1 - 8\tau\sqrt{\frac{k(\log(2n/k)+1)+\log(48/\delta)}{n}}\right) L_f(w, b) \leq \left(\sqrt{\hat{L}_f(w, b)} + C_\delta(w)/\sqrt{n}\right)^2.$$

*Proof.* Note that if condition (9) holds under $f$, then it also holds under all $\{f_\lambda : \lambda \geq 0\}$ because $f_\lambda = \frac{\lambda}{1+\lambda}f$. Moreover, we check the assumption on VC-dimension of Corollary 1 in Lemma 10. From this, we get uniformly over $\lambda, w, b$ that

$$\frac{\lambda}{1+\lambda}\left(1 - 8\tau\sqrt{\frac{k(\log(2n/k)+1)+\log(48/\delta)}{n}}\right) L_f(w, b) \leq \hat{L}_f(w, b) + \frac{\lambda C_\delta(w)^2}{n}.$$

Multiplying through by $(1 + \lambda)/\lambda$, we can rewrites the above as

$$\left(1 - 8\tau\sqrt{\frac{k(\log(2n/k)+1)+\log(48/\delta)}{n}}\right) L_f(w, b) \leq \left(1 + \frac{1}{\lambda}\right)\hat{L}_f(w, b) + (1 + \lambda)\frac{C_\delta(w)^2}{n}$$

and optimizing over $\lambda$ gives

$$\left(1 - 8\tau\sqrt{\frac{k(\log(2n/k)+1)+\log(48/\delta)}{n}}\right)L_f(w,b)$$

$$\leq \hat{L}_f(w,b) + \frac{C_\delta(w)^2}{n} + \inf_{\lambda \geq 0}\frac{1}{\lambda}\hat{L}_f(w,b) + \lambda\frac{C_\delta(w)^2}{n}$$

$$= \hat{L}_f(w,b) + \frac{C_\delta(w)^2}{n} + 2\sqrt{\hat{L}_f(w,b)\frac{C_\delta(w)^2}{n}} = \left(\sqrt{\hat{L}_f(w,b)} + C_\delta(w)/\sqrt{n}\right)^2. \qquad \square$$

Finally, as an illustrative example, we consider the misspecified model mentioned in the main text where the true regression function is a polynomial. In this case, we show explicitly how to get an expression for $\tau$ in (9) using Gaussian hypercontractivity. The following theorem is the Gaussian space analogue of Theorem 9.21 in O'Donnell (2014) and can be proved using the same argument by Theorem 11.23 and replacing the Fourier basis on $\{-1,1\}^n$ with the Hermite polynomials on $\mathbb{R}^n$.

**Theorem 8** (O'Donnell 2014). *Let* $f : \mathbb{R}^d \to \mathbb{R}$ *be a polynomial of degree at most* $k$. *Then for any* $q \geq 2$, *it holds that*

$$\mathbb{E}_{z\sim\mathcal{N}(0,I_d)}[|f(z)|^q]^{1/q} \leq (q-1)^{k/2}\mathbb{E}_{z\sim\mathcal{N}(0,I_d)}[|f(z)|^2]^{1/2}. \tag{39}$$

**Theorem 9.** *Suppose that in* (2), *we have*

$$y = m(\eta_1, ..., \eta_k) + s(\eta_1, ..., \eta_k) \cdot \xi$$

*where* $m, s$ *are both polynomials of degree at most* $l$ *and* $\xi$ *has finite eighth moment, then*

$$\frac{\mathbb{E}[(\langle w,x\rangle + b - y)^8]^{1/8}}{\mathbb{E}[(\langle w,x\rangle + b - y)^2]^{1/2}} \leq \sqrt{2}\cdot\sqrt{7}^l\left(\frac{E[\xi^8]^{1/8}}{E[\xi^2]^{1/2}}\right). \tag{40}$$

*Proof.* By triangular inequality in the $\ell_p$ space and independence between $x$ and $\xi$

$$\mathbb{E}[(\langle w,x\rangle + b - y)^8]^{1/8} \leq \mathbb{E}[(\langle w,x\rangle + b - m(\eta_1,...,\eta_k))^8]^{1/8} + \mathbb{E}[(s(\eta_1,...,\eta_k)\cdot\xi)^8]^{1/8}$$

$$= \mathbb{E}[(\langle w,x\rangle + b - m(\eta_1,...,\eta_k))^8]^{1/8} + \mathbb{E}[s(\eta_1,...,\eta_k)^8]^{1/8}\cdot\mathbb{E}[\xi^8]^{1/8}$$

Since $\langle w,x\rangle, \eta_1,...,\eta_k$ are jointly Gaussian, we can apply Theorem 8 and upper bound the above by

$$\sqrt{7}^l\left(\mathbb{E}[(\langle w,x\rangle + b - m(\eta_1,...,\eta_k))^2]^{1/2} + \mathbb{E}[s(\eta_1,...,\eta_k)^2]^{1/2}\cdot\mathbb{E}[\xi^8]^{1/8}\right)$$

$$\leq \sqrt{7}^l\left(\frac{E[\xi^8]^{1/8}}{E[\xi^2]^{1/2}}\right)\left(\mathbb{E}[(\langle w,x\rangle + b - m(\eta_1,...,\eta_k))^2]^{1/2} + \mathbb{E}[s(\eta_1,...,\eta_k)^2]^{1/2}\cdot\mathbb{E}[\xi^2]^{1/2}\right)$$

$$\leq \sqrt{7}^l\left(\frac{E[\xi^8]^{1/8}}{E[\xi^2]^{1/2}}\right)\sqrt{2}\cdot\sqrt{\mathbb{E}[(\langle w,x\rangle + b - m(\eta_1,...,\eta_k))^2] + \mathbb{E}[s(\eta_1,...,\eta_k)^2]\cdot\mathbb{E}[\xi^2]}$$

where we use $E[\xi^8]^{1/8} \geq E[\xi^2]^{1/2}$ in the second inequality and $\sqrt{a} + \sqrt{b} \leq \sqrt{2(a+b)}$ in the last inequality. The desired conclusion follows by observing

$$\mathbb{E}[(\langle w,x\rangle + b - y)^2] = \mathbb{E}[(\langle w,x\rangle + b - m(\eta_1,...,\eta_k))^2] + \mathbb{E}[s(\eta_1,...,\eta_k)^2]\cdot\mathbb{E}[\xi^2]$$

because $x$ and $\xi$ are independent. $\qquad \square$

**Remark 4.** The assumption that $\xi$ has finite eighth moment can be significantly relaxed because there is a version of Theorem 2 in Vapnik (1982) that replaces the exponent of 4 by $1 + \epsilon$. However, allowing heavier tails of $\xi$ comes at the cost of a larger constant in front of $\tau$ or a slower convergence rate with respect to $n$ in the low-dimensional concentration term.

### E.3  Linear Classification

**Lemma 11.** *Suppose* $f$ *is the squared hinge loss, then the VC-dimension of the class*

$$\{(x,y) \mapsto \mathbb{1}\{f_\lambda(\langle\tilde{w},\phi(x)\rangle + \tilde{b}, y) > t\} : (\tilde{w},\tilde{b}) \in \mathbb{R}^{k+2}, t \in \mathbb{R}, \lambda \in \mathbb{R}_{\geq 0}\}$$

*is no larger than* $k + 3$.

*Proof.* Since the squared hinge loss is non-negative, we only need to consider $t \geq 0$. Recall that $f_\lambda = \frac{\lambda}{1+\lambda} f$ and so

$$f_\lambda(\langle \tilde{w}, \phi(x)\rangle + \tilde{b}, y) > t \iff (1 - y(\langle \tilde{w}, \phi(x)\rangle + \tilde{b}))_+^2 > \frac{(1+\lambda)t}{\lambda}$$

$$\iff 1 - y(\langle \tilde{w}, \phi(x)\rangle + \tilde{b}) > \sqrt{\frac{(1+\lambda)t}{\lambda}}$$

$$\iff \left\langle \begin{pmatrix} \tilde{w} \\ \tilde{b} \end{pmatrix}, \begin{pmatrix} -y\phi(x) \\ -y \end{pmatrix} \right\rangle + \left( 1 - \sqrt{\frac{(1+\lambda)t}{\lambda}} \right) > 0.$$

In particular, if this class can shatter $m$ points, so can the class of nonhomogenous halfspaces in $\mathbb{R}^{k+2}$. But theorem 7 shows that it cannot shatter more than $k+4$ points, and so the VC-dimension cannot be larger than $k+3$. □

By the same proof as Corollary 2, we have

**Corollary 4.** *Suppose $f$ is the squared hinge loss and the surrogate distribution $\tilde{\mathcal{D}}$ satisfies assumption (9) uniformly over $(w, b) \in \mathbb{R}^{k+1}$, then with probability at least $1 - \delta$, uniformly over all $w, b$ we have*

$$\left( 1 - 8\tau \sqrt{\frac{k(\log(2n/k) + 1) + \log(48/\delta)}{n}} \right) L_f(w, b) \leq \left( \sqrt{\hat{L}_f(w, b)} + C_\delta(w)/\sqrt{n} \right)^2.$$

For illustration, we show how to check hypercontractivity (9) under some example generative assumptions on $y$. In the first and simpler example, suppose that there is an arbitrary constant $\eta > 0$ such that

$$\min\{\Pr(y = 1 \mid x), \Pr(y = -1 \mid x)\} \geq \eta$$

almost surely. This assumption is satisfied, for example, if the data is generated by an arbitrary function of $\eta_1, \dots, \eta_k$ combined with Random Classification Noise (see e.g. Blum et al. (2003)), i.e. the label is flipped with some probability. Then if $\hat{y} = \langle w, x\rangle + b$ is the prediction, we have

$$\mathbb{E}\max(0, 1 - y\hat{y})^2 \geq \eta \, \mathbb{E}(1 + |\hat{y}|)^2 \geq \eta(1 + \mathbb{E}[\hat{y}^2]),$$

and on the other hand we always have

$$\mathbb{E}\max(0, 1 - y\hat{y})^8 \leq \mathbb{E}(1 + |\hat{y}|)^8 \leq 2^8(1 + \mathbb{E}[\hat{y}^8]) \leq 2^{16}(1 + \mathbb{E}[\hat{y}^2]^4) \leq 2^{16}(1 + \mathbb{E}[\hat{y}^2])^4$$

where the second-to-last inequality follows from the fact that $\hat{y}$ is marginally Gaussian and using standard formula for the moments of a Gaussian. It follows that

$$\frac{\mathbb{E}[\max(0, 1 - y\hat{y})^8]^{1/8}}{\mathbb{E}[\max(0, 1 - y\hat{y})^2]^{1/2}} \leq \frac{4}{\sqrt{\eta}}$$

which verifies (9) in this setting.

We now consider a more general situation and show that if there is a *non-negligible* portion of $x$'s such that that $y$ is noisy, hypercontractivity is still guaranteed to hold. Let $A_\eta$ be the event that $\min\{\Pr(y = 1 \mid x), \Pr(y = -1 \mid x)\} \geq \eta$. Then

$$\mathbb{E}\max(0, 1 - y\hat{y})^2 \geq \mathbb{E}[\mathbb{1}(A_\eta)\max(0, 1 - y\hat{y})^2] \geq \eta \, \mathbb{E}[\mathbb{1}(A_\eta)(1 + |\hat{y}|)^2]$$

$$\geq \eta Q(\Pr(A_\eta)) \, \mathbb{E}[(1 + |\hat{y}|)^2]$$

where $Q$ is defined below. In the last step, we considered the worst case event $A_\eta$ for given $\Pr(A_\eta)$, which corresponds to chopping the tails off of $\hat{y}$; considering this example, we see the inequality holds where where $Q : (0, 1] \to (0, 1]$ is an explicit function

$$Q(p) := \min \left\{ \frac{\int_{-z_p}^{z_p} |x|e^{-x^2/2}dx}{2}, \frac{\int_{-z_p}^{z_p} x^2 e^{-x^2/2}dx}{\sqrt{2\pi}} \right\} \tag{41}$$

and $z_p$ is defined such that $\Pr_{g \sim N(0,1)}[|g| > z_p] = p$. Repeating the argument above yields the following result:

**Theorem 10.** *Suppose that under* (2), *there exists* $\eta > 0$ *such that* $p_\eta := \Pr(\min\{\Pr(y = 1 \mid x), \Pr(y = -1 \mid x)\} \geq \eta) > 0$. *Then for any* $w, b$ *we have that for* $\hat{y} = \langle w, x \rangle + b$,

$$\frac{\mathbb{E}[\max(0, 1 - y\hat{y})^8]^{1/8}}{\mathbb{E}[\max(0, 1 - y\hat{y})^2]^{1/2}} \leq \frac{4}{\sqrt{\eta Q(p_\eta)}}$$

For another example, if $y$ follows a logistic regression model $\mathbb{E}[y \mid x] = \tanh(\beta w_1^* \cdot x)$ with normalization $\langle w_1^*, \Sigma w_1^* \rangle = 1$, then by Theorem 10 with e.g. $\eta = 1/2$, we verify (9) with $\tau$ a constant depending only on $\beta$. The result also holds for more general models like $\mathbb{E}[y \mid x] = \tanh(f(\eta_1, \ldots, \eta_k))$ as long as $f$ is not always very large.

### E.3.1 Squared Hinge Loss and Zero-One Loss

In the previous section, we discussed how our generalization bound controls the population squared hinge loss, one of the standard losses used in classification. In the context of benign overfitting, this is the canonical loss to look at because it is implicitly optimized by the max-margin predictor, also known as Hard SVM (see Theorem 3, as well as Shamir 2022).

On the other hand, it is also very natural to look at the zero-one loss of a classifier. In general, the squared hinge loss and zero-one loss are different loss functions, and their population global optima will differ. Nevertheless, in many cases the minimizer of the squared hinge loss will also have good zero-one loss. We discuss a few situations where this occurs below.

**General Bound on Zero-One Loss from Margin Loss.**    First of all, the following bound comparing the zero-one loss and margin loss always holds — the analogous bound for the (non-squared) hinge loss is very standard and the same argument applies to squared hinge loss:

**Theorem 11** (Classical, see e.g. Shalev-Shwartz and Ben-David 2014). *For any* $w, b$, *we have that*

$$\Pr(sgn(\langle w, x \rangle + b) \neq y) \leq L_f(w, b)$$

*where* $f$ *is the squared hinge loss.*

*Proof.* Observe that if $sgn(\hat{y}) \neq y$, then

$$\hat{f}(\hat{y}, y) = \max(0, 1 - y\hat{y})^2 \geq 1.$$

Taking the expectation over $\hat{y} = \langle w, x \rangle + b$ and $y$ gives the result. $\qquad\square$

In particular, when we are in the *realizable* setting, where there exists a halfspace with positive margin with zero-one loss equal to zero, then as long as we can find a near-minimizer of the squared hinge test loss, Theorem 11 will guarantee near-optimal zero-one loss.

**Improved Comparison in a Noisy Setting.**    It is clear from the proof that Theorem 11, while very general, is not always tight. For example, T. Zhang (2004b) and Bartlett et al. (2006) give improved bounds which are very useful in the case that the minimizer of the squared hinge loss over *all measurable functions* is contained in the class. This includes the realizable case considered above; on the other hand, it will not generally be the case that the class of linear functions includes the minimizer over all measurable functions when there is label noise. We now describe a noisy situation where minimizing the squared hinge test loss will also minimize the zero-one test loss.

For simplicity, we consider the special case of our general setup where the response $y$ is binary (classification) and also $k = 1$, so it follows a *single-index* model, or equivalently

$$y = g(\eta_1, \xi) \tag{42}$$

where $\eta_1 = \langle w_1^*, x \rangle$ and $\xi$ is independent of the covariate $x$. Note that in the following discussion, we use the additional covariance splitting notation introduced in Appendix C.

The following lemma shows that any near-minimizer of the loss $L_f$ will have $r(w) = \|w^\perp\|_\Sigma \approx 0$, i.e. such $w$ will be essentially along the direction of the ground truth $w_1^*$.

**Lemma 12.** *Suppose $(x, y) \sim \mathcal{D}$ follows a single-index model* (42)*, and suppose the loss functional $f(\hat{y}, y)$ is of the form*

$$f(\hat{y}, y) = \ell(y\hat{y}) \tag{43}$$

*for some convex function $\ell$. Then for any $w, b$ we have*

$$L_f(w, b) - L_f(w^{\|}, b) = \mathbb{E}[\ell(y(\langle w^{\|}, x \rangle + b) + g\|w^{\perp}\|_{\Sigma})] - \mathbb{E}[\ell(y(\langle w^{\|}, x \rangle + b))]$$

*where $g$ is a standard Gaussian random variable independent of everything else, and so by Jensen's inequality, we have*

$$L_f(w, b) \geq L_f(w^{\|}, b).$$

*Furthermore, suppose $\ell$ is not the constant function, then the equality holds iff $\|w^{\perp}\|_{\Sigma} = 0$.*

*Proof.* Let $w^{\|} = (I - Q)w$ and $w^{\perp} = Qw$. Expanding the definition, we have

$$L_f(w, b) - L_f(w^{\|}, b) = \mathbb{E}[\ell(y(\langle w^{\|}, x \rangle + \langle w^{\perp}, x \rangle + b))] - \mathbb{E}[\ell(y(\langle w^{\|}, x \rangle + b))].$$

By the definition of $Q$, $\langle w_1^*, \Sigma w^{\perp} \rangle = 0$ and so $\langle w^{\perp}, x \rangle$ is independent of $\langle w_1^*, x \rangle$ and $\langle w^{\|}, x \rangle$. Hence, it also independent of $y$ due to (42). Let $g \sim \mathcal{N}(0, 1)$ be a standard Gaussian random variable independent of $x$, then it follows that

$$L_f(w, b) - L_f(w^{\|}, b) = \mathbb{E}[\ell(y(\langle w^{\|}, x \rangle + b + g\|w^{\perp}\|_{\Sigma}))] - \mathbb{E}[\ell(y(\langle w^{\|}, x \rangle + b))].$$

Moreover, since $y$ is $\{\pm 1\}$ valued and independent of $g$, $gy$ is equal in law to $g$ conditioned on $y$ and

$$L_f(w, b) - L_f(w^{\|}, b) = \mathbb{E}[\ell(y(\langle w^{\|}, x \rangle + b) + g\|w^{\perp}\|_{\Sigma})] - \mathbb{E}[\ell(y(\langle w^{\|}, x \rangle + b))].$$

The nonnegativity of this expression now follows from Jensen's inequality, since $\ell$ is assumed to be convex, and if $\ell$ is assumed to be non-constant then the equality holds iff $\|w^{\perp}\|_{\Sigma} = 0$. □

Since $\ell$ is only assumed to be convex, this includes the logistic loss, squared hinge loss, hinge loss, and squared loss (in the classification setting). The previous lemma directly implies that $\|w^{\perp}\|_{\Sigma} \to 0$ for any $w$ which approaches the optimal squared hinge loss. This means that $w$ will align with the true direction $w_1^*$; we now show that in the zero bias case, this leads to the near-optima of the squared hinge loss having optimal zero-one loss. Note that this may not be the case in more general settings, as even if $w$ is aligned with $w_1^*$, the relative size of $w$ and the bias $b$ also needs to match the ground truth in order to truly minimize the zero-one loss.

**Theorem 12.** *Suppose that $f(\hat{y}, y)$ is the squared hinge loss, so $\ell(z) = \max(0, 1 - z)^2$ in the notation of* (43)*. Suppose with probability 1, it holds that*

$$\eta_1 \cdot \mathbb{E}_{\xi}[g(\eta_1, \xi)] > 0. \tag{44}$$

*Then every global optima of the squared hinge loss with zero bias term, $L_f(w, 0)$, is of the form $w = \alpha w_1^*$ with $\alpha > 0$. Furthermore, for any $w$ we have the inequality*

$$L_f(w, 0) \geq L_f(w^{\|}, 0) \geq \inf_w L_f(w, 0)$$

*and so we have that for any sequence $w_n$ that $L_f(w_n, 0) \to \inf_w L_f(w, 0)$, it holds that*

$$\Pr[sgn(\langle w_n, x \rangle) \neq y] \to \Pr[sgn(\langle w_1^*, x \rangle) \neq y].$$

*Proof.* By Lemma 12, it suffices to consider $w$ along the direction $w_1^*$ and show that the optimal $w$ cannot point in the direction opposite to $w_1^*$. To this end, observe that

$$\frac{\partial \ell}{\partial z} = 2(z - 1)\mathbb{1}\{z \leq 1\}$$

and by the chain rule, using that $L_f(\alpha w_1^*, 0) = \mathbb{E}[\ell(y\alpha\langle w_1^*, x \rangle)]$, we have

$$\frac{\partial}{\partial \alpha} L_f(\alpha w_1^*, 0) = 2\mathbb{E}[(y\alpha\langle w_1^*, x \rangle - 1)\mathbb{1}\{y\alpha\langle w_1^*, x \rangle \leq 1\}y\langle w_1^*, x \rangle].$$

Evaluating this at $\alpha = 0$ gives

$$\frac{\partial}{\partial \alpha} L_f(\alpha w_1^*, 0)\Big|_{\alpha=0} = -2\mathbb{E}[y\langle w_1^*, x \rangle].$$

Applying the law of total expectation, we have shown

$$\frac{\partial}{\partial \alpha} L_f(\alpha w_1^*, 0)\Big|_{\alpha=0} = -2\,\mathbb{E}[\mathbb{E}[y \mid x]\langle w_1^*, x\rangle] < 0$$

under the assumption of the Lemma. It is easy to see that $L_f(\alpha w_1^*, 0)$ is convex in $\alpha$, which concludes the proof of the first part. We can also have final conclusion because $L_f(w_n, 0) - L_f(w_n^{\|}, 0) \to 0$ implies $\|w_n^\perp\|_\Sigma \to 0$ by Lemma 12, and $\liminf_{n\to\infty}\langle w_n, w_1^*\rangle > 0$ by the first part of the theorem. $\qquad\square$

The condition (44) is mild and easy to check for standard generative models like logistic regression, where we have that $\mathbb{E}[y \mid x] = \tanh(\beta\langle w_1^*, x\rangle)$ and so $\mathbb{E}_\xi[\eta_1 y \mid x] > 0$ by Chebyshev's correlation inequality (using that $\tanh$ is an increasing function). Finally, we note that the last conclusion of Theorem 12 means that near-minimizers of the test loss $L_f(w, 0)$ are near-minimizers of the zero one loss, under the further well-specified assumption that $sgn(\langle w_1^*, x\rangle)$ achieves the Bayes-optimal classification rate (i.e. minimum of zero-one loss over all functions).

### E.4  Sharpness of Improved Lipschitz Contraction

In this section, we show that the Lipschitz contraction bound (11) for 1-Lipschitz loss functions $f$,

$$(1 - o(1))L_f(w) \le \hat{L}_f(w) + \sqrt{\frac{C_\delta(w)^2}{n}}$$

has sharp constants in the case of the $L_1$ loss $f(\hat{y}, y) := |y - \hat{y}|$. This shows that the only way to tighten the bound further is to consider one with a different functional form (e.g. the Moreau envelope bound with the Huber test loss). In particular, the Moreau envelope version of the bound is significantly more useful when looking at interpolators.

**Data Distribution.**  We will show tightness in the setting of the junk features model. Let's consider

$$x \sim \mathcal{N}(0, \Sigma), \quad y \sim \mathcal{N}(0, \sigma^2)$$

where the response $y$ is independent of the covariate $x$ and the covariance $\Sigma$ is given by

$$\Sigma = \begin{bmatrix} 1 & 0 \\ 0 & \frac{\lambda_n}{d_J} I_{d_J} \end{bmatrix}.$$

In addition, following Zhou et al. (2020), we consider the asymptotics where first, for fixed $n$, we take $d_J \to \infty$, and then we take $n \to \infty$ with $\lambda_n = \sqrt{n}$.

**Predictor.**  The $w$ which demonstrates tightness is of the form

$$w = (r, w_{\sim 1})$$

where $r > 0$ is a parameter and $w_{\sim 1}$ is constructed based on the training data $(x_i, y_i)_{i=1}^n$ to minimize $\|w_{\sim 1}\|_2$ given the constraint

$$\langle w_{\sim 1}, x_{i, \sim 1}\rangle = \sigma \cdot \text{sgn}(y_i - r x_{i,1}).$$

**Tightness.**  Since $w_{\sim 1}$ plays no role in a new prediction[4], we have

$$\lim_{d_J \to \infty} L_f(w) = \mathbb{E}\,|y - r x_1|$$

and as $n \to \infty$

$$\hat{L}_f(w) = \frac{1}{n}\sum_{i=1}^n |y_i - r x_{i,1} - \langle w_{\sim 1}, x_{i,\sim 1}\rangle| = \frac{1}{n}\sum_{i=1}^n |y_i - r x_{i,1} - \sigma \cdot \text{sgn}(y_i - r x_{i,1})|$$

$$= \frac{1}{n}\sum_{i=1}^n \big||y_i - r x_{i,1}| - \sigma\big| \approx \mathbb{E}\,|y - r x_1| - \sigma$$

---

[4]This is because $w_{\sim 1}$ lies in the span of $x_{i,\sim 1}$, but a new sample from $x_{\sim 1}$ will be almost surely orthogonal to all $x_{i,\sim 1}$ in the training set as $d_J \to \infty$.

because $\Pr(|y - rx_1| < \sigma) \to 0$ as $r \to \infty$ and $\frac{1}{n}\sum_{i=1}^{n}|y_i - rx_{i,1}| \to \mathbb{E}|y - rx_1|$ by the law of large numbers. Therefore, the actual generalization gap for $w$ will be

$$\lim_{r\to\infty}\lim_{n\to\infty}\lim_{d_J\to\infty} L_f(w) - \hat{L}_f(w) = \sigma. \tag{45}$$

On the other hand, following the analysis from Zhou et al. (2020, Appendix B), we have[5]

$$\lim_{d_J\to\infty}\|w\|_2^2 = r^2 + \frac{\sigma^2 n}{\lambda_n},$$

and by taking $C_\delta(w)$ as in Lemma 1 and using $\operatorname{Tr}\Sigma = 1 + \lambda_n$, the bound (11) gives

$$L_f(w) - \hat{L}_f(w) \le \|w\|_2\sqrt{\frac{1 + \lambda_n}{n}}. \tag{46}$$

Since $\|w\|_2 \approx \sigma\sqrt{\frac{n}{\lambda_n}}$ and $\sqrt{\frac{1+\lambda_n}{n}} \approx \sqrt{\frac{\lambda_n}{n}}$, the value of the bound converges to $\sigma$ as $n \to \infty$.

# F   Proof of Theorem 4

**Theorem 4.** *Let $\mathcal{K}, \mathcal{B}$ be bounded convex sets, and let $f(\hat{y}, y)$ be convex in $\hat{y}$. Suppose that $\tau$ is such that with probability at least $1 - \delta$, for $(\tilde{x}, \tilde{y})_{i=1}^{n}$ sampled i.i.d. from $\tilde{\mathcal{D}}$ we have*

$$\min_{\tilde{w}\in\phi(\mathcal{K}),b_0\in\mathcal{B}}\max_{\lambda\ge 0}\left[\frac{1}{n}\sum_{i=1}^{n}f_\lambda(\langle\tilde{w},\tilde{x}\rangle + b_0, y_i) - \frac{\lambda}{n}\max_{w_0\in\phi^{-1}(\tilde{w})\cap\mathcal{K}}\langle x, Qw_0\rangle^2\right] \le \tau. \tag{17}$$

*Then with probability at least $1 - 2\delta$, $\min_{w\in\mathcal{K},b\in\mathcal{B}}\hat{L}_f(w, b) \le \tau$.*

*Proof.* We can write the training error as a minmax problem by introducing a variable $\hat{y} = Xw$ and using Lagrange multipliers to write the minimum of the training loss (Primary Optimization) as

$$\Phi := \min_{w\in\mathcal{K},b_0\in\mathcal{B},\hat{y}}\max_{\lambda}\frac{1}{n}\sum_{i=1}^{n}f(\hat{y}_i, y_i) + \langle\lambda, \hat{y} - X^\|w^\| - X^\perp w^\perp - b_0\rangle.$$

Note that here we are using the additional covariance splitting notation introduced in Appendix C, and we interpret the subtraction of $b_0$ as entrywise (equivalently, as subtracting the vector $b_0\vec{1}$).

Similarly, define the Auxiliary Optimization problem (which will be related to the Primary Optimization below) as a random variable depending on independent random vectors $g \sim \mathcal{N}(0, I_n)$ and $h \sim \mathcal{N}(0, I_d)$ as

$$\Psi := \min_{w\in\mathcal{K},b_0\in\mathcal{B},\hat{y}}\max_{\lambda}\frac{1}{n}\sum_{i=1}^{n}f(\hat{y}_i, y_i) + \langle\lambda, \hat{y} - X^\|w^\| - b_0\rangle - \langle\lambda, g\rangle\|w^\perp\|_{\Sigma^\perp} - \langle h, Q\Sigma^{1/2}w^\perp\rangle\|\lambda\|$$

and truncated versions of both problems

$$\Phi_s := \min_{w\in\mathcal{K},b_0\in\mathcal{B},\hat{y}}\max_{\|\lambda\|\le s}\frac{1}{n}\sum_{i=1}^{n}f(\hat{y}_i, y_i) + \langle\lambda, \hat{y} - X^\|w^\| - X^\perp w^\perp - b_0\rangle$$

and

$$\Psi_s := \min_{w\in\mathcal{K},b_0\in\mathcal{B},\hat{y}}\max_{\|\lambda\|\le s}\frac{1}{n}\sum_{i=1}^{n}f(\hat{y}_i, y_i) + \langle\lambda, \hat{y} - X^\|w^\| - b_0\rangle - \langle\lambda, g\rangle\|w^\perp\|_{\Sigma^\perp} - \langle h, Q\Sigma^{1/2}w^\perp\rangle\|\lambda\|$$

By definition, we have $\Psi_s \le \Psi$ and by applying Lemma 7 and Theorem 6 we have that $\Pr(\Phi > t) \le \lim_{s\to\infty}\Pr(\Phi_s > t) \le 2\lim_{s\to\infty}\Pr(\Psi_s > t) \le 2\Pr(\Psi > t)$.

---

[5]Again, this is because the vectors $x_{1,\sim 1}, \ldots, x_{n,\sim 1}$ will asymptotically be orthogonal to each other and have norm $\sqrt{\lambda_n}$ and we use each of them to fit a label of size $\sigma$.

It remains to prove a high probability upper bound on the Auxiliary Optimization $\Psi$. Observe that we can rewrite

$$\Psi = \min_{w \in \mathcal{K}, b_0 \in \mathcal{B}, \hat{y}} \max_{\lambda} \frac{1}{n} \sum_{i=1}^{n} f(y_i, \hat{y}_i) + \langle \lambda, \hat{y} - X^{\|} w^{\|} - g\|w^{\perp}\|_{\Sigma^{\perp}} - b_0 \rangle - \langle h, (\Sigma^{\perp})^{1/2} w^{\perp} \rangle \|\lambda\|$$

and then solving the optimization over $\lambda$ gives

$$\Psi = \min_{w \in \mathcal{K}, b_0 \in \mathcal{B}, \hat{y}: \|\hat{y} - X^{\|} w^{\|} - g\|w^{\perp}\|_{\Sigma^{\perp}} - b_0\| \leq \langle (\Sigma^{\perp})^{1/2} h, w^{\perp} \rangle} \frac{1}{n} \sum_{i=1}^{n} f(y_i, \hat{y}_i)$$

$$= \min_{w \in \mathcal{K}, b_0 \in \mathcal{B}, \hat{y}: \|\hat{y} - X^{\|} w^{\|} - g\|w^{\perp}\|_{\Sigma^{\perp}} - b_0\| \leq |\langle (\Sigma^{\perp})^{1/2} h, w^{\perp} \rangle|} \frac{1}{n} \sum_{i=1}^{n} f(y_i, \hat{y}_i)$$

where the last equality is by observing that if $\langle \Sigma^{\perp} h, w^{\perp} \rangle$, we can flip the sign of $w^{\perp}$ to get a feasible point of the constraint with the absolute value and with the same objective value. Next, applying Lemma 7 we can rewrite this as

$$\Psi = \lim_{r \to \infty} \min_{w \in \mathcal{K}, b_0 \in \mathcal{B}, \hat{y}} \max_{\lambda \in [0, r]} \frac{1}{n} \sum_{i=1}^{n} f(y_i, \hat{y}_i) + \lambda \left( \frac{1}{n} \|\hat{y} - X^{\|} w^{\|} - g\|w^{\perp}\|_{\Sigma^{\perp}} - b_0\|^2 - \frac{1}{n} \langle (\Sigma^{\perp})^{1/2} h, w^{\perp} \rangle^2 \right)$$

$$= \lim_{r \to \infty} \min_{w \in \mathcal{K}, b_0 \in \mathcal{B}} \max_{\lambda \in [0, r]} \frac{1}{n} \sum_{i=1}^{n} f_{\lambda}(y_i, (X^{\|} w^{\|})_i + g_i \|w^{\perp}\|_{\Sigma^{\perp}} + b_0) - \lambda \frac{1}{n} \langle (\Sigma^{\perp})^{1/2} h, w^{\perp} \rangle^2$$

$$\leq \min_{w \in \mathcal{K}, b_0 \in \mathcal{B}} \max_{\lambda \geq 0} \frac{1}{n} \sum_{i=1}^{n} f_{\lambda}(y_i, (X^{\|} w^{\|})_i + g_i \|w^{\perp}\|_{\Sigma^{\perp}} + b_0) - \lambda \frac{1}{n} \langle (\Sigma^{\perp})^{1/2} h, w^{\perp} \rangle^2$$

where in the second equality we used the definition of the Moreau envelope and the minimax theorem (Sion 1958) to move the minimum over $\hat{y}$ inside the max.

Next, observing that the first term only depends on $\phi(w)$ we can write this equivalently as

$$\min_{\substack{\phi(w): w \in \mathcal{K} \\ b_0 \in \mathcal{B}}} \max_{\lambda \geq 0} \left[ \frac{1}{n} \sum_{i=1}^{n} f_{\lambda}(y_i, (X^{\|} w^{\|})_i + g_i \|w^{\perp}\|_{\Sigma^{\perp}} + b_0) - \lambda \frac{1}{n} \max_{u \in \mathcal{K}: \phi(u) = \phi(w)} \langle (\Sigma^{\perp})^{1/2} h, u^{\perp} \rangle^2 \right]$$

which proves the conclusion, using that $(X^{\|} w^{\|})_i + g_i \|w^{\perp}\|_{\Sigma^{\perp}} + b_0$ is equivalent in law to $\langle \tilde{w}, \tilde{x} \rangle + b_0$ where $\tilde{w} = \phi(w)$. $\qquad\square$

### F.1 Geometric Interpretation

In this section, we elaborate on the discussion from Section 7 to explain how the result Theorem 4 is a dual result which witnesses tightness of Theorem 1, and to give a geometric interpretation of both results by connecting them to summary functional $\psi(w, b)$ defined in (49). A couple of new results are also established in this subsection, but they are not used in the rest of the paper.

Recall that the main result of this paper, Theorem 1, establishes an upper bound on the test error of an arbitrary predictor $w$ in terms of the training error $\hat{L}_f(w, b)$ and complexity functional $C_{\delta}(w)$. How can we choose the complexity functional $C_{\delta}(w)$ to optimize the bound? In this section, we show that when analyzing the Constrained Empirical Risk Minimizer over $(w, b) \in \mathcal{K} \times \mathcal{B}$ with $\mathcal{K}, \mathcal{B}$ bounded convex sets

$$(\hat{w}, \hat{b}) = \arg \min_{w \in \mathcal{K}, b \in \mathcal{B}} \hat{L}_f(w, b)$$

choosing $C_{\delta}(w)$ based on the *local Gaussian width* of the projected set $Q\mathcal{K}$ will result in an essentially tight generalization bound. (Recall from Definition 4 that $Q$ is the projection orthogonal to the space $w_1^*, \ldots, w_k^*$ which the true regression function in the GLM depends upon.)

The characterization of the performance of constrained ERM we present connects to and builds upon ideas and themes explored previously in a long line of work in the M-estimation literature. For instance, the previous work of Thrampoulidis et al. (2018) (see also references within and our Section 2) gives a similar asymptotic characterization for the performance of constrained/regularized ERM. Compared with that work, here we focus on non-asymptotic results, which apply outside

of the proportional scaling limit, and we establish a connection between this characterization and generalization bounds (which apply to all predictors, not just the ERM). Another difference to that result is that ours applies to generative models of the data beyond just linear regression, in particular GLMs, a setting which has been considered in other works in the CGMT literature (e.g. Montanari et al. 2019; Liang and Sur 2020; Thrampoulidis et al. 2020). In the special case of regression with the squared loss, we recover the nonasymptotic local Gaussian width theory of Zhou et al. (2021).

**Informal Summary.** Before stating the formal results, we start with an informal discussion summarizing the key results and their geometric interpretation. First, we observe that the conclusion of our main result (Theorem 1) can be naturally rearranged as a lower bound on the training loss:

$$\max_{\lambda \geq 0} \left[ L_{f_\lambda}(w, b) - \frac{\lambda C(w)^2}{n} \right] \leq \hat{L}_f(w, b), \tag{47}$$

where for this informal overview we write $C(w) = C_\delta(w)$ to omit the dependence on the failure probability, and also ignore the small error term $\epsilon_{\lambda,\delta}$. A key observation at this point is that the test error $L_{f_\lambda}(w, b)$ depends on $w$ only through its projection $\phi(w)$ from Definition 4: in other words, via its projection onto the span of $w_1^*, \ldots, w_k^*$ and its Mahalanobis norm in the orthogonal space $\|\Sigma^{1/2} Q w\|$. It is natural to choose $C(w)$ depending only on $\phi(w)$.

Hence a natural choice of $C(w)$ is the (local) *Gaussian width*

$$C(w) := \mathop{\mathbb{E}}_{x \sim \mathcal{N}(0, \Sigma)} \sup_{v \in \mathcal{K}_{\phi(w)}} \langle Q v, x \rangle \tag{48}$$

where the localized set $\mathcal{K}_{\phi(w)}$ is defined as

$$\mathcal{K}_{\phi(w)} := \{ v \in \mathcal{K} : v^\parallel = w^\parallel, r(v) \leq r(w) \}$$

and the notation indicates that this set only depends on $w$ through $\phi(w)$, equivalently $w^\parallel$ and $r(w)$. With this choice of $C(w)$, we define the summary functional

$$\psi(w, b) = \psi(\phi(w), b) := \max_{\lambda \geq 0} \left[ L_{f_\lambda}(w, b) - \frac{\lambda C(w)^2}{n} \right] \tag{49}$$

to be the left hand side of (47) (where the notation $\psi(\phi(w), b)$ is used to indicate that $\psi$ depends on $w$ only through $\phi(w)$). We will obtain two major conclusions:

1. Formalizing the previous discussion, the conclusion of Theorem 13 is that with some small finite sample corrections, this choice of $C(w)$ satisfies the assumption of Theorem 1 and so $\psi(w, b)$ indeed lower bounds the training error $\hat{L}_f(w, b)$ as in (47).

2. The conclusion of Theorem 4 is that the lower bound in (47) with this $C(w)$ is tight for the constrained ERM. In other words, with high probability

$$\min_{w \in \mathcal{K}, b \in \mathcal{B}} \hat{L}_f(w, b) \approx \min_{w \in \mathcal{K}, b \in \mathcal{B}} \psi(w, b)$$

   where the right-hand-side is deterministic (and the right hand side optimization depends on $w$ only through the low-dimensional vector $\phi(w)$). This is established by upper bounding the training error via an application of the Convex Gaussian Minmax Theorem.

Combining the two conclusions, we see that when we apply our generalization bound (Theorem 1) with a sufficiently tight choice of $C(w)$ based on the local gaussian width and the optimal envelope parameter $\lambda$, it will predict the actual generalization error of the constrained ERM. So our generalization bound is tight in a pretty general situation; in particular, when the constrained ERM is consistent under proportional scaling (the setting most commonly considered in the asymptotic CGMT literature).

To clarify the geometric interpretation of this result, we also show in Lemma 14 that with this choice of $C(w)$, the left hand side of (47) will be convex in $w$ and $b$; hence, for a fixed upper bound on the training error there is a corresponding sublevel set of the convex function which consists of the points whose training error satisfy the constraint, and as the upper bound shrinks this set will narrow around the minimum of the convex function.

**Formal Results.** First, we formalize the idea that $\psi(w, b)$ is a lower bound on the training error $\hat{L}_f(w, b)$. As in the general Theorem 1, we take the one-sided concentration of the low-dimensional surrogate problem as an assumption to state a general result, since the precise details of that concentration estimate will depend on the exact setting. To give a finite sample result, we define a straightforward approximation $C_{\delta,\rho}(w)$ of the local gaussian width functional (48) which is defined based on a $\rho$-net approximation of $\phi(\mathcal{K})$, and includes the dependence on the failure probability $\delta$; since $\phi(\mathcal{K})$ is a low-dimensional set living in $\mathbb{R}^{k+2}$, the contribution of this correction (just like the contribution from the error term in the low-dimensional concentration assumption (6)) will become negligible if we consider an asymptotic setting $n \to \infty$ with $k$ fixed.

**Lemma 13.** *Let $\mathcal{K} \subset \mathbb{R}^d$ and $\mathcal{B} \subset \mathbb{R}$. Suppose that we have assumption (6) from Theorem 1 with error parameter $\epsilon_{\lambda,\delta}(\tilde{w}, \tilde{b})$ uniformly over envelope parameter $\lambda \geq 0$. Let $\rho > 0$ be arbitrary, and let $\mathcal{S}$ be a proper $\rho$-covering in Euclidean norm of the set $\{\phi(w) : w \in \mathcal{K}\}$ so that for every $w \in \mathcal{K}$ there exists $w'$ with $\phi(w') \in \mathcal{S}$ such that*

$$\|\phi(w) - \phi(w')\|_2 < \rho.$$

*and define (where as above, $w'$ denotes the element in the covering corresponding to $w$)*

$$C_{\delta,\rho}(w) := \mathop{\mathbb{E}}_{x \sim \mathcal{N}(0,\Sigma)} \left[ \sup_{v \in \mathcal{K}_{\phi(w'),\rho}} \langle Qv, x \rangle \right] + (r(w') + \rho)\sqrt{2\log(16|\mathcal{S}|/\delta)}$$

*where*

$$\mathcal{K}_{\phi(w),\rho} := \{v \in \mathcal{K} : \|v^{\|} - w^{\|}\|_\Sigma < \rho, r(v) \leq r(w) + \rho\}.$$

*Then:*

1. *With probability at least $1 - \delta/4$, we have for all $w \in \mathcal{K}$ that*

$$\langle Qw, x \rangle \leq C_{\delta,\rho}(w),$$

   *i.e. the assumption (7) of Theorem 1 is satisfied.*

2. *As an immediate consequence of Theorem 1, we have with probability at least $1 - \delta$ that*

$$\sup_{\lambda \geq 0} \left[ L_{f_\lambda}(w, b) - \epsilon_{\lambda,\delta}(\phi(w), b) - \lambda \frac{C_{\delta,\rho}(w)^2}{n} \right] \leq \hat{L}_f(w, b)$$

   *uniformly over $w \in \mathcal{K}$, $b \in \mathcal{B}$.*

*Proof.* We only need to check the first conclusion, since the second one follows immediately by Theorem 1. First, observe from expanding the definitions that

$$\|w^{\|} - (w')^{\|}\|_\Sigma^2 + (r(w) - r(w'))^2 = \|\phi(w) - \phi(w')\|_2^2 < \rho$$

so that $w \in \mathcal{K}_{\phi(w'),\rho}$. Next, observe by applying Gaussian concentration (Theorem 5) and the union bound over $\mathcal{S}$ that with probability at least $1 - \delta/4$, for $x \sim \mathcal{N}(0, \Sigma)$ and every $w'$ with $\phi(w') \in \mathcal{S}$ we have that

$$\sup_{v \in \mathcal{K}_{\phi(w'),\rho}} \langle Qv, x \rangle \leq \mathop{\mathbb{E}}_{x \sim \mathcal{N}(0,\Sigma)} \left[ \sup_{v \in \mathcal{K}_{\phi(w'),\rho}} \langle Qv, x \rangle \right] + (r(w') + \rho)\sqrt{2\log(16|\mathcal{S}|/\delta)}$$

where we use that the supremum is $(r(w') + \rho)$-Lipschitz because every $v \in \mathcal{K}_{\phi(w),\rho}$ satisfies $\|\Sigma^{1/2}Qv\| = r(v) \leq r(w') + \rho$, and the supremum of Lipschitz functions is Lipschitz with the same constant. Since we showed that $w \in \mathcal{K}_{\phi(w'),\rho}$, we then have that

$$\langle Qw, x \rangle \leq C_{\delta,\rho}(w)$$

as desired. $\qquad\qquad\qquad\qquad\qquad\qquad\qquad\qquad\qquad\qquad\qquad\qquad\qquad\qquad\qquad\qquad\square$

We now discuss how Theorem 4 formalizes the idea that the training error of ERM is the minimum of $\psi(w, b)$. To understand the statement, take $w_0, b_0$ to be minimizers of $\psi(w, b)$. We observe that there exists such minimizers so that

$$C(w) = \mathop{\mathbb{E}}_{x \sim \mathcal{N}(0,\Sigma)} \sup_{v \in \mathcal{K}_{\phi(w)}} \langle Qv, x \rangle = \mathop{\mathbb{E}}_{x \sim \mathcal{N}(0,\Sigma)} \sup_{\phi(v)=\phi(w)} \langle Qv, x \rangle,$$

i.e. so that the optimizing $v$ satisfies $r(v) = r(w)$, otherwise we can replace $w$ by $v$ without reducing $\psi$. Given this observation, we have that the quantity (17) will concentrate about $\psi(w_0, b_0)$ and the best choice of $w_0, b_0$ to make is the minimizer of this quantity, so that we set $\tau$ to be

$$\tau \approx \min_{w_0 \in \mathcal{K}, b_0 \in \mathcal{B}} \psi(w_0, b_0)$$

and this upper bounds the training error of constrained ERM as discussed in the informal overview. Again, see Theorem 4 for the formal version of this.

Finally, we formalize the claim that the summary functional $\psi(w, b)$ defined in (49) is convex. This is not used in the proofs of the main results above, but (as explained earlier) makes the geometric interpretation of the result clearer, and generalizes the convexity of analogous summary functionals observed in previous work for the well-specified regression setting, including Thrampoulidis et al. 2018; Zhou et al. 2021. We note that this convexity will be approximate for the finite-sample version $\sup_{\lambda \geq 0} \left[ L_{f_\lambda}(w, b) - \epsilon_{\lambda, \delta}(\phi(w), b) - \lambda \frac{C_{\delta, \rho}(w)^2}{n} \right]$ in the conclusion of Theorem 13, because of the finite-sample error terms like $\epsilon_{\lambda, \delta}$. In some settings, the finite-sample version of the functional can also be made to be convex: see Zhou et al. 2021 for the case of regression with squared loss.

**Lemma 14.** *Given that the loss $f(y, \hat{y})$ is convex in $\hat{y}$ and $\mathcal{K}, \mathcal{B}$ are convex sets, the functional $C(w) = C(\phi(w))$ defined in (48) is concave as a function of $\phi(w)$ and $\psi(w, b) = \psi(\phi(w), b)$ defined in (49) is convex as a function of $(\phi(w), b)$.*

*Proof.* First we show $C(w)$ is concave as a function of $\phi(w)$. Recall from (48) that

$$C(w) = \mathbb{E}_{x \sim \mathcal{N}(0, \Sigma)} \sup_{v \in \mathcal{K}_{\phi(w)}} \langle Qv, x \rangle$$

where

$$\mathcal{K}_{\phi(w)} = \{v \in \mathcal{K} : v^{\|} = w^{\|}, r(v) \leq r(w)\}.$$

It suffices to prove that for any $x$, the function

$$F(w) = F(\phi(w)) := \sup_{v \in \mathcal{K}_{\phi(w)}} \langle Qv, x \rangle$$

is concave in $\phi(w)$. If $\phi(w) = \alpha\phi(w_1) + (1 - \alpha)\phi(w_2)$, $v_1$ is a maximizer of $F(w_1)$ and $v_2$ is a maximizer of $F(w_2)$ then

$$r(\alpha v_1 + (1 - \alpha)v_2) \leq \alpha r(v_1) + (1 - \alpha)r(v_2) \leq \alpha r(w_1) + (1 - \alpha)r(w_2) = r(w)$$

so $\alpha v_1 + (1 - \alpha)v_2 \in \mathcal{K}_{\phi(w)}$ and so

$$F(w) \geq \langle Q(\alpha v_1 + (1 - \alpha)v_2), v \rangle = \alpha F(w_1) + (1 - \alpha)F(w_2)$$

which proves the concavity.

Next we prove convexity of $\psi$. By expanding the definition of the Moreau envelope, we see that

$$\begin{aligned}
\psi(w, b) &= \max_{\lambda \geq 0} \left[ L_{f_\lambda}(w, b) - \frac{\lambda C(w)^2}{n} \right] \\
&= \max_{\lambda \geq 0} \left[ \mathbb{E} \min_u f(y, u) + \lambda(u - \langle w, x \rangle - b)^2 - \frac{\lambda C(w)^2}{n} \right] \\
&= \max_{\lambda \geq 0} \left[ \min_g \mathbb{E} f(y, \langle w, x \rangle + b + g(x, y)) + \lambda g(x, y)^2 - \frac{\lambda C(w)^2}{n} \right] \\
&= \min_{g: \sqrt{\mathbb{E} g(x, y)^2} \leq C(w)/\sqrt{n}} \mathbb{E} f(y, \langle w, x \rangle + b + g(x, y))
\end{aligned}$$

and we claim the final expression is convex in $w$ and $b$. This follows from Lemma 15 because the objective $\mathbb{E} f(y, \langle w, x \rangle + b + g(x, y))$ is jointly convex in $\phi(w), g, b$, and the minimization is over the constraint $\sqrt{\mathbb{E} g(x, y)^2} - C(w)/\sqrt{n} \leq 0$ which is a jointly convex constraint. □

The following lemma is a version of a standard fact in convex analysis, see e.g. Section 3.2.5 of Boyd et al. 2004.

**Lemma 15.** *Suppose that real-valued functions $f(x, y)$ and $g(x, y)$ are both jointly convex in $(x, y) \in \mathcal{X} \times \mathcal{Y}$ where $\mathcal{X}, \mathcal{Y}$ are convex sets. Then*

$$h(x) := \inf_{y \in \mathcal{Y} : f(x,y) \leq 0} g(x, y)$$

*is a convex function on $\mathcal{X}$.*

*Proof.* Suppose that $x = \alpha x_1 + (1 - \alpha)x_2$ and $y_1, y_2$ are arbitrary points such that both $f(x_1, y_1), f(x_2, y_2) \leq 0$. By joint convexity, we have that

$$f(\alpha x_1 + (1 - \alpha)x_2, \alpha y_1 + (1 - \alpha)y_2) \leq \alpha f(x_1, y_1) + (1 - \alpha)f(x_2, y_2) \leq 0$$

and so

$$h(x) \leq g(\alpha x_1 + (1 - \alpha)x_2, \alpha y_1 + (1 - \alpha)y_2) \leq \alpha g(x_1, y_1) + (1 - \alpha)g(x_2, y_2).$$

Taking the infimum over all such $y_1, y_2$ such $f(x_1, y_1), f(x_2, y_2) \leq 0$ proves that

$$h(x) \leq \alpha h(x_1) + (1 - \alpha)h(x_2)$$

which shows the convexity. $\qquad\square$

*A simple example.* To sketch how the summary functional $\psi$ works and connect to the previous literature, we consider a simple example (Ordinary Least Squares). To start with, we consider a well-specified model with $y = \langle w^*, x \rangle + \xi$ where $\xi$ is noise independent of $x$ with variance $\sigma^2$ and bounded eighth moment. Then the summary functional for $f$ the squared loss and taking $C(w) \approx \|Qw\|_\Sigma \sqrt{d}$ is (using Lemma 20)

$$\psi(w, b) = (\sqrt{L(w, b)} - \|Qw\|_\Sigma \sqrt{d/n})^2 = (\sqrt{\sigma^2 + \|w - w^*\|_\Sigma^2 + b^2} - \|Qw\|_\Sigma \sqrt{d/n})^2.$$

Note $\|w - w^*\|_\Sigma^2 = \|w^\| - w^*\|_\Sigma^2 + \|Qw\|_\Sigma^2$ by the Pythagorean Theorem. To minimize $\psi$, it is optimal to take $w^\| = w^*$ and $b = 0$ which leaves choosing $r(w) = \|Qw\|_\Sigma$ to minimize

$$(\sqrt{\sigma^2 + r(w)^2} - r(w)\sqrt{d/n})^2$$

and this in turn is minimized at $r(w) = \sigma^2(d/n)/(1 - d/n)$, which will be the excess test loss of the constrained ERM. Note that to make the calculation easy, we considered a well-specified model and the summary functional reduced to the same one as in Zhou et al. 2021 once we solved the optimization over $\lambda$, and the calculation can be made rigorous and nonasymptotic following the arguments there; see also Thrampoulidis et al. 2018 and references for related asymptotic results. In this example, it can be checked that the calculation generalizes in a straightforward way to misspecified models under our general assumptions, if we let $w^*$ to be the minimizer of the population squared loss (i.e. the oracle predictor.) and defining the excess test loss to be the gap compared to $w^*$.

# G $\ell_2$ Benign Overfitting

In this section, we give the proofs of the result for benign overfitting under the $\ell_2$ condition. We continue to make use of the additional covariance split notation introduced in Appendix C.

## G.1 Properties of Sqrt-Lipschitz Functions

In this section, we establish some elementary properties of the squares of Lipschitz functions. This is a natural class to consider since in particular, the squared loss and squared hinge loss both fall into this class of functions. We say a function $f : \mathbb{R} \to \mathbb{R}_{\geq 0}$ is $L$-sqrt-Lipschitz if $\sqrt{f}$ is $L$-Lipschitz. Since

$$\frac{1}{2}f(x)^{-1/2}f'(x) = \frac{d}{dx}\sqrt{f(x)}$$

we can equivalently say that a function $f$ is $L$-sqrt-Lipschitz if

$$|f'(x)| \leq 2L\sqrt{f(x)}$$

for all $x$. Based on this characterization, one can observe that any $H$-smooth and nonnegative function is $\sqrt{H}$-sqrt-Lipschitz; this is proved in Lemma 2.1 of Srebro et al. 2010 although not using this terminology. We proceed to establish some useful properties of sqrt-Lipschitz functions. First, we show that $L$-sqrt-Lipschitz functions form a convex set.

**Lemma 16.** *If $f$ is L-sqrt-Lipschitz convex and $g$ is L-sqrt-Lipschitz convex then so is $(1-\alpha)f + \alpha g$ for any $\alpha \in [0,1]$.*

*Proof.* Observe that

$$|(1-\alpha)f'(x) + \alpha g'(x)| \leq (1-\alpha)|f'(x)| + \alpha|g'(x)| \leq 2L[(1-\alpha)\sqrt{f(x)} + \alpha\sqrt{g(x)}]$$
$$\leq 2L\sqrt{(1-\alpha)f(x) + \alpha g(x)}$$

where the second step is the assumption that $f$ and $g$ are $L$-sqrt-Lipschitz and the last step uses the concavity of the square-root function. □

Next, the following lemma formalizes the idea that sqrt-Lipschitz functions satisfy a local and scale-sensitive version of the Lipschitz property.

**Lemma 17.** *Suppose that $f(x)$ is convex and L-sqrt-Lipschitz. Then for any $\epsilon > 0$,*

$$f(x+h) \geq (1-\epsilon)f(x) - L^2 h^2/\epsilon.$$

*Proof.* Observe that

$$f(x+h) \geq f(x) + f'(x)h \geq f(x) - 2L\sqrt{f(x)}|h| \geq f(x) - \epsilon f(x) - L^2 h^2/\epsilon$$

where the first inequality is by convexity, the second inequality is by the $L$-sqrt-Lipschitz property, and the third inequality is the AM-GM inequality. □

This leads to a corresponding local Lipschitz property of the training loss.

**Lemma 18.** *Let $\epsilon \in (0,1)$ be arbitrary, let $w_0 \in \mathbb{R}^d$ and $b_0 \in \mathbb{R}$. Suppose that nonnegative loss function $f(\hat{y}, y)$ is convex and L-sqrt-Lipschitz in $\hat{y}$. The following inequality holds determinsitically for any $x_1, \ldots, x_n \in \mathbb{R}^d$, $y_1, \ldots, y_n \in \mathbb{R}$, $w \in \mathbb{R}^d$, and $b \in \mathbb{R}$:*

$$(1-\epsilon)\hat{L}_f(w,b) \leq \hat{L}_f(w_0, b_0) + \frac{2L^2}{\epsilon n}\sum_{i=1}^n \langle w - w_0, x_i\rangle^2 + 2(b-b_0)^2/\epsilon$$

*Proof.* By applying Lemma 17, we have that

$$f(\langle w_0, x_i\rangle + b_0, y_i) \geq (1-\epsilon)f(\langle w, x_i\rangle + b, y_i) - L^2(\langle w - w_0, x_i\rangle + (b-b_0))^2/\epsilon$$

and then applying the inequality $(a+b)^2 \leq 2a^2 + 2b^2$ gives

$$f(\langle w_0, x_i\rangle + b_0, y_i) \geq (1-\epsilon)f(\langle w, x_i\rangle + b, y_i) - 2L^2\langle w - w_0, x_i\rangle^2 - 2(b-b_0)^2/\epsilon.$$

Summing this inequality over $i$ from 1 to $n$ and rearranging gives the conclusion. □

## G.2 Norm Bounds

**Lemma 2.** *Suppose that $f(\hat{y}, y)$ is either squared loss or squared hinge loss. Let $(w^\sharp, b^\sharp) \in \mathbb{R}^{d+1}$ be an arbitrary vector satisfying $Qw^\sharp = 0$ and with probability at least $1 - \delta/4$,*

$$\hat{L}_f(w^\sharp, b^\sharp) \leq L_f(w^\sharp, b^\sharp) + \rho_1(w^\sharp, b^\sharp) \tag{13}$$

*for some $\rho_1(w^\sharp, b^\sharp) > 0$. Then for any $\rho_2 \in (0,1)$, provided $\Sigma^\perp = Q^T\Sigma Q$ satisfies*

$$R(\Sigma^\perp) = \Omega\left(\frac{n\log^2(4/\delta)}{\rho_2}\right), \tag{14}$$

*we have that with probability at least $1 - \delta$ that $\min_{\|w\| \leq B} L_f(w, b^\sharp) = 0$ for $B > 0$ defined by $B^2 = \|w^\sharp\|_2^2 + (1 + \rho_2)\frac{n}{\text{Tr}(\Sigma^\perp)}(L_f(w^\sharp, b^\sharp) + \rho_1)$.*

*Proof.* By Theorem 4 it suffices to show that with probability at least $1 - \delta/2$,

$$\min_{w_0 \in \mathcal{K}, b_0 \in \mathcal{B}} \max_{\lambda \geq 0} \left[ \frac{\lambda}{1+\lambda} \frac{1}{n} \sum_{i=1}^{n} f(y_i, (X^\| w_0^\|)_i + b_0 + g_i \|w_0^\perp\|_{\Sigma^\perp}) - \frac{\lambda}{n} \langle Qx, w_0^\perp \rangle^2 \right] = 0.$$

Using Lemma 20, it suffices to show with probability at least $1 - \delta/2$ that there exists $w_0, b_0$ such that

$$\frac{1}{n} \sum_{i=1}^{n} f(y_i, (X^\| w_0^\|)_i + b_0 + g_i \|w_0^\perp\|_{\Sigma^\perp}) \leq \frac{1}{n} \langle Qx, w_0^\perp \rangle^2.$$

Decompose $w_0 = w_0^\| + w_0^\perp$ where $w_0^\perp = Q w_0$; then using Lemma 18, we have that for any $\epsilon > 0$

$$(1-\epsilon)\hat{L}_f(w, b_0) \leq \hat{L}_f(w^\|, b_0) + \frac{2}{\epsilon n} \sum_{i=1}^{n} g_i^2 \|w_0^\perp\|_{\Sigma^\perp}^2$$

so it suffices to show that with probability $1 - \delta/2$, there exists $w_0, b_0$ and $\epsilon > 0$ with

$$\frac{1}{1-\epsilon} \hat{L}_f(w^\|, b_0) + \frac{2}{\epsilon(1-\epsilon)n} \sum_{i=1}^{n} g_i^2 \|w_0^\perp\|_{\Sigma^\perp}^2 \leq \frac{1}{n} \langle Qx, w_0^\perp \rangle^2.$$

We consider $w_0^\perp = \alpha \frac{Qx}{\|Qx\|}$ for some constant $\alpha > 0$ to be determined later. Observe that $Qx$ is equal in law to $(\Sigma^\perp)^{1/2} H$ for $H \sim \mathcal{N}(0, I_d)$ with $H$ independent of $X^\|$ and $y_1, \ldots, y_n$. Plugging this in, what we want to show is

$$\frac{1}{1-\epsilon} \hat{L}_f(w^\|, b_0) + \frac{2}{\epsilon(1-\epsilon)n} \alpha^2 \sum_{i=1}^{n} g_i^2 \frac{\|(\Sigma^\perp)H\|_2^2}{\|(\Sigma^\perp)^{1/2}H\|_2^2} \leq \frac{\alpha^2}{n} \|(\Sigma^\perp)^{1/2}H\|_2^2. \tag{50}$$

By the union bound, the following occur together with probability at least $1 - \delta/2$ for some absolute constant $C > 0$:

1. Using the first part of Lemma 19, we have

$$\|(\Sigma^\perp)^{1/2} H\|_2^2 \geq \left( 1 - C \frac{\log(4/\delta)}{\sqrt{R(\Sigma^\perp)}} \right) \text{Tr}(\Sigma)$$

2. Using the last part of Lemma 19, we have

$$\frac{\|\Sigma^\perp H\|_2^2}{\|(\Sigma^\perp)^{1/2}H\|_2^2} \leq C \log(4/\delta) \frac{\text{Tr}((\Sigma^\perp)^2)}{(\text{Tr}\,\Sigma)^2}$$

3. Using subexponential Bernstein's inequality (Theorem 2.8.1 of Vershynin 2018), requiring $n = \Omega(\log(1/\delta))$,

$$\frac{1}{n} \sum_i g_i^2 \leq 2.$$

4. Using (13),

$$\hat{L}_f(w^\sharp, b^\sharp) \leq L_f(w^\sharp, b^\sharp) + \rho_1.$$

Taking $w_0^\| = w^\sharp$ and $b_0 = b^\sharp$, we therefore have

$$\frac{1}{1-\epsilon} \hat{L}_f(w^\sharp, b^\sharp) + \frac{2}{\epsilon(1-\epsilon)n} \alpha^2 \sum_{i=1}^{n} g_i^2 \frac{\|\Sigma^\perp H\|_2^2}{\|(\Sigma^\perp)^{1/2}H\|_2^2}$$

$$\leq \frac{1}{1-\epsilon} (L_f(w^\sharp, b^\sharp) + \rho_1) + \frac{4C}{\epsilon(1-\epsilon)} \alpha^2 \log(4/\delta) \frac{\text{Tr}((\Sigma^\perp)^2)}{\text{Tr}(\Sigma^\perp)}$$

$$\leq \frac{1}{1-\epsilon} (L_f(w^\sharp, b^\sharp) + \rho_1) + \frac{4Cn}{\epsilon(1-\epsilon)R(\Sigma^\perp)} \log(4/\delta) \frac{\alpha^2 \text{Tr}(\Sigma^\perp)}{n}$$

where in the last step we used the definition of $R(\Sigma^\perp)$ and on the other hand we have

$$\frac{\alpha^2 \|(\Sigma^\perp)^{1/2} H\|_2^2}{n} \geq \left(1 - C\frac{\log(4/\delta)}{\sqrt{R(\Sigma^\perp)}}\right) \frac{\alpha^2 \operatorname{Tr}(\Sigma^\perp)}{n}$$

which means we have the desired (50) provided

$$\left(1 - C\frac{\log(4/\delta)}{\sqrt{R(\Sigma^\perp)}} - \frac{4Cn\log(4/\delta)}{\epsilon(1-\epsilon)R(\Sigma^\perp)}\right)\alpha^2 \geq \frac{n}{\operatorname{Tr}(\Sigma^\perp)}\frac{1}{1-\epsilon}(L_f(w^\sharp, b^\sharp) + \rho_1)$$

and this satisfies the constraint $\|w^\sharp\|^2 + \alpha^2 \leq B^2$ provided that

$$\frac{1}{(1-\epsilon)\left(1 - C\frac{\log(4/\delta)}{\sqrt{R(\Sigma^\perp)}} - \frac{4Cn\log(4/\delta)}{\epsilon(1-\epsilon)R(\Sigma^\perp)}\right)} \leq 1 + \rho_2.$$

Taking $\epsilon = \rho_2/10$, this can be guaranteed if

$$R(\Sigma^\perp) = \Omega\left(\frac{n\log^2(4/\delta)}{\rho_2}\right).$$

$\square$

Below are some supporting lemmas used in the proof.

**Lemma 19** (Lemma 10 of Koehler et al. 2021). *For any covariance matrix $\Sigma$ and $H \sim \mathcal{N}(0, I_d)$, it holds that with probability at least $1 - \delta$,*

$$1 - \frac{\|\Sigma^{1/2}H\|_2^2}{\operatorname{Tr}(\Sigma)} \lesssim \frac{\log(4/\delta)}{\sqrt{R(\Sigma)}} \tag{51}$$

*and*

$$\|\Sigma H\|_2^2 \lesssim \log(4/\delta)\operatorname{Tr}(\Sigma^2). \tag{52}$$

*Therefore, provided that $R(\Sigma) \gtrsim \log(4/\delta)^2$, it holds that*

$$\left(\frac{\|\Sigma H\|_2}{\|\Sigma^{1/2}H\|_2}\right)^2 \lesssim \log(4/\delta)\frac{\operatorname{Tr}(\Sigma^2)}{\operatorname{Tr}(\Sigma)}. \tag{53}$$

**Lemma 20.** *Suppose that $a, b > 0$. Then if $a/b > 1$, we have*

$$\max_{\lambda \geq 0}\left[\frac{\lambda}{1+\lambda}a - \lambda b\right] = (\sqrt{a} - \sqrt{b})^2,$$

*and if $a/b \leq 1$ then*

$$\max_{\lambda \geq 0}\left[\frac{\lambda}{1+\lambda}a - \lambda b\right] = 0.$$

*Proof.* Observe that the objective can be rewritten as

$$g(\lambda) := a - \frac{1}{1+\lambda}a - \lambda b$$

and the derivative of this expression with respect to $\lambda$ is

$$g'(\lambda) = \frac{1}{(1+\lambda)^2}a - b.$$

Therefore the unique critical point of $g$ on the domain $(-1, \infty)$ is at $1 + \lambda = \sqrt{a/b}$. This is the global maximum of $g$ on this domain because $g$ goes to $-\infty$ as $\lambda \to -1$ and as $\lambda \to \infty$. At this point, we have that

$$g(\lambda) = a - \sqrt{ab} - (\sqrt{a/b} - 1)b = a + b - 2\sqrt{ab} = (\sqrt{a} - \sqrt{b})^2.$$

If $a/b > 1$ this is the global maximum on $[0, \infty)$. Otherwise, the maximum is at the boundary at $\lambda = 0$. $\square$

### G.3 Consistency

**Lemma 1.** *In the setting of Theorem 1, letting $\Sigma^\perp = Q^T \Sigma Q$, the following $C_\delta(w)$ will satisfy (7):*
$C_\delta(w) = \|w\|_2 \left[ \sqrt{\mathrm{Tr}(\Sigma^\perp)} + 2\sqrt{\|\Sigma^\perp\|_{op} \log(8/\delta)} \right].$

*Proof.* First, we have by Jensen's inequality that

$$\mathbb{E}\left[ \sup_{\|w\| \leq 1} \langle Qx, w \rangle \right] = \mathbb{E}\|Qx\|_2 \leq B\sqrt{\mathbb{E}\|Qx\|_2^2} = \sqrt{\mathrm{Tr}(\Sigma^\perp)}.$$

Applying Theorem 5 gives that with probability at least $1 - \delta/4$,

$$\sup_{\|w\| \leq 1} \langle Qx, w \rangle \leq \sqrt{\mathrm{Tr}(\Sigma^\perp)} + 2\left( \sup_{\|u\| \leq 1} \|(\Sigma^\perp)^{1/2} u\|_2 \right) \sqrt{\log(8/\delta)}.$$

$\square$

**Lemma 21.** *In the setting of Lemma 1, suppose that the loss $f$ is the squared loss or squared hinge loss, and correspondingly $\epsilon_{\lambda,\delta}(w) = \frac{\lambda}{1+\lambda}\epsilon_\delta(w)$. Then with probability at least $1 - \delta$,*

$$L_f(w,b) - \epsilon_\delta(\phi(w), b) \leq \left( \sqrt{\hat{L}_f(w,b)} + \frac{\|w\|_2}{\sqrt{n}} \left[ \sqrt{\mathrm{Tr}(\Sigma^\perp)} + 2\|(\Sigma^\perp)^{1/2}\|_{op}\sqrt{\log(2/\delta)} \right] \right)^2.$$

*Proof.* This follows by combining Lemma 1, Corollary 2, and Corollary 4. $\square$

**Theorem 3.** *Let $(\hat{w}, \hat{b}) = \arg\min_{w \in \mathbb{R}^d, b \in \mathbb{R}\,:\,\hat{L}_f(w,b)=0} \|\hat{w}\|_2$ be the minimum-$\ell_2$ norm predictor with zero training error. In the setting of Lemma 2, we have*

$$L_f(\hat{w}, \hat{b}) - \epsilon_\delta(\phi(\hat{w}), \hat{b}) \leq (1 + \rho_3) \inf_{w^\sharp \in \mathbb{R}^d, b^\sharp \in \mathcal{B}} \left( L_f(w^\sharp, b^\sharp) + \rho_1(w^\sharp, b^\sharp) + \frac{\|w^\sharp\|_2^2 \, \mathrm{Tr}(\Sigma^\perp)}{n} \right),$$

*where $\rho_3 > 0$ is defined by $1 + \rho_3 = (1 + \rho_2)\left[ 1 + 2\sqrt{\frac{\log(2/\delta)}{r(\Sigma^\perp)}} \right]^2$ and we recall $\rho_1(w^\sharp, b^\sharp)$ from (13).*

*Proof.* It suffices to prove the inequality for fixed $w^\sharp, b^\sharp$: the conclusion follows automatically from the right-continuity of the CDF of $L_f(\hat{w}, \hat{b})$.

From Lemma 2 we have with probability at least $1 - \delta/2$

$$\|\hat{w}\|^2 \leq \|w^\sharp\|_2^2 + (1 + \rho_2)\frac{n}{\mathrm{Tr}(\Sigma_2^\perp)}(L_f(w^\sharp, b^\sharp) + \rho_1)$$

and from Lemma 21 we have for any $w, b$ that with probability at least $1 - \delta/2$

$$L_f(w,b) - \epsilon_\delta(\phi(w), b) \leq \left( \sqrt{\hat{L}_f(w,b)} + \frac{\|w\|_2}{\sqrt{n}} \left[ \sqrt{\mathrm{Tr}(\Sigma^\perp)} + 2\|(\Sigma^\perp)^{1/2}\|_{op}\sqrt{\log(2/\delta)} \right] \right)^2$$

and so for $\hat{w}, \hat{b}$ we have

$$L_f(\hat{w}, \hat{b}) - \epsilon_\delta(\phi(\hat{w}), \hat{b})$$
$$\leq \frac{\|w\|_2^2}{n} \left[ \sqrt{\mathrm{Tr}(\Sigma^\perp)} + 2\|(\Sigma^\perp)^{1/2}\|_{op}\sqrt{\log(2/\delta)} \right]^2$$
$$\leq \left( \|w^\sharp\|_2^2/n + (1 + \rho_2)\frac{1}{\mathrm{Tr}(\Sigma_2^\perp)}(L_f(w^\sharp, b^\sharp) + \rho_1) \right) \left[ \sqrt{\mathrm{Tr}(\Sigma^\perp)} + 2\|(\Sigma^\perp)^{1/2}\|_{op}\sqrt{\log(2/\delta)} \right]^2$$
$$= \left( \frac{\|w^\sharp\|_2^2 \, \mathrm{Tr}(\Sigma^\perp)}{n} + (1 + \rho_2)(L_f(w^\sharp, b^\sharp) + \rho_1) \right) \left[ 1 + 2\|(\Sigma^\perp)^{1/2}\|_{op}\sqrt{\frac{\log(2/\delta)}{\mathrm{Tr}(\Sigma^\perp)}} \right]^2$$

which proves the result (recalling the definition of $r(\Sigma^\perp)$). $\square$

**Corollary 3.** *Suppose that $\mathcal{D}_n$ is a sequence of data distributions following our model assumptions (2), with $k_n$ such that $y = g(\eta_1, \ldots, \eta_{k_n}, \xi)$, and projection operator $Q_n$ defined as in (4). Suppose $f$ is either the squared loss or the squared hinge loss, and define $(w_n^\sharp, b_n^\sharp) = \arg\min_{w,b} L_{f,n}(w, b)$ where $L_{f,n}(w, b)$ is the population loss over distribution $\mathcal{D}_n$ with loss $f$. Suppose that the hypercontractivity assumption (9) holds with some fixed $\tau > 0$ for all $\mathcal{D}_n$. Define $\Sigma_n := \mathbb{E}_{\mathcal{D}_n}[xx^T]$ and $\Sigma_n^\perp = Q_n^T \Sigma_n Q_n$. Suppose that as $n \to \infty$, we have*

$$\frac{n}{R(\Sigma_n^\perp)} \to 0, \quad \frac{\|w_n^\sharp\|_2^2 \operatorname{Tr}(\Sigma_n^\perp)}{n} \to 0, \quad \frac{k_n}{n} \to 0. \tag{15}$$

*Then we have the following convergence in probability, as $n \to \infty$:*

$$\frac{L_{f,n}(\hat{w}_n, \hat{b}_n)}{L_{f,n}(w_n^\sharp, b_n^\sharp)} \to 1, \tag{16}$$

*where $(\hat{w}_n, \hat{b}_n) = \arg\min_{w \in \mathbb{R}^d, b \in \mathbb{R} : \hat{L}_f(w,b) = 0} \|w\|_2$ is the minimum-norm interpolator, and $\hat{L}_{f,n}$ is the training error based on $n$ i.i.d. samples from the distribution $\mathcal{D}_n$.*

*Proof.* The first assumption in (15) directly implies that we can choose a sequence $\rho_{2,n} \to 0$ where $\rho_{2,n}$ is the parameter in (14). Recalling the general fact that $r(\Sigma^\perp)^2 \geq R(\Sigma^\perp)$ (Bartlett et al. 2020), we see that the same assumption implies $1/r(\Sigma^\perp) \to 0$ which implies $\rho_{3,n} \to 0$ where $\rho_{3,n}$ is as defined in Theorem 3.

Combining this with (the proof of) Corollary 1 and using the assumption $k_n/n \to 0$ allows us to handle the $\epsilon_\delta(\phi(\hat{w}), \hat{b})$ term, guaranteeing it is negligible compared to the population loss $L_{f,n}(\hat{w}_n, \hat{b}_n)$.

To see why we can take $\rho_1 \to 0$, we use Chebyshev's inequality after observing

$$\operatorname{Var}(\hat{L}_{f,n}(w_n^\sharp, b_n^\sharp)) = \frac{1}{n} \operatorname{Var}(f(\langle w^\sharp, x \rangle + b, y)) \lesssim \frac{1}{n}(\mathbb{E} f(\langle w^\sharp, x \rangle + b, y))^2$$

where we used independence and the hypercontractivity assumption. $\qquad \square$