# OpenReview forum: "A Non-Asymptotic Moreau Envelope Theory for High-Dimensional Generalized Linear Models"
_NeurIPS.cc/2022/Conference — NeurIPS 2022 Accept_

### Official Review · Reviewer_S5Ha · 2022-07-06

**Rating:** 6
**Confidence:** 3
**Soundness:** 3 good
**Presentation:** 3 good
**Contribution:** 2 fair

**Summary:**

analysis of (one-sided) generalization error using Moreau envelopes.

theorem 1: if the empirical Moreau loss is unformly not much smaller than the expected Moreau loss whp then the expected Moreau loss is less than the empirical loss plus a complexity term, so ERM leads to small expected Moreau loss.

this is applied to develop
1. vc bounds for the expected Moreau loss
2. bounds linear regression with square loss and classification with sq-hinge loss (for the expected loss instead of expected Moreau loss, by exploiting properties of Moreau envelope of square/sq-hinge resp.)
3.  discussion of l1 and hinge loss implications, and relationship to huber loss
4. general Lipschitz losses (improves existing results by a constant factor)
5. general smooth losses given interpolation (generalizes existing work)
6. recover results on benign overfitting



**Questions:**

- The matrix Q in Eq (4): I verified that Q is idempotent, but it is not clear that it is symmetric. are you sure this is the correct definition?

**Limitations:**

- limited to linear multi-index models and gaussian covariates

**Strengths And Weaknesses:**

Strengths:
- clear writing
- easily interpreted results
- results appear to be sound
- the approach to studying generalization via Moreau envelope analysis is an interesting, and the present work initiates a nice connection between recent work in high dimensional statistics with generalization bounds.
- the literature review is quite thorough

Weaknesses:
- the consequences of the main result are not really novel. better or essentially the same (up to constants) results have been obtained using other techniques before.
- the present work would be stronger and more compelling if there was an example where these methods provided a strictly better rate than existing work.
- some of the results only bound the expected Moreau loss, not the expected Loss
- limited to linear multi-index models and gaussian covariates

---

> ### Author Response · Authors · 2022-08-02
> **Author Response Part 1**
>
> Thank you for your feedback. We’ll address some comments and questions below, and attempt to make them clearer in the paper in revision.
>
> > the consequences of the main result are not really novel. better or essentially the same (up to constants) results have been obtained using other techniques before.
>
> While many results can be obtained “up to constants”, the difference of constants that we study here is extremely important.
>
> We are primarily interested in the modern high-dimensional setting where the capacity of the class of functions being fit is large. (Important special cases of this setting include those where benign overfitting  occurs and also the setting of proportional asymptotics.) Recall that in this setting, there has been serious interest and concern as to how generalization bounds apply or if they are even useful (see e.g. “Failures of Model-dependent Generalization Bounds for Least-norm Interpolation” by Bartlett and Long, “Uniform convergence may be unable to explain generalization in deep learning” by Nagarajan and Kolter, etc.). An extra constant factor of two in the generalization bound will result in (for example) a failure to establish consistency when benign overfitting occurs — for discussion of this see e.g. Section 3.4 of “Fit without Fear: remarkable mathematical phenomena of deep learning through the prism of interpolation” by Belkin ‘21, or the cited paper of Zhou et al ‘20.
>
>
> > the present work would be stronger and more compelling if there was an example where these methods provided a strictly better rate than existing work.
>
> Note that the existing asymptotic Moreau envelope framework for M-estimators, as discussed in the beginning of Related Work (see e.g. reference Thrampoulidis et al ‘18), generally does not come with any finite sample guarantee: they give the limiting value of the test loss when the number of samples and dimension go to infinity and are proportional (“proportional scaling”), but do not give explicit control on how fast this convergence is. Our framework is nonasymptotic and so allows us to derive finite-sample results with explicit convergence rates. Previously, such a nonasymptotic analogue of the Moreau envelope theory was only known in the case of a well-specified model,  squared loss, and Gaussian noise (Zhou et al ‘21); that paper works out examples such as the OLS. Even in the well-specified squared loss case, our results generalize the nonasymptotic results to handle heavy-tailed noise.
>
>
> > some of the results only bound the expected Moreau loss, not the expected Loss
>
> The appearance of the Moreau envelope in the test loss is not a weakness of our approach, but an essential strength and a reflection of how M-estimators actually behave in high dimensions.
>
> Here we are benefiting in part from the wisdom of the previous works in asymptotic statistics where the authors realized that the Moreau envelope naturally comes up (see e.g. references cited at the beginning of Section 2). To illustrate why it is so important in our context, let’s consider the behavior of the generalization bounds when looking at the $L_1$ training loss (so $f = |y - \\hat y|$).
>
> - First, we can consider the example of a well-specified Gaussian model where benign overfitting occurs. In this case, in Appendix E.4 we showed that the inequality
> $$L\_f(w) - \hat{L}\_f(w) \\le \\sqrt{C(w)^2/n} $$
> is sharp in the sense that it cannot be improved by a constant factor. Note that this bound directly relates the $L_1$ test loss and $L_1$ training loss, as proposed.
>
>   For interpolators (i.e. w with training error zero), this specializes to the bound
> $$L\_f(w) \\le \\sqrt{C(w)^2/n}  \\tag{*}$$
>
>   Now if we look at the Moreau envelope of the training loss, then our result gives the inequality for $g(y, \hat y) = (y - \hat y)^2$ that
> $$L_g(w) \le \sqrt{C(w)^2/n}$$
>   Using that $\\mathbb E_{Z \\sim N(0,1)} |Z| = \\sqrt{2/\\pi}$, this implies a bound on the *$L_1$ test loss* of
> $$L_f(w) \\le \sqrt{2/\\pi} \\sqrt{C(w)^2/n}$$
> which is clearly much better than (\*) for interpolators. In particular, (\*) can never explain consistency of interpolating noisy data, because the right hand side is always at least $\\sqrt{\\pi/2}$ times larger than the left hand side, whereas the bottom inequality established via the Moreau envelope can explain consistency.
>
>  To summarize, even if we care about the L1 test loss, when we look at interpolators we should use the optimal Moreau envelope generalization bound. (Start of next comment discusses misspecified case)
>
> ----- continued in next comment

---

> > ### Author Response · Authors · 2022-08-02
> > **Author Response Part 2**
> >
> > - When the model is not well-specified and #parameters > #samples, the zero-regularization limit of LAD and least-squares is the same (the minimum-norm interpolator). Our Corollary 3 shows that this predictor is consistent for the squared loss, which will be the relevant Moreau envelope of the $L_1$ loss in this case. In the misspecified setting, the minimizer of the squared loss is generically distinct from the minimizer of the $L_1$ loss so this predictor is _not consistent_ for the test $L_1$ loss. So, if we want to say something strong about the test error of the predictor, we had better look at the squared test loss. The Moreau envelope framework naturally explains how we arrive at this test loss even if we started with a different training loss.
> >
> >
> > > limited to linear multi-index models and gaussian covariates
> >
> > This is indeed a limitation of our work. Please also see our separate comment about the Gaussianity assumption above.
> >
> >
> > > The matrix Q in Eq (4): I verified that Q is idempotent, but it is not clear that it is symmetric. are you sure this is the correct definition?
> >
> > Yes, this is indeed the correct definition (the matrix is not symmetric) and this is why we carefully distinguished $Q$ vs $Q^T$ throughout the paper. (In other words, it is a projection but not an orthogonal projection.) Geometrically, the reason is because $Q$ is a parameter-space representation of an orthogonal projection in function space, where the relevant inner product structure (covariance between mean-zero functions) is not the Euclidean dot product between parameter vectors.
> >
> > To see how $Q$ represents orthogonal projection in function space, we can consider the case $k = 1$ where
> > $Q = I - w^\*\_1 (w^\*\_1)^T \\Sigma$.
> > When we apply this to a vector $w$, we get
> > \begin{align*}
> > Q w
> > &= w - w^\*_1  \\langle w^\*_1, \Sigma w \\rangle
> > \\\\&= w - w^\*\_1 \\operatorname{Cov}(\\langle w^\*\_1,  x \\rangle, \\langle w, x \\rangle)
> > \end{align*}
> > which implies that
> > $$
> > \\operatorname{Cov}( \\langle Q w, x \\rangle, \\langle w^\*\_1, x \\rangle)
> > = \\operatorname{Cov}( \\langle w, x \\rangle, \\langle w^\*\_1, x \\rangle) - \\operatorname{Var}(\\langle w^\*\_1, x \\rangle) \\operatorname{Cov}( \\langle w^\*\_1, x \\rangle, \\langle w, x \\rangle)
> > ;$$
> > since $w^\*\_1$ is normalized to have unit variance, this covariance is zero. Thus $\\langle Q w, x \\rangle$ is the part of the function $\\langle w, x \\rangle$ that is uncorrelated with $\\langle w^\*\_1, x \\rangle$.

---

> > > ### Comment · Reviewer_S5Ha · 2022-08-09
> > > **response to response**
> > >
> > >
> > > > > the present work would be stronger and more compelling if there was an example where these methods provided a strictly better rate than existing work.
> > >
> > > > Note that the existing asymptotic Moreau envelope framework for M-estimators, as discussed in the beginning of Related Work (see e.g. reference Thrampoulidis et al ‘18), generally does not come with any finite sample guarantee: they give the limiting value of the test loss when the number of samples and dimension go to infinity and are proportional (“proportional scaling”), but do not give explicit control on how fast this convergence is. Our framework is nonasymptotic and so allows us to derive finite-sample results with explicit convergence rates. Previously, such a nonasymptotic analogue of the Moreau envelope theory was only known in the case of a well-specified model, squared loss, and Gaussian noise (Zhou et al ‘21); that paper works out examples such as the OLS. Even in the well-specified squared loss case, our results generalize the nonasymptotic results to handle heavy-tailed noise.
> > >
> > > The authors are arguing here that the results and methods are new from the perspective of what is possible using the Moreau envelope approach. I would not disagree with that, and it is a worthwhile contribution. But I don't feel that it refutes my comment.
> > >
> > >
> > >
> > > >> some of the results only bound the expected Moreau loss, not the expected Loss
> > >
> > > > The appearance of the Moreau envelope in the test loss is not a weakness of our approach, but an essential strength and a reflection of how M-estimators actually behave in high dimensions.
> > > Here we are benefiting in part from the wisdom of the previous works in asymptotic statistics where the authors realized that the Moreau envelope naturally comes up (see e.g. references cited at the beginning of Section 2).
> > > ... which is clearly much better than (\*) for interpolators. In particular, (\*) can never explain consistency of interpolating noisy data, because the right hand side is always at least $\sqrt{\pi/2}$ times larger than the left hand side, whereas the bottom inequality established via the Moreau envelope can explain consistency.
> > >
> > > In this case the improvement is a multiplicative factor of ~1.25. This is a material improvement, but it is not of the scale that typical estimates of generalization for parameterized models over estimate the generalization game, it is order of magnitudes smaller, and constant across dimensionality. I am not sure what this illustration was meant to convey in response to my comment then.
> > >
> > > ___
> > > Thanks for the explanation of Q being an oblique projection.

---

> > > > ### Author Response · Authors · 2022-08-10
> > > > **response before end of discussion period**
> > > >
> > > > >>> the present work would be stronger and more compelling if there was an example where these methods provided a strictly better rate than existing work.
> > > >
> > > > >> Note that the existing asymptotic Moreau envelope framework for M-estimators, as discussed in the beginning of Related Work (see e.g. reference Thrampoulidis et al ‘18), generally does not come with any finite sample guarantee ....
> > > >
> > > > > The authors are arguing here that the results and methods are new from the perspective of what is possible using the Moreau envelope approach. I would not disagree with that, and it is a worthwhile contribution. But I don't feel that it refutes my comment.
> > > >
> > > > Even before worrying about convergence rate, there are not many options to compute the asymptotic test and training error of M-estimators in high dimensions (e.g. under proportional scaling of # of parameters and # of dimensions). For ridge regression random matrix theory can be used (though nonasymptotic results may be nontrivial) but for more complex regularizations like \ell_1 (LASSO) the asymptotics are done via Moreau envelope based on gaussian process or message passing techniques (see Related work). So there is general, no alternative approach which identifies the correct limiting error and gives convergence rates.
> > > >
> > > > As illustration, we review a few particular applications where our results improve the existing work. Our result for benign overfitting in regression+classification with misspecified models (Theorem 3) is new as discussed in the paper and comes with explicit convergence rates; for other examples we mentioned in the previous response, e.g. OLS under proportional scaling with misspecification (see previous response and end of Appendix F), we are not aware of alternative proofs of these results. We obtain all of our results by specializing uniform-convergence bounds which apply to all predictors in a class (e.g. to near-empirical risk minimizers, not just the exact ERM) as stated in Sections 5 and 6 and these generalization bounds with sharp constants are all new except the special case covered in (Zhou et al ‘21).
> > > >
> > > > >>> some of the results only bound the expected Moreau loss, not the expected Loss
> > > >
> > > > >>The appearance of the Moreau envelope in the test loss is not a weakness of our approach, but an essential strength and a reflection of how M-estimators actually behave in high dimensions....
> > > >
> > > > > In this case the improvement is a multiplicative factor of ~1.25. This is a material improvement, but it is not of the scale that typical estimates of generalization for parameterized models overestimate the generalization game, it is order of magnitudes smaller, and constant across dimensionality. I am not sure what this illustration was meant to convey in response to my comment then.
> > > >
> > > > To summarize, the purpose of this illustration was to answer the comment “some of the results only bound the expected Moreau loss, not the expected Loss” which was listed as a weakness in the original review. The point is that in this example, a generalization bound with sharp constants which controls the L1 test loss by the L1 training loss (eqn (*) in the comment) gives a bound which does not explain consistency of benign overfitting, whereas the sharp approach which goes via the Moreau envelope as an intermediate step does. So this explains why it is better (not a weakness) to have the generalization bound that controls the Moreau loss, even if we ultimately care about test error under the original loss.
> > > >
> > > > In this example, improving the bound by any factor larger than \sqrt{\pi/2} ~ 0.25 is impossible and would lead to a mathematical contradiction. This is because the data is noisy, so the L1 test loss cannot be smaller than the average absolute value of the noise; hence it is not possible to improve the bound further (e.g. by factors depending on the dimension). The improvement by this constant factor is very important, because it shows the “excess risk” (here, risk of the minimum norm interpolator minus risk of the optimal predictor) goes to zero, i.e. the estimator is consistent, and also gives bounds on the rate of convergence whereas the non-Moreau based bound on the excess risk does not even go to zero.
> > > >
> > > > At a higher level, this means that overfitting the training data in this example does not come at the cost of consistency, which means we can explain in this instance the well-known “benign overfitting” phenomena observed in many works on double descent, etc. Aspects like the dimension dependence in GLMs have been well-studied and there are matching lower bounds for the dimension dependence via information-theoretic arguments and other means ---  those aspects of existing bounds based on e.g. local rademacher complexity are already correct and won’t be improvable. We are working on the constant factor because this part is not optimal in standard generalization bounds and it is crucial for understanding modern phenomena in machine learning, especially benign overfitting.

---

### Official Review · Reviewer_dxeC · 2022-07-11

**Rating:** 6
**Confidence:** 2
**Soundness:** 3 good
**Presentation:** 3 good
**Contribution:** 2 fair

**Summary:**

This paper studies the generalization properties of high-dimensional generalized linear models (GLM). Assuming that the inputs are Gaussian and the underlying data distribution follows a multi-index model, the authors obtained upper bounds for the population Moreau envelope risk in terms of the training loss and the Rademacher complexity of the function class in a variety of overparameterized settings. Specializing their results to several applications, they show connections to existing results in the literature such as the optimistic rates for linear regression and the benign overfitting.

**Questions:**

Major questions/comments:
1. The Gaussian input assumption is quite restrictive. If a well-specified model is assumed, can you relax this assumption?
2. How easy is it to verify Eq (6)? Can you give some examples?
3. One line 198, there is a claim "in many settings of interest, this can be directly checked using the fact that $x$ is Gaussian". I would like to see concrete results to support this claim. What can you say when $x$ is not Gaussian?
4. In Cor. 3, it is required that the inputs are Gaussian and Eq. (9) holds for all $\mathcal{D}_n$. However, Bartlett et al. (2020) do not have these assumptions when the model is well-specified. In the well-specified setting, can you verify or relax these assumptions?

Minor questions/comments:
1. The sentence "the first term is negligible since the dimension is small" on line 158 is confusing. Isn't the main focus here the high-dimensional setting?
2. What are $x_i$s in Eq. (5)? Are they the coordinates of $\tilde x$? If so, what happens to the $(k+1)$-th coordinate?
3. Should the term $\epsilon_{\lambda, \delta}$ in Eq. (6) also depend on $n$?
4. Should the term $C_\delta(w)$ in Eq. (7) depend on $k$, $d$, and $\Sigma$?
5. Why is the claim on line 245 true? Isn't $f_\lambda \rightarrow 0$ as $\lambda \rightarrow 0$?
6. On line 260, is $\tilde f$ convex? If not, how did you get $\hat L_{\tilde f}(w) = 0$?

**Limitations:**

Yes.

**Strengths And Weaknesses:**

Strengths:
The main results in the paper are novel due to their generality -- they holds for GLMs with possible model misspecification. The results in the main text are clearly stated and intuitions are given to parse them. The proofs of these results I checked seem to be technically solid. Finally, the connections to existing literature are very interesting and show the generality of their results.

Weaknesses:
The main issue I have with this paper is that some of the assumptions are not transparent and seem to be difficult to verify. This is the case in the main theorem in Sec. 4 as well as in the results derived in Secs. 5 and 6. One important question the authors did not answer is that whether these assumptions hold in a specific setting, e.g., well-specified linear regression. This, together with the Gaussian input assumption, question the applicability of their results. See the Questions section for more details.

---

> ### Author Response · Authors · 2022-08-02
> **Author Response (part 1/2)**
>
> Thank you for your feedback. We’ll address some comments and questions below, and attempt to make them clearer in the paper in revision.
>
> > The main issue I have with this paper is that some of the assumptions are not transparent and seem to be difficult to verify. This is the case in the main theorem in Sec. 4 as well as in the results derived in Secs. 5 and 6. One important question the authors did not answer is that whether these assumptions hold in a specific setting, e.g., well-specified linear regression. This, together with the Gaussian input assumption, question the applicability of their results. See the Questions section for more details.
>
> We answer this in more detail below, but to quickly summarize our main result (Theorem 1 of Section 4) is a quite general and nice-to-use result given the crucial assumption that the data follows the Gaussian Multi-Index model. While the results are specific to the Gaussian Multi-Index Model, this is a big step beyond the previous theory (e.g. specific to well-specified minimum-$\\ell_2$ norm interpolation, or to well-specified Gaussian linear regression with squared loss) and it already introduces significant complexities and yields new insights.
>
> For example, in our setting it is not a priori clear what test loss the minimum-norm interpolator of the data will try to minimize: it achieves zero training error for many different losses simultaneously, but the corresponding test losses have different global minima. Our theory answers this question (benignly overfit interpolators minimize the test squared loss) and explains the more general mechanism at work — when ERM overfits the data, it solves a type of bias-variance tradeoff where the “bias” term is the test error under a Moreau envelope of the training loss (Theorem 1 + 4, see discussion in Appendix F).
>
> Relatedly, while almost all of the previous work on benign overfitting treats classification and regression separately, our theorems handle both at the same time, and the analysis reveals simple parallels between these two settings.
>
> While the results are specific to the Gaussian Multi-Index Model, our simulations (see Appendix B) suggest the bounds may hold much more generally; it’s indeed an interesting open question how to extend them.
>
> Theorem 1 is a general result for reducing the problem of establishing high-dimensional generalization bounds in Gaussian space to low-dimensional ones. This can work in conjunction with any method for establishing the low-dimensional generalization bound. To illustrate the power of this method, we considered its combination with a standard VC-theoretic generalization bound (Theorem 2 which is cited from Vapnik’s book), but you can use Theorem 1 with any bound and just propagate the corresponding error terms $\\epsilon_{\\lambda,\\delta}$ appropriately.
>
>
> > 1. The Gaussian input assumption is quite restrictive. If a well-specified model is assumed, can you relax this assumption?
>
> We agree; please see our separate comment on this issue above.
>
>
> > 2. How easy is it to verify Eq (6)? Can you give some examples?
>
> The assumption of Equation (6) is exactly a one-sided generalization bound for a linear model and loss $f_{\\lambda}$, with the crucial feature that the dimension of the covariates is $k + 1$ instead of the original (much larger) dimension $d$. (E.g. for a well-specified model, $k = 1$.) This means that any of a vast number of different results could be applied to prove it; since the number of parameters in $(\\tilde w, \\tilde b)$ is small, a particularly good way to prove such generalization bounds is to apply results from Vapnik-Chervonenkis theory. This is exactly what we did in Theorem 2, by applying one particular result from Vapnik’s book, but you could also apply other similar results.
>
> For example, if we don’t want to depend on the hypercontractive constant $\\tau$, we can apply VC theory results which assume bounds on the $p$th moments of the loss instead. In particular we can apply Assertion 3 from Chapter 7.8 of the cited Vapnik ‘82 book — if the fourth moment of the loss for any function in the class is bounded by $M^4$, this result says (6) is true with an additive error term $O(M \\sqrt{\\log(2/\\delta) / n})$. Another nice generalization bound that you could use is Theorem 2 from the Panchenko ‘03 paper cited (which reduces to computing a version of Rademacher complexity).

---

> > ### Comment · Reviewer_dxeC · 2022-08-09
> > **Thank You for Your Response**
> >
> > Thank you for your detailed response. I have no more questions and keep my score unchanged. I recommend the authors incorporate the response in the revised manuscript.

---

> ### Author Response · Authors · 2022-08-02
> **Author Response (Part 2/2)**
>
> > 3. One line 198, there is a claim "in many settings of interest, this can be directly checked using the fact that $x$ is Gaussian". I would like to see concrete results to support this claim. What can you say when $x$ is not Gaussian?
>
> An illustrative example of the first claim is already given in our appendix: Theorem 9 of Section E.2 shows in the regression case why hypercontractivity is satisfied if $x$ is Gaussian and $y$ is a low-degree polynomial plus noise. (This was briefly mentioned in the main text in Section 5.1, but we’ll make this and the link to the appendix more prominent.) In Section E.3 we also gave examples in the classification setting.
>
> Hypercontractivity (assumption (9)) by itself is a very well-studied notion in learning theory, probability theory, etc. and holds in many settings. (We discuss this more in our response to the next question.) However, as noted above, Theorem 1 does rely on the gaussian assumption and that is the key obstacle towards obtaining results for non-gaussian $x$.
>
> > 4. In Cor. 3, it is required that the inputs are Gaussian and Eq. (9) holds for all $\\mathcal D\_n$. However, Bartlett et al. (2020) do not have these assumptions when the model is well-specified. In the well-specified setting, can you verify or relax these assumptions?
>
> First, we wish to clarify that Eq. (9) by itself (hypercontractivity) is a comparatively weak assumption. In the case of the squared loss and a well-specified model, the hypercontractivity is in particular implied by the assumption of Gaussianity or the subgaussian assumptions in Bartlett et al (2020). This is because hypercontractivity in this case is simply the assertion that for any $F = w \\cdot X$, that the eighth and second moments of $F$ are comparable. The subgaussian assumptions in Bartlett et al. guarantee that any constant-degree moment is comparable to the second moment (see e.g. Vershynin’s book for the precise characterization of subgaussianity in terms of moments).
>
> If we literally only want to generalize Corollary 3 in the well-specified setting, it is possible to use more problem-specific techniques. For regression, the paper by Tsigler and Bartlett is a current state of the art result that fits the bill.
>
>
>
> > 1. The sentence "the first term is negligible since the dimension is small" on line 158 is confusing. Isn't the main focus here the high-dimensional setting?
>
> Here we are referring to the much smaller parameter $k$, the dimension of the space which the label $y$ depends on. This is indeed confusing and we’ll rephrase — thanks for pointing it out.
>
> > 2. What are $x\_i$s in Eq. (5)? Are they the coordinates of $\\tilde x$? If so, what happens to the $(k+1)$-th coordinate?
>
> Yes, there is a small typo here, it should say $\\tilde x_i$ which is the coordinate of $\\tilde x$. The $(k+1)$th coordinate is not used for $y$.
>
> > 3. Should the term $\varepsilon\_{\lambda,\delta}$ in Eq. (6) also depend on $n$?
> > 4. Should the term $C\_\\delta(w)$ in Eq. (7) depend on $k$, $d$, and $\\Sigma$?
>
> Yes, both functions are allowed to depend arbitrarily on the “model parameters” $k,d,\\Sigma,n$, which makes sense since it is a nonasymptotic statement. We will edit the theorem statement to make this clearer.
>
> > 5. Why is the claim on line 245 true? Isn't $f\_\\lambda \\to 0$ as $\\lambda \\to 0$?
>
> The claim is true. As we take $\\lambda \\to 0$, both sides of the generalization bound will go to zero. If we renormalize by $(1/\\lambda)$ on both sides, then the left hand side converges to the test squared loss (with the same $1 - o(1)$ factor) and the right hand side converges to $C_{\\delta}(w)^2/n$.
>
> > 6. On line 260, is $\tilde f$ convex? If not, how did you get $\hat L_{\tilde f}(w) = 0$?
>
> $\\tilde f$ may not be convex, but we did use our assumption that $f$ is nonnegative. Here is a detailed proof, which we will include in revision:
>
> By the definition of Moreau Envelope, we have that
> $$
>  f \\leq \\tilde{f} \\implies \\tilde{f} \\geq 0
> $$
> and so if $f(\hat{y}, y) = 0$, there exists $u$ such that
> \begin{equation*}
> 	\begin{split}
> 		\tilde{f}(u, y) + \lambda(u-\hat{y})^2 = 0
> 		\implies \tilde{f}(u, y)  = \lambda(u-\hat{y})^2 = 0
> 		\implies u = \hat{y}, \quad \tilde{f}(\hat{y}, y) = 0.
> 	\end{split}
> \end{equation*}
> For any $w$ such that $\\hat{L}\_f(w) = 0$, by non-negativity it must be the case that $f(\\hat{y\_i}, y\_i) = 0$ for every $i$ and so $\hat{L}\_{\tilde{f}}(w) = 0$.
> Therefore we also have
> $$ \forall i, \quad \tilde{f}(\hat{y\_i}, y\_i) = 0 \implies \hat{L}\_{\tilde{f}}(w) = 0.$$

---

### Official Review · Reviewer_vsNX · 2022-07-11

**Rating:** 7
**Confidence:** 3
**Soundness:** 4 excellent
**Presentation:** 3 good
**Contribution:** 3 good

**Summary:**

This paper establishes generalization bounds for a broad class of losses over linear predictors by utilizing Moreau envelope theory. The authors show that particular quantities that depend on a localized Gaussian width can control the prediction error under Moreau envelopes of the loss, which can then translate to generalization bounds on the original loss as well. The setting is fairly broad, as it allows for models misspecification and more general loss functions. A number of previous results in the literature were extended.

**Questions:**

What is the intuition behind the use of the surrogate distribution in the main theorem (Theorem 1)? Does this aid in analyzing the interaction between the noise and the low-dimensionality of the data? Also, is the main hurdle in extending the Gaussian data assumption the use of the GMT framework for the analysis? I would also suggest a short discussion after the introduction of the Moreau envelope about why analyzing the Moreau envelope of the loss is useful. The results showcase the utility of the framework, but it could be good to provide some intuition beforehand.

**Limitations:**

Yes. No societal impacts need to be discussed.

**Strengths And Weaknesses:**

Originality: There has been significant interest in better understanding generalization properties, especially of interpolating solutions. To the reviewer's knowledge, the application of the Moreau envelope framework to tackling generalization appears novel.

Quality: The general theory provided here seems to give a number of improvements to previous results, including classical and recent ones. While the Gaussian data assumption is fairly restrictive, obtaining a complete understanding of this regime is still important.

Clarity: The paper is very well-written and easy to follow, even though it is quite technical. Here are some small comments:
- need a space between sentences in line 163
- the footnote at the top of page 8 takes you to page 1 instead of the bottom of the page

Significance: The work appears quite significant.

---

> ### Author Response · Authors · 2022-08-02
> **Author Response**
>
> Thank you for your feedback; we’ll incorporate your editing suggestions into revision. To answer your specific questions:
>
> > What is the intuition behind the use of the surrogate distribution in the main theorem (Theorem 1)? Does this aid in analyzing the interaction between the noise and the low-dimensionality of the data?
>
> Yes — the surrogate distribution encapsulates all of the information in the original problem which is relevant to the response variable $y$, i.e. its (arbitrary) dependence on the low-dimensional part of the data and the law of the noise. Theorem 1 is a mechanism to automatically reduce from proving generalization bounds in the original, potentially very high-dimensional space (which is hard to do, especially tightly) to proving these bounds for the low-dimensional surrogate distribution. In simple cases (e.g. where the noise is Gaussian and the model is well-specified) analyzing the surrogate distribution is straightforward; in more general cases, it may not be. For example, if the model has heavy-tailed noise, then the concentration of the training loss around the test loss becomes worse.
>
> Difficulties can arise even when there is no “noise” in the model: as an arbitrary example, if we have a one-dimensional model of $(x,y)$ where $$x \\sim \\mathcal N(0,1) \\qquad y = e^{A^2/2} \\mathbb{I}( x > A )$$, then for any fixed $n$, the training loss fails to concentrate around the test loss for sufficiently large $A$. (This is consistent with our Corollary 1, because this example has a large hypercontractivity constant $\\tau$.) Fortunately, there are many existing tools for analyzing generalization in low dimensions which we can appeal to, like VC theory.
>
> Also note that moving to the surrogate distribution, the test loss of the function with parameters $(\\tilde w, \\tilde b) := (\\phi(w), b) $ is the same as the test loss in the original distribution of the function with parameters $(w,b)$. So this map $(w,b) \\mapsto (\\tilde w, \\tilde b)$ is in some sense a compression that retains the information that is relevant for computing the test loss.
>
> > Also, is the main hurdle in extending the Gaussian data assumption the use of the GMT framework for the analysis?
>
> Yes, exactly; see our separate comment about Gaussianity for more.

---

> > ### Comment · Reviewer_vsNX · 2022-08-08
> > **Updated response**
> >
> > Thank you for your responses and comments regarding the review. After reading your responses and the other reviews, I will maintain my score. I think this is paper is solid in terms of its contributions and offers interesting insights for generalization theory.

---

### Author Response · Authors · 2022-08-02
**A comment on Gaussianity**

Since all reviewers asked about the limitation to Gaussian data, we give some further comments on that here (which will be incorporated into future revisions of the paper).

The limitation to Gaussian covariates is based on our key use of the Gaussian Minimax Theorem, which does not readily generalize to broader distributions. We do, however, have some indications that our results (or something close to them) should hold more broadly than only on Gaussian data, based on empirical evidence from our simulations in Appendix B; the question remains what techniques can prove this.

Others have in the past been able to generalize Gaussian-based results beyond that setting via suitable universality arguments: see e.g. “Universality laws for high-dimensional learning with random features” by Hong Hu and Yue Lu for a proof of the Gaussian equivalence theorem in the context of random features models; “A precise high-dimensional asymptotic theory for boosting and minimum-L1-norm interpolated classifiers” by Liang and Sur for results on universality of min-L1-norm interpolants in certain asymptotic settings; “Universality of empirical risk minimization” by Montanari and Saeed for results on near-empirical risk minimizers under a certain set of assumptions. In the general case of our setting, however, we do not yet know a nice way to extend our full framework.

---

### Meta-Review · Area_Chair_q1W3 · 2022-08-28

**Recommendation:** Accept
**Confidence:** Certain

**Metareview:**

The manuscript proves a new (finite-sample) generalization bound for generalized linear models, using Moreau envelope theory. The paper also provides experimental validation.

While the results only hold for Gaussian data, I believe there is some interesting novel results which might inspire some future work in the learning theory community. (In comparison, several novel frameworks have been created in the past for other problems, in which the initial versions assumed Gaussianity, e.g., the work on support recovery on sparse linear regression, for instance.)

Several technical clarifications regarding the assumptions (e.g., surrogate distribution, comparison to prior results) were asked by the reviewers, which the authors thoroughly and successfully addressed during the rebuttal phase. I recommend adding this to the camera-ready version of the paper, as well as other discussions and clarifications raised by all the reviewers.

**Award:**

No

---

### Decision · Program_Chairs · 2022-09-14

Accept